# Synergy and Diversity in CLIP: Enhancing Performance Through Adaptive Backbone Ensembling

**Cristian Rodriguez-Opazo**[1]    **Ehsan Abbasnejad**[1]    **Damien Teney**[2,1]
**Hamed Damirchi**[1]    **Edison Marrese-Taylor**[3]    **Anton van den Hengel**[1]
[1]Australian Institute for Machine Learning    University of Adelaide
[2]Idiap Research Institute, Switzerland    [3]The University of Tokyo
`{cristian.rodriguezopazo,ehsan.abbasnejad}@adelaide.edu.au`

## Abstract

Contrastive Language-Image Pretraining (CLIP) stands out as a prominent method for image representation learning. Various architectures, from vision transformers (ViTs) to convolutional networks (ResNets) have been trained with CLIP to serve as general solutions to diverse vision tasks. This paper explores the differences across various CLIP-trained vision backbones. Despite using the same data and training objective, we find that these architectures have notably different representations, different classification performances across datasets, and different robustness properties to certain types of image perturbations. Our findings indicate a remarkable possible synergy across backbones by leveraging their respective strengths. In principle, classification accuracy could be improved by over 40 percent with an informed selection of the optimal backbone per test example. Using this insight, we develop a straightforward yet powerful approach to adaptively ensemble multiple backbones. The approach uses as few as one labeled example per class to tune the adaptive combination of backbones. On a large collection of datasets, the method achieves a remarkable increase in accuracy of up to 39.1% over the best single backbone, well beyond traditional ensembles.

## 1    Introduction

Large pre-trained models are transforming machine learning, computer vision (Radford et al., 2021; He et al., 2022; Kirillov et al., 2023; Liu et al., 2023), and natural language processing (Devlin et al., 2018; Brown et al., 2020; Touvron et al., 2023). These models are typically trained with self-supervised objectives, eschewing the need for manual annotations and enabling the use of very large datasets (Jia et al., 2021; Schuhmann et al., 2022).

Contrastive Language–Image Pre-training (CLIP) (Radford et al., 2021) is one such approach that enables various downstream applications involving vision and language. CLIP learns to align text and image representations when training a pair of vision and language encoders, which can be implemented with various architectures. These encoders are then used in downstream applications to obtain representations of images and text, whose similarity can be simply evaluated with a dot product. This enables e.g. zero-shot classification (Zhai et al., 2022; Li et al., 2022a; Jia et al., 2021) and cross-modal retrieval (Li et al., 2020a;b; Yu et al., 2022).

Despite extensive prior work on CLIP, there is still a gap in comparing the representations from different vision backbones trained with this paradigm. Existing work has compared backbones for their generalization capabilities (Goldblum et al., 2023; Li et al., 2022a; Zhang et al., 2021; Gao et al., 2021), finding that larger architectures generally perform better. This paper contributes to this area with an empirical study that compares CLIP-trained backbones. We present new observations on how performance and robustness (e.g. invariance to image transformations) vary substantially across architectures. Within the same backbone family (e.g. different ViTs), different models present different patterns of performance across datasets (Figure 4). The relation with model scale is also more complicated than suggested in prior work (Figure 5).

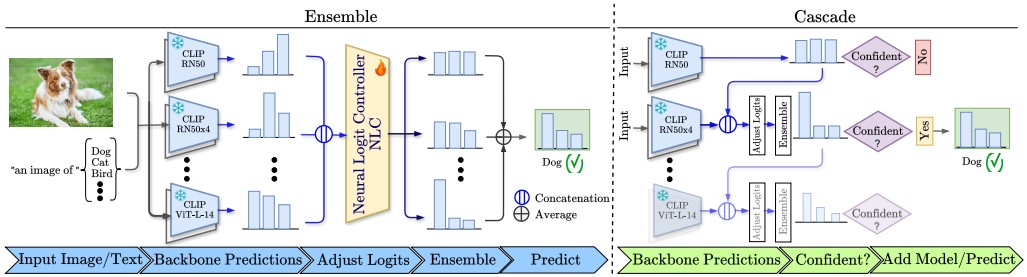

Figure 1: We propose a method to improve CLIP's effectiveness for image classification by combining the strengths of different backbones. **(Left)** For a given test image, the logits from different backbones are combined with a temperature scaling that weights their contribution to the final prediction. The scaling is implemented in the Neural Logit Controller (NLC, a small MLP) that is learned from as little as one labeled example per class. **(Right)** To reduce the computational load, our method can be combined with the Cascade framework (Wang et al., 2022a).

Our empirical findings suggest that CLIP's effectiveness for image classification could be improved by combining the strengths of different backbones. We thus present an ensemble method that combines backbones adaptively to each test example, using a temperature scaling that weights each backbone's contribution to the final prediction condition on the input image (see Figure 1).

A naive combination of backbones with traditional ensembling techniques (Lakshminarayanan et al., 2017; Dietterich, 2000) (i.e. averaging the outputs) does not consistently enhance generalization performance. These approaches focus on prediction agreements rather than leveraging diversity. In contrast, our method uses a temperature scaling based on the idea of calibrating each backbone's confidence. This intuition is further supported by the "modality gap" and variation in performance observed in prior work when adjusting CLIP's logits (Shi et al., 2023). We compare our approach to more advanced ensembles, such as SuperLearner Cheng Ju and van der Laan (2018), which also combines the logits of multiple models by learning temperature scaling. However, unlike our method, SuperLearner does not adaptively adjust these temperatures based on the input features, implying that SuperLearner cannot exploit the diversity of predictions (Tab. C.1.)

Our proposed method, named the Neural Logit Controller (NLC), uses a few labeled examples (as little as one per class) to tune the combination of backbones. Hence, we benchmark it against the state-of-the-art methods in a few-shot setting (Zhang et al., 2021; Radford et al., 2021; Zhou et al., 2022; Gao et al., 2021). NLC shows remarkable performance and consistently outperforms other approaches. Furthermore, we show that NLC complements existing methods like Tip-

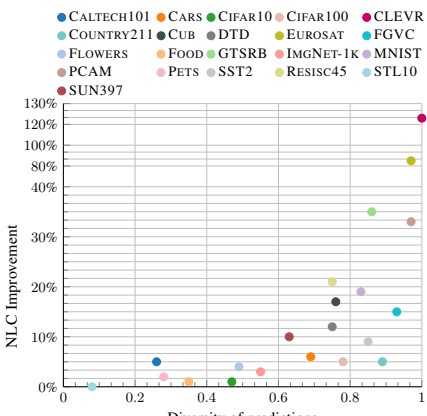

Figure 2: **(Y axis)** Relative improvement of NLC over best backbone vs. **(X axis)** predictions diversity [1]. NLC always improves over the best backbone. Moreover, the clear correlation shows that higher diversity in predictions tends to result in greater improvements with NLC.

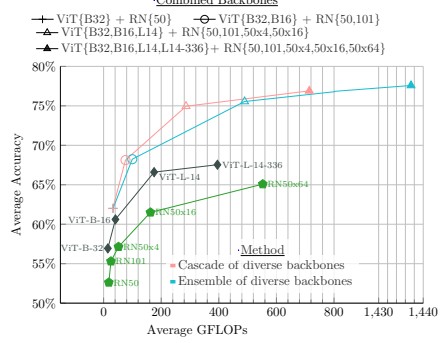

Figure 3: Average accuracy across 21 datasets for zero-shot ResNets and ViTs backbones, NLC Ensemble, and Cascade using 2 to 9 backbones, plotted against the Average GFLOPs. Demonstrates that ensembles can surpass the best zero-shot backbone with fewer GFLOPs.

---

[1]The diversity of predictions is measured as: $1 - \frac{\text{\# samples correctly predicted by } \textit{all} \text{ backbones}}{\text{\# samples correctly predicted by } \textit{any} \text{ backbone}}$.

Adapter (Zhang et al., 2021). Combining it with NLC in its original experimental framework yields improvements in accuracy of over 15%.

Finally, in-depth experiments reveal a clear correlation between NLC's improvement over the best backbone and the diversity of correct predictions in each dataset (see Figure 2). It achieves over 120% of relative improvement when the diversity of predictions is close to one, and never degrades the performance of the best backbone. In terms of efficiency, NLC outperforms the best backbone by combining only the top-four most efficient backbones, while using approximately 300 fewer GFLOPs (see Figure 3). We also show how to use the Cascade framework (Wang et al., 2022a; Varshney and Baral, 2022) with NLC to enhance performance while maintaining computational requirements within the bounds of the original backbones.

Our contributions are summarized as follows.

- We perform extensive experiments across 21 datasets that reveal diverse predictions across CLIP backbones and distinct robustness properties to image transformations.
- We evaluate the complementarity of backbones by measuring the potential improvement in classification accuracy (up to 43.5%) of an optimal oracle selection of backbone per test example.
- To leverage this complementarity, we propose an adaptive ensembling method (NLC) that uses temperature scaling and requires a single labeled example per class. It improves accuracy over the best backbone by 9.1% on average, surpassing previous ensembling frameworks.
- We demonstrate significant advantages in computational efficiency. Combining the top four most efficient backbones enables NLC to outperform the best backbone with ∼300 fewer GFLOPs. Integrating NLC with the Cascade approach (Wang et al., 2022a) maintains computational efficiency.
- We demonstrate that NLC consistently outperforms state-of-the-art few-shot methods. Moreover, integrating NLC with Tip-Adapter (Zhang et al., 2021) gives the latter a performance boost of over 15%.

## 2 INVESTIGATING DIFFERENCES ACROSS VISION BACKBONES

**CLIP for zero-shot image classification.** CLIP (Radford et al., 2021) is a general technique to train a pair of vision and text encoders $\phi_l$ and $\phi_v$ on paired image/text data. The method uses a self-supervised objective to align the encoded representations of matching images and text descriptions. Any image and text can then be mapped to a unified semantic space. This enables zero-shot image classification by computing a compatibility score between an image and a class description ("prompt") as $\text{score}(x, l) = \phi_v(x)^\top \phi_l(l)$. We follow previous work (Menon and Vondrick, 2023) for the generating of prompts e.g. `an image of {label}, which is a {concept}`. Subsequently, we calculate the compatibility score between the image feature $\phi_v(x)$ and the prompt representations $\phi_l(y)$, where $y$ represents a label within the class set. The prompt with the highest similarity is then chosen as the label for the given image.

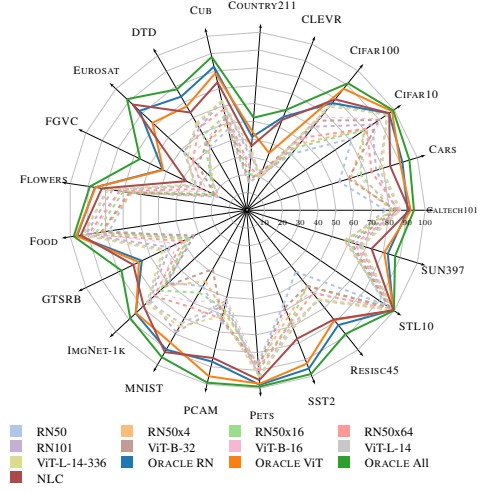

Figure 4: Zero-shot classification accuracy of various CLIP models on 21 datasets, and of the upper-bound "ORACLE" combination of ResNets (RN), ViTs, and all backbones.

### 2.1 COMPLEMENTARITY OF DIFFERENT BACKBONES

The vision encoder or "backbone" trained with CLIP can be implemented using various architectures. We want to identify whether different backbones have distinctive behaviours. We analyze the output of 9 of them in a zero-shot classification setting on 21 standard benchmark datasets (see Section 4 for details). We propose two approaches to assess the differences in their zero-shot predictions and their possible complementary behaviour. First, we define an ORACLE prediction that combines the output

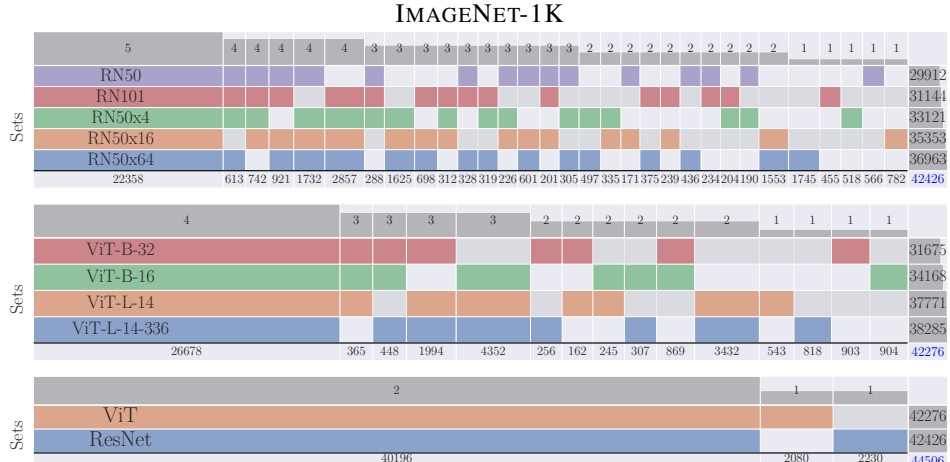

Figure 5: Linear Venn diagrams showing the overlap of test images from ImageNet-1k correctly classified by different backbones (rows). Each column represents a subset of images correctly classified by a specific group of backbones (group size in column header). Row/column sums indicate the number of correct predictions per backbone/subset. We observe that **(1)** different backbones agree on a large part of the data. **(2)** They also make additional correct predictions on different subsets. **(3)** Accuracy usually grows with architectures size, but even within a same family (ViTs, ResNets), different models show different patterns of (in)correct predictions.

of the 9 models on a per-image basis, using a correct prediction if any of the models is correct, any other otherwise. This simulates a scenario where an informed choice of the optimal backbone could be made for each test image to obtain the highest classification accuracy. Second, we use overlap diagrams to visualize the patterns of (in)correct predictions across backbones and their different strengths on different subsets of images.

Figure 4 and Table B.1 summarize the performance of the different backbones. Note that they are pre-trained using the exact same dataset and objectives, as provided in the OpenClip project (Ilharco et al., 2021). Our results show that backbones from different families (ViTs vs. ResNets) show significant differences while controlling for model size (number of parameters). For example, ViT-B-32 outperforms ResNet-50x4 in the CIFAR10 and CIFAR100 benchmarks. However, this trend is not consistent across all datasets, e.g. on IMGNET-1K. Additionally, the results indicate that backbones with more parameters do not always outperform smaller ones.

We then assess the ORACLE predictions using backbones from the same family (ORACLE RN for ResNets, ORACLE ViT for ViTs, and ORACLE All for a combination of all backbones). The ORACLE All obviously outperforms the best individual backbone in every case. For instance, on EUROSAT, CLEVR, CUB, DTD, CIFAR100, and CARS, the ORACLE All shows improvements of 43.5%, 36.0%, 25.1%, 21.8%, 16.6% and 16.0%, respectively. **The large magnitude of these improvements highlights the complementarity of backbones. In other words, the correctly-classified data is not simply growing as one considers larger and better models. Instead, different backbones perform well on different subsets of the data.** This insight is the key motivation for the ensemble method we propose in Section 3. We also found that similar findings have been made in domains other than ours Roth et al. (2024); Zhong et al. (2021); Ramé et al. (2022).

We visualize in Figure 5 the overlap between different subsets of IMGNET-1K correctly predicted by different backbones. We observe comparable accuracy between the largest ViT (ViT-L-14-336, correct predictions) and ResNet (RN50x64) with $38,285$ vs. $36,963$ correct predictions. However, there are significant differences within each family: for example, among ResNets, only $22,358$ images are correctly classified by all ResNets models, out of the $42,426$ ones correctly classified by *any* ResNet.[1]

We further observe that every model has unique strengths by examining the performance of the oracle models. Specifically, ORACLE RN, ORACLE ViT, and ORACLE All increase the number of correct predictions to $42,426$, $42,276$, and $44,506$ out of $50,000$. Each backbone exhibits correct predictions unique to itself. In the ORACLE RN, RN50, RN101, RN50x4, RN50x16, and RN50x64 contribute 566, 455, 518, 782, and $1,745$ exclusive predictions, respectively. In the ORACLE ViT,

B-32, B-16, L-14, and L-14-336 present $903$, $904$, $543$, and $818$ unique predictions, respectively. Finally, when all backbones are used, ResNet contributes $2,230$ exclusive predictions, and ViT contributes $2,080$. It is important to note that correct predictions of any given model are not simply a subset of those by the best model in its family.

## 2.2 OTHER SOURCES OF DIVERSITY

To better understand the effects of backbone complementarity, and contrast these effects against the role of variables that are known to drive performance in ensembles such random parameter initialization and learning via stochastic gradient descent, we propose to evaluate the diversity of correct predictions when combining multiple backbones, which as seen in our experiments above (Figure 2) strongly correlates with ensemble performance gains.

Concretely, we isolate the effects of three variables on prediction diversity: (1) backbone architecture, (2) pretraining dataset, and (3) number of training steps. For (1), we evaluate the complementarity of ViT-L-14, ViT-B-32, and ViT-B-16 backbones using the same dataset. For (2), we analyze each ViT's performance using different pretraining datasets (LAION-400M vs. OpenCLIP). For (3), we consider ViTs pretrained on LAION-400M using the checkpoints at two different epochs (31 and 32).

Since, to the best of our knowledge, there are no publicly released CLIP models that differ only on their initialization, studies on this variable would, in principle, require us to train CLIP backbones from scratch, which is extremely compute-intensive.

Our diversity metric considers the aggregate of test examples correctly classified and it is the ratio between the instances correctly classified by all backbones versus the instances correctly classified by any backbone [1]. It can be thought of as (1 - IoU), where IoU is the "intersection over union" of $n$ sets of correctly predicted samples, where $n$ is the number of methods considered. If every sample in this aggregate is correctly classified by all backbones (i.e. they all agree while being correct), *diversity=0*. Otherwise, if every sample can only be correctly classified by a single backbone, *diversity=1*. This allows us to evaluate the complementarity of models.

Figure 6 (Tab. I.2) demonstrates that combining multiple backbones achieves higher average diversity than using the same backbone across different datasets or nearby

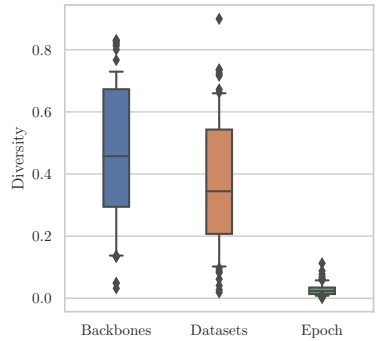

Figure 6: Diversity of predictions of (1) different **Backbones** with same pretrained dataset, (2) same backbone with different pre-trained **Datasets**, and (3) same backbone and same pre-trained dataset in two different **Epoch**s. Results show complementarity is higher when we combine different backbones.

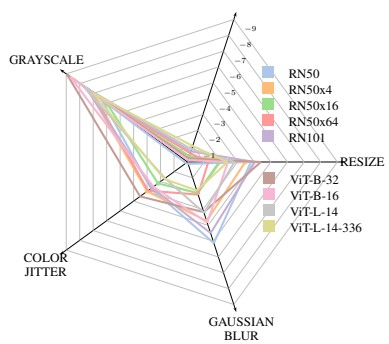

Figure 7: Impact on test classification accuracy of image transformations. Smaller values (closer to center) mean higher robustness, suggesting models are particularly robust to specific transformations.

solutions in the loss landscape. This indicates that training stochasticity and pretraining dataset are not the primary factors. Instead, the most meaningful complementarity comes from diverse backbones, as they excel on different data subsets, as confirmed by the ORACLE results in Table I.1.

## 2.3 ROBUSTNESS TO IMAGE TRANSFORMATIONS

To explore qualitative differences in behaviour of different backbones, we selected a subset of each benchmark's test set and applied specific image transformations (`resize`, `flip`, `grayscale`, `color jitter`, and `Gaussian blur`). We then evaluate the zero-shot classification performance of each backbone on these corrupted images. Figure 7 shows that ResNet 50 and ViT-B-16 are the most resilient to `flip`, RN50x64 to `resize`, and RN50x16 to `grayscale`. ViT-L-14 and

L-14-336 are the most robust to `color jitter` and `Gaussian blur`. These findings suggest that the backbones' ability to deal with out-of-distribution images varies widely. Certain models prove particularly robust to specific transformations.

## 3 PROPOSED ENSEMBLING METHOD

The previous section showed that different backbones have different strengths and perform well on different types of data. We now use this insight to improve classification performance by combining multiple CLIP backbones.

**Combining models by temperature scaling.** Our approach is technically inspired by prior work on network calibration, which we apply to an ensemble of models. Specifically, our technique can be seen as a variation of **Platt scaling** (Platt et al., 1999). This classical method uses the logits of a model as features for a logistic regression, which is trained on the validation set to produce calibrated probabilities. More precisely, given logit scores $z_i$ for an example $i$, Platt scaling learns two scalars $a$ and $b$ and produces calibrated probabilities as $\hat{q}_i = \sigma(az_i + b)$. The parameters $a$ and $b$ can be optimized using the negative log likelihood (NLL) loss over the validation set, with the model weights being frozen during this process.

A common use of Platt scaling is known as **temperature scaling** (Guo et al., 2017). In this approach, a single scalar parameter $t > 0$ is used for all classes of a given model. The new, calibrated confidence prediction is given by $\hat{q}_i = \max_k \text{softmax}(z_i/t)^{(k)}$, where $t$ is called the temperature, $z_i$ are the logits for example $i$ returned by the uncalibrated model. Usually, $t$ is optimized with respect to the NLL on the validation set aiming to reduce the overconfidence of the model on its predictions and to produce more reliable predictions, but because the parameter $t$ does not change the maximum of the softmax function, the class prediction $\hat{y}_i$ remains unchanged, meaning that the performance of a given model remains the same.

**Learning temperature coefficients.** In our method, we aim to jointly optimize a set of temperature parameters $t_b$ with $b \in [1, \ldots, B]$ for a set of $\mathbb{B}$ backbones. The aim is to combine their predictions by adjusting each backbone's confidence depending on the confidence of the others, and the input example. We learn the temperatures $t_b$ that weigh the logit $z_i^b$ for a backbone $b$ and example $x$ using the cross-entropy loss, and then we combine the logits via a weighted sum.

Concretely, we train a one-layer MLP (our Neural Logit Controller) to predict the set of temperatures that best calibrate the backbone mixture. The MLP takes as input the concatenated representations obtained by passing the images through the encoder $\phi_v$ of each backbone $b \in \mathbb{B}$. As depicted in Figure 1, The MLP directly produces a vector of temperatures $t \in \mathbb{R}^B$ and is trained on a holdout set for the training set of each target dataset using the cross-entropy loss between final predictions and ground truth labels.

## 4 EXPERIMENTS

**Backbones.** We consider the original selection of backbones described by Radford et al. (2021). This includes the ResNets (He et al., 2016) RN50, RN50x4, RN50x16, RN50x64 and RN101, and the ViTs (Dosovitskiy et al., 2021) B-16, B-32, L-14 and L-14-336. All models are obtained through the open-source project OpenCLIP (Ilharco et al., 2021).

**Datasets.** We use a selection of 21 popular image classification datasets: CALTECH101 (Li et al., 2022b), CARS (Krause et al., 2013), CIFAR10 (Krizhevsky et al., 2009), CIFAR100 (Krizhevsky et al., 2009), CLEVR (Johnson et al., 2017), CUB (Wah et al., 2011), DTD (Cimpoi et al., 2014), EUROSAT (Helber et al., 2018), FGVC (Maji et al., 2013), FLOWERS (Nilsback and Zisserman, 2008), FOOD (Bossard et al., 2014), GTSRB (Houben et al., 2013), IMGNET-1K (Deng et al., 2009) MNIST (Deng, 2012), PCAM (Veeling et al., 2018), PETS (Parkhi et al., 2012), RenderedSST2 (Socher et al., 2013), RESISC45 (Cheng et al., 2017), STL10 (Coates et al., 2011) and SUN397 (Xiao et al., 2010). They include diverse images of nature, animals, places, medical scans, satellite images, and man-made objects. The evaluation metric is simply the accuracy of each dataset's test split. The aggregated performance is the average accuracy across all datasets weighted equally.

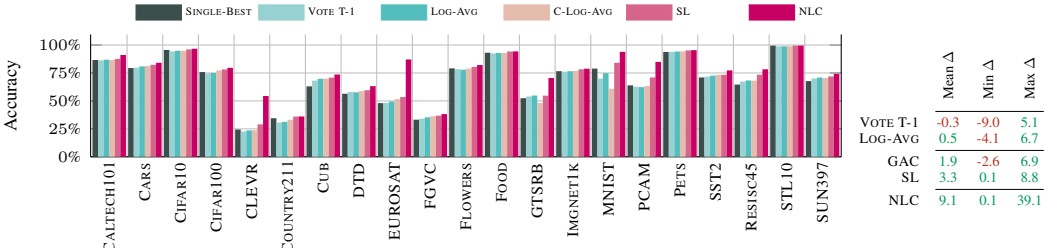

Figure 8: Comparison of the top-performing single backbone (SINGLE-BEST) with NLC and other ensemble strategies on top of zero-shot CLIP backbones. In the table, Mean, Max and Min Δ summarize the difference in performance across datasets with respect to the SINGLE-BEST backbone.

**Baselines.** Our experiments consider several ensembling baselines. Non-parametric approaches:

- **Logit averaging** (LOG-AVG) simply takes the average of the logits (scores) of the models.

- **Voting** (VOTE T-1) uses a majority vote. Each model casts a vote for its max-scored class. The final prediction is the class with the highest number of votes. Compared to LOG-AVG, this does not use the *soft* score values and the uncertainty reflected therein.

Parametric approaches:

- **Calibrated logit averaging** (C-LOG-AVG) first calibrates each model independently using temperature scaling. They are then combined by simply averaging their logits.

- **Super Learner** (SL) (Cheng Ju and van der Laan, 2018) a flexible ensemble learning framework that optimally combines predictions from multiple base learners. In contrast to our method, SuperLearner learns temperature scaling factors without considering the input. Thus, finding the best scaling for the total validation set.

- **Mixture of experts** (MOE) is a popular method to improve performance by combining specialized submodels of experts (Chen et al., 2022; Lepikhin et al., 2020; Rau, 2019; Zoph et al., 2022), each one of which focuses on a specific input region. A gating network determines which expert is relevant for a given input. We use the Sparse MoE implementation (Zoph et al., 2022) by Rau (2019), where the MoE layer's input is the concatenation of vision features $x_b$ from $\phi_b$. We use standard hyperparameters: 9 experts trained for classification with cross-entropy loss with Adam (Kingma and Ba, 2014) and a learning rate of $2e-5$ for 300 epochs.

See Appendix C a discussion of additional baselines. The appendix also contains additional results on the combination of backbones and how they perform under distribution shifts on the IMGNET-1K dataset (Section E). Finally, Section H presents dataset-specific results including overlap diagrams, an analysis of possible model combinations using NLC and our ORACLE, as well as the dataset-specific learned temperature values.

## 4.1 COMPARISON OF BACKBONE-ENSEMBLING METHODS

In this section, we evaluate the proposed NLC approach along with various non-parametric and parametric ensembling techniques. The non-parametric ones are "static" while the parametric ones use the **training split** of each target dataset to adjust the ensemble adaptively. The following results will show the importance of this adaptive setting to leverage the backbones' respective strengths, and the superiority of our method over baseline parametric techniques.

Figure 8 and Table C.1 show the performance of each technique. Among **non-parametric baselines**, LOG-AVG fails to enhance overall performance beyond the best backbone. One notable exception is the COUNTRY dataset where LOG-AVG shows a substantial improvement over the best backbone. Overall, though, simple score averaging does not seem to exploit the complementarity of the backbones, with an improvement in accuracy of only +1.3%. When the backbones are calibrated with C-LOG-AVG, performance improves compared to the non-calibrated version, across all datasets except GTSRB, MNIST, RESISC45, and SUN397. **Vote T-1** is mostly ineffective and does not significantly improve over the best backbone. Among **parametric methods**, the proposed NLC shows the most substantial improvement, up to +39.1% on the EUROSAT dataset. On average, we obtain a commendable improvement of +9.1% over the SINGLE-BEST backbone across all datasets.

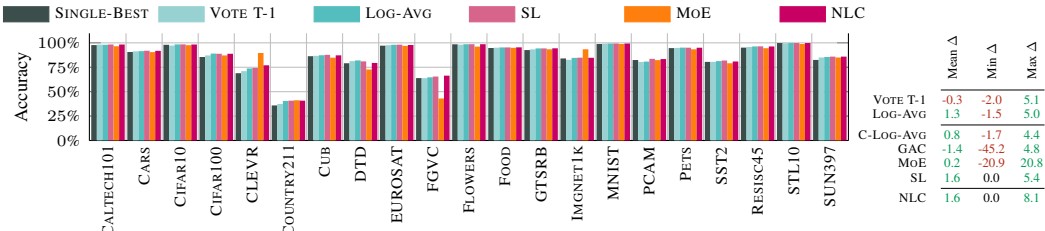

Figure 9: Comparison of the top-performing single backbone (SINGLE-BEST) with NLC and ensemble strategies on top of CLIP backbones linear classifier, where Mean, Max and Min Δ summarize the difference in performance across datasets with respect to the SINGLE-BEST.

## 4.2 INVESTIGATING BENEFITS BEYOND DATASET-SPECIFIC LINEAR CLASSIFIERS

We now design an additional experiment to investigate the benefits of the proposed NLC. A standard approach to adapt CLIP is to train a dataset-specific linear classifier on frozen visual features. In this section, we first train such linear classifiers, using each backbone's frozen features and each dataset's training split. Whereas NLC normally acts on raw CLIP scores, we now train it to act on the logits from these classifiers instead. Since the linear classifiers already are strong dataset-specific models, one might expect that only small additional improvements are possible. Yet, the following results will show that the NLC obtains further performance gains thanks to the *instance*-level adaptation. In contrast to alternative approaches, it also never reduces the performance compared to the best single linear classifier.

**Setup.** We follow the setup of Li et al. (2022a). We initialize the weights of the linear classifiers using language weights, which was shown to be more stable than a random initialization. The output of each backbone $\phi_b$ is L2-normalized before being passed to each linear classifier, which is trained using 90% of the target dataset's training split. The remaining 10% are used to train NLC.

**Results.** Figure 9 and Table D.2 present the results combining linear classifiers with both non-parametric and parametric approaches. Similar to Section 4.1, the proposed NLC consistently improves performance across all datasets, with an average improvement of +1.6% over the SINGLE-BEST classifier. As expected,

Table 1: Accuracy (%) in a few-shot setting

| Few-shot | 1 | 2 | 4 | 8 | 16 |
|---|---|---|---|---|---|
| Linear-probe CLIP Radford et al. (2021) | 22.2 | 31.9 | 41.2 | 49.5 | 56.1 |
| CoOP Zhou et al. (2022) | 47.6 | 50.9 | 56.2 | 59.9 | 63.0 |
| CLIP-Adapter Gao et al. (2021) | 61.2 | 61.5 | 61.8 | 62.7 | 63.6 |
| Tip-Adapter Zhang et al. (2021) | 60.7 | 61.0 | 61.0 | 61.5 | 62.0 |
| Tip-Adapter-F Zhang et al. (2021) | 61.3 | 61.7 | 62.5 | 64.0 | 65.5 |
| NLC | **78.2** | **78.1** | **78.2** | **78.3** | **78.4** |

the improvement is reduced compared to the original NLC since some of the gains are now realised by the dataset-specific linear classifier. Yet Table D.1 clearly shows that additional gains can be made over the linear classifiers, i.e. that there remains an exploitable complementarity across backbones. By examining the ORACLE performance in this setting, we can see that the potential benefits of combining multiple backbones remain. Interestingly, the MOE approach does not reach the performance level of the best backbone on several datasets. It shows a degradation sometimes down to −20.9%. This suggests that MoEs may struggle to partition the input space effectively into distinct clusters, specific to certain experts, potentially limiting its functionality.

## 4.3 COMPARISON WITH FEW-SHOT ADAPTER METHODS

We now examine the performance of the proposed NLC in a few-shot setting. We also show that it complements existing few-shot adapter methods and improves their accuracy across various datasets.

We compare our method against Tip-Adapter, Tip-Adapter-F (Zhang et al., 2021), few-shot Linear-probe CLIP (Radford et al., 2021), CoOP (Zhou et al., 2022), and CLIP-Adapter (Gao et al., 2021). Linear-probe CLIP fine-tunes a linear classifier on a few-shot training set on the frozen CLIP backbones. CoOP (Zhou et al., 2022) generates different prompt designs to make prompts learnable. CLIP-Adapter (Gao et al., 2021) enhances few-shot classification by introducing a feature adapter on CLIP's visual and textual encoders. Tip-Adapter (Zhang et al., 2021) achieves comparable performance to CLIP-Adapter without requiring training, using a key-value cache model from the

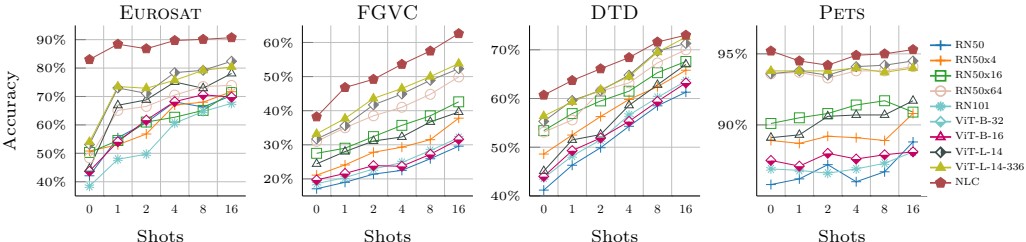

Figure 10: Peformance of TiP-Adapter applied to different zero-shot CLIP backbones, and the combination of these with NLC, showing how our method provides complementary benefits.

few-shot training set. Tip-Adapter-F can further enhance performance by fine-tuning the cache. As shown in Table 1, our NLC shows an outstanding performance over compared methods and consistently surpasses other few-shot methods by learning how to combine multiple backbones, which is a complementary approach to any of these adapters. In the appendix, Table F.2 shows that current few-shot adapter methods also have performance differences with various backbones, and still, our NLC surpasses the best backbone reported by these existing methods.

Finally, we integrate Tip-Adapter (Zhang et al., 2021) with our NLC and conduct experiments on EUROSAT, FGVC, DTD, and PETS. Initially, we apply Tip-Adapter and Tip-Adapter-F independently on each backbone, following the protocol from Zhang et al. (2021) with 1, 2, 4, 8, and 16 shots. Subsequently, we employ our ensembling mechanism to fuse the adapted backbones. To train NLC, we follow their setting and use the same validation split as used by Tip-Adapter to linearly combine their logits with CLIP.

See Figure 10 and Tables F.3–F.6 for a comparison of the performance of Tip-Adapter over different zero-shot CLIP backbones, and the combination of all these Tip-Adapter versions with NLC. Across the four evaluated datasets, NLC improves over each version of Tip-Adapter in the few-shot setting. Notably, NLC improves the performance of the best Tip-Adapter backbone (L-14-336) of up to 10% for EUROSAT using 16 shots.

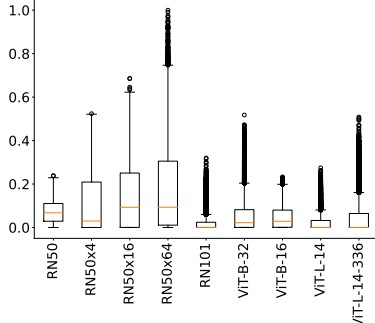

Figure 11: Distribution of max-normalized temperature values learned by our NLC method on the CLEVR dataset.

Moreover, when we compare NLC and SL in the Tip-Adapter setting, NLC shows better performance, Table F.3–F.6.

### 4.4 EXAMINING THE LEARNED TEMPERATURE VALUES

We visualise the learned temperature values to understand how the NLC method assigns weights to different backbones, revealing its strategy for leveraging the strengths of diverse architectures. Figure 11 shows the learned temperature values for NLC on the CLEVR dataset (see Section H for other datasets). By examining the learned temperatures, we can gain insight into the contribution of each backbone. We note that the backbone with the largest overall weight is not consistently the deepest one, such as ResNet-101. The uniform distribution of learned temperatures across backbones indicates that NLC effectively utilizes the strengths of multiple backbones, resulting in a more balanced and accurate labeling process, complementing our analysis in Figure 2. These findings highlight that NLC benefits from the diversity in model predictions, reinforcing the advantage of combining various backbones.

## 5 RELATED WORK

**Foundation models in computer vision.** The trend in computer vision over the past decade has been to train larger and larger models on increasingly diverse tasks and increasingly larger datasets (Gadre et al., 2023; Schuhmann et al., 2022). Scale and generality have proved to be key enablers to the performance and robustness of these models (Fang et al., 2022; Gan et al., 2022; Mayilvahanan et al., 2023; Santurkar et al., 2022; Tu et al., 2024). Various methods have been proposed to pre-train such

models (He et al., 2022; Radford et al., 2021; Yu et al., 2022). CLIP (Radford et al., 2021) is a notable one, specifically designed to exploit noisy image-text pairs scraped from the internet. CLIP's significance lies in its scalability and its ability to generate meaningful alignments with prompts, facilitating zero-shot classification gao2021clip,zhang2021tip,li2022elevater. CLIP's popularity is due to its scalability and ability to handle diverse text prompts that enable a variety of downstream applications (Gao et al., 2021; Zhang et al., 2021; Li et al., 2022a). Various versions of CLIP have been trained on different datasets and vision backbones such as ResNets (He et al., 2016) and ViTs (Dosovitskiy et al., 2021).

**Comparing vision backbones**. Goldblum et al. (2023) recently conducted a comprehensive comparison of pretrained backbones across various tasks including image classification, object detection, segmentation, and image retrieval. In comparison, this paper presents complementary observations of the strengths of different backbones on different datasets and types of data. Moreover, we leverage these findings with a novel ensembling approach this complementarity of different backbones. Other works investigating differences across vision architectures include (Angarano et al., 2022; Pinto et al., 2022; 2021; Wang et al., 2022b) and others specific to CNNs (Abello et al., 2021; Hermann et al., 2020) and ViTs (Naseer et al., 2021). Finally, we find that there are works analysing the diversity of models for knowledge distillation Roth et al. (2024), complementarity on models trained don different subset of data Ramé et al. (2022) and Zhong et al. (2021) exploring the diversity on LLMs.

**Model ensembling.** Combining multiple machine learning models is a classical approach for improving predictive performance. Simply averaging the outputs of several models is a simple, effective technique (Bauer and Kohavi, 1999; Breiman, 1996; Dietterich, 2000; Lakshminarayanan et al., 2017) with studies dating back more than three decades ago. The approach has also been applied to deep neural networks (deep ensembles (Lakshminarayanan et al., 2017)), which has shown benefits in various domains including higher accuracy under distribution shift (Ovadia et al., 2019; Teney et al., 2018). The diversity of the combined models (in terms of uncorrelated errors) has also been shown to be critical to these improvements (Hao et al., 2024; Wortsman et al., 2022). The closest to our work is the SuperLearner framework (Cheng Ju and van der Laan, 2018), which similarly investigates multiple classical ensembles and introduces a new ensemble that learns temperature scaling. However, our study not only evaluates the performance of various classical ensembles but also explores the complementarity of CLIP backbones across 21 datasets. Furthermore, unlike SuperLearner, our method adapts its temperature scaling factors based on the input, which plays a crucial role in ensembling CLIP backbones.

## 6 Conclusions

This paper presents an analysis of vision backbones in the CLIP framework, focusing on the task of image classification. Unlike prior studies that span various downstream tasks (e.g. Goldblum et al. (2023)) our emphasis lies in identifying the unique strengths of various backbones. Our experiments revealed a distinctive complementarity across architectures and an avenue for enhancing CLIP's performance by synergistic combination. We proposed an ensemble approach that reweights the logits from each backbone condition on the input data to yield more accurate predictions in image classification across a variety of datasets.

**Limitations.** First, our evaluation focused mostly on image classification, even though CLIP can be used for a variety of downstream tasks. Although we test our method on out-of-distribution Sec. E and also obtain the upper bound performance on Image-Text retrieval Sec. K, more work needs to be done on how to combine the representations. Second, our approach relies on a late fusion of backbones by adaptively adjusting their logits. Alternative approaches that leverage the same initial motivation use knowledge distillation, which requires training the student backbone again Roth et al. (2024). Third, although we employed the Cascade method (Wang et al., 2022a) to reduce the computations, our method still needs to pass a test image multiple times through vision encoders. This adds complexity and computational overhead, which can be important if a large number of backbones are considered. A possible solution would involve fusing backbones at the training stage to require only a single forward pass at test time.

**Future work.** We could emulate the behavior of a given backbone by learning their representations using the main branch's early layers. Other efficient methods could also be developed to combine predictions from multiple backbones to enhance the scalability of the idea.

ACKNOWLEDGE

This research is funded in part by the Australian Government through the Australian Research Council (Project DP240103278) and the Centre of Augmented Reasoning at the Australian Institute for Machine Learning, established by a grant from the Department of Education. We thank the NVIDIA corporation for donating one of the GPUs used for the experiments. Finally, we would like to thank Stephen Gould for his valuable feedback on the paper.

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

## A    REPRODUCIBILITY

The one-layer MLP is trained with ADAM optimizer with a learning rate of 2e-4. We use a weight decay of 0.01. It receives as input the concatenation of the features, the hidden layer has a width of 128 and its outcome is the temperature values for the used backbones. All the backbones used in the paper are pre-trained using the same dataset called `openai` and objectives from the OpenAI Fundation `https://github.com/mlfoundations/open_clip`. In the Cascade Wang et al. (2022a), we use the probability as the confident measurement with a threshold of 0.9. When the method is confident with a probability bigger than 0.9 we produce a final prediction, otherwise, we add the next backbone. All our experiments are done in one NViDIA GeForce RTX 4090 GPU, in a small server with 64 GB of RAM and 32 CPU cores.

## B    ORACLE PERFORMANCE

Table B.1 presents the results depicted in Figure 4 from the main paper. It includes the number of parameters and GFLOPs/Image for each backbone, the performance of zero-shot CLIP (OpenAI) on the selected datasets, and the performance of the proposed ORACLE method.

We observe that backbones with a similar number of parameters from different families can show significant differences in performance. For instance, ViT-B-32 outperforms ResNet-50x4 in the CIFAR10 and CIFAR100 benchmarks. However, this trend is not consistent across all datasets, as illustrated by the results on IMGNET-1K. Additionally, a higher number of parameters does not always correlate with better performance across different backbone families.

Notably, the ORACLE method, which leverages all possible backbones, shows substantial improvements on several datasets: 43.5% on EUROSAT, 36.0% on CLEVR, 25.1% on CUB, 21.8% on DTD, 16.6% on CIFAR100, and 16.0% on CARS. This clearly demonstrates the complementarity of each zero-shot CLIP backbone. In Table B.2, we present the results of the artificial image variations shown in Figure 7. In this experiment, we selected a subset of each test set from the benchmarks and applied specific artificial image variations -*i.e.*, Resize, Flip, Grayscale, Color Jitter, and Gaussian Blur- to evaluate the zero-shot classification performance on these "corrupted" images for each backbone.

The results indicate that ResNet-50 and ViT-B-16 are the most resilient to the Flip transformation, RN50x64 shows the highest resilience to Resize, and RN50x16 performs best for Grayscale. ViT-L-14 and L-14-336 exhibit the most robustness against Color Jitter and Gaussian Blur transformations. These findings suggest that each backbone has a unique ability to handle specific types of image variations.

## C    ADDITIONAL BASELINES

To complete the experiments and establish a comprehensive baseline compared to other approaches, we introduce two non-parametric and one parametric more ensemble methods to our benchmarks:

Table B.1: Zero-shot accuracy performance comparison of CLIP backbones on 21 datasets, showcasing the number of parameters and GFLOPs per image for each backbone. Additionally, explore the performance of our empirical upper-bound prediction, denoted as 'ORACLE', representing the ideal combination of ResNet (RN), ViT, and All backbones.

| | | Num Parameters | GFLOPs/Image | CALTECH101 | CARS | CIFAR10 | CIFAR100 | CLEVR | COUNTRY211 | CUB | DTD | EUROSAT | FGVC | FLOWERS | FOOD | GTSRB | IMGNET-1K | MNIST | PCAM | PETS | SST2 | RESISC45 | STL10 | SUN397 |
|---|---|---|---|---|---|---|---|---|---|---|---|---|---|---|---|---|---|---|---|---|---|---|---|---|
| CLIP-ResNet | 50 | 102.0M | 18.2 | 77.9 | 54.2 | 71.5 | 40.3 | 21.6 | 15.4 | 46.6 | 41.2 | 28.2 | 17.1 | 66.1 | 77.9 | 35.1 | 59.8 | 49.1 | 63.9 | 85.8 | 56.6 | 44.0 | 94.2 | 58.6 |
| | 101 | 119.7M | 25.5 | 81.8 | 61.1 | 80.8 | 47.6 | 24.4 | 16.9 | 49.6 | 43.7 | 26.4 | 18.6 | 65.2 | 81.9 | 37.5 | 62.3 | 46.0 | 58.1 | 86.9 | 63.9 | 54.5 | 96.8 | 57.2 |
| | 50x4 | 178.3M | 51.8 | 82.0 | 66.9 | 79.4 | 45.1 | 20.5 | 20.4 | 54.2 | 48.6 | 27.4 | 21.1 | 69.9 | 85.5 | 36.2 | 66.2 | 53.3 | 56.7 | 88.9 | 67.2 | 54.3 | 96.6 | 59.7 |
| | 50x16 | 291.9M | 162.7 | 85.1 | 73.2 | 81.3 | 52.1 | 19.6 | 24.4 | 57.9 | 53.4 | 41.2 | 27.5 | 72.0 | 89.7 | 39.8 | 70.7 | 62.7 | 62.5 | 90.1 | 67.5 | 59.4 | 97.8 | 63.6 |
| | 50x64 | 623.3M | 552.7 | 83.9 | 75.9 | 85.1 | 59.9 | 22.7 | 29.8 | 62.7 | 53.1 | 48.0 | 31.0 | 76.1 | 91.1 | 47.9 | 73.9 | 78.9 | 53.9 | 93.6 | 71.0 | 63.9 | 98.3 | 66.0 |
| CLIP-ViT | B-32 | 151.3M | 14.9 | 83.7 | 59.7 | 89.8 | 64.2 | 23.2 | 17.2 | 53.0 | 44.0 | 37.4 | 19.7 | 66.5 | 82.6 | 32.6 | 63.3 | 38.1 | 62.3 | 87.5 | 58.6 | 54.2 | 97.1 | 61.3 |
| | B-16 | 149.6M | 41.1 | 86.6 | 64.6 | 90.8 | 66.9 | 21.2 | 22.8 | 55.3 | 45.1 | 44.1 | 24.4 | 71.4 | 87.9 | 43.3 | 68.3 | 59.5 | 50.7 | 89.1 | 60.5 | 58.3 | 98.2 | 63.7 |
| | L-14 | 427.6M | 175.3 | 85.9 | 77.8 | 95.6 | 75.8 | 19.4 | 31.9 | 62.1 | 55.3 | 47.2 | 31.7 | 79.1 | 92.3 | 50.6 | 75.5 | 73.3 | 52.0 | 93.6 | 68.9 | 64.6 | 99.4 | 66.7 |
| | L-14-336 | 429.9M | 395.2 | 86.1 | 79.4 | 95.0 | 74.4 | 20.0 | 34.5 | 63.0 | 56.4 | 45.0 | 33.2 | 78.3 | 93.1 | 52.4 | 76.6 | 75.2 | 60.7 | 93.8 | 70.6 | 63.7 | 99.4 | 67.7 |
| ORACLE | RN | | | 90.2 | 91.3 | 96.5 | 76.8 | 56.2 | 41.2 | 82.5 | 73.5 | 81.9 | 52.3 | 86.0 | 96.1 | 65.2 | 84.9 | 90.2 | 87.0 | 98.1 | 95.3 | 82.3 | 99.3 | 82.7 |
| | ViT | | | 91.9 | 89.9 | 98.5 | 87.1 | 34.4 | 44.0 | 78.9 | 67.2 | 72.0 | 51.9 | 85.7 | 96.6 | 67.0 | 84.6 | 85.6 | 95.4 | 97.5 | 92.3 | 78.7 | 99.7 | 80.6 |
| | All | | | 93.7 | 95.4 | 99.3 | 91.0 | 60.4 | 52.0 | 88.1 | 78.2 | 91.5 | 66.3 | 88.9 | 98.0 | 77.9 | 89.0 | 95.1 | 99.1 | 99.1 | 98.6 | 88.7 | 99.8 | 87.0 |

Table B.2: Impact of artificial alterations of the input image on each backbone in terms of delta accuracy concerning the original performance

|  | RN | | | | | ViT | | | |
|---|---|---|---|---|---|---|---|---|---|
|  | 50 | 50x4 | 50x16 | 50x64 | 101 | B-32 | B-16 | L-14 | L-14-336 |
| Resize | -3.81 | -3.47 | -1.90 | **-1.25** | -4.05 | -4.41 | -2.43 | -2.82 | -2.41 |
| Flip | **0.12** | -0.36 | -0.36 | -0.55 | -0.67 | -0.22 | -0.12 | -0.80 | -1.01 |
| Grayscale | -6.36 | -7.37 | **-5.98** | -7.01 | -7.52 | -8.94 | -8.70 | -7.73 | -6.99 |
| Color Jitter | -2.71 | -3.00 | -2.19 | -2.93 | -2.64 | -3.52 | -2.53 | **-1.67** | -1.73 |
| Gaussian Blur | -5.12 | -3.72 | -1.96 | -2.06 | -4.49 | -3.17 | -3.83 | -3.21 | **-1.80** |

**Confidence (CONF):**   This method utilizes the Shannon entropy to assess the confidence of each backbone in making a prediction. The backbone with the highest confidence for a given prediction is selected as the source of the final prediction.

**Voting (VOTE T-1 and VOTE T-3)**   : Voting combines different classifiers to predict the class label. In the VOTE T-1 approach, we consider the top-1 prediction from each backbone, and the final prediction is determined by the label with the highest number of votes. In cases where multiple labels receive the same number of votes, we select the one with the highest probability. Additionally, we explore the VOTE T-3 method, which seeks consensus among the backbones by considering the three most likely predictions from each backbone. These predictions are weighted based on their position within the top-3 list.

**Genetic Algorithm** (GAC). To showcase the importance of adaptively adjusting the logits, we use a genetic algorithm to find the optimal temperatures for combining the backbones. These temperatures are fixed after training, ensuring they remain constant regardless of the input.

**Calibrated confidence (C-CONF)**   For this approach, we first calibrate the probabilities of each backbone independently using temperature scaling. Then follow the procedure above and utilize the Shannon entropy of each backbone to select the one with the highest confidence

Table C.1 presents the baselines' results in the combination of zero-shot CLIP backbones. In non-parametric baselines, consistently with the presented in the main paper, leveraging the confidence of each backbone in its predictions consistently fails to enhance the overall performance beyond that of the best backbone. The lack of calibrated probabilities in the CONF contributes to overconfidence in some backbones, resulting in performance degradation when combined. Calibrating the confidence (C-CONF) leads to improvements, although the performance still falls short of matching the best backbone, except for DTD, CALTECH101, EUROSAT and CARS. This trend persists in the LOG-AVG approach, where averaging performance across backbones does not effectively exploit their complementarity between prediction, yielding an average delta accuracy of 0.1%. Notably, the LOG-AVG approach shows substantial improvement for the CUB dataset compared to the best backbone. When the backbones are calibrated C-LOG-AVG, the LOG-AVG approach enhances its performances across all datasets except GTSRB, MNIST, RESISC45, and SUN397 when compared to the non-calibrated version. Intriguingly, conventional ensemble techniques such as **Vote T-1** and **Vote T-3** prove ineffective in providing a significant boost to prediction accuracy beyond that of the best backbone.

Table C.1: Our results on ensembling the zero-shot predictions of CLIP backbones, which we group intro non-parametric and parametric techniques, and also compare to the best-performing single backbone (SINGLE-BEST). Mean, Max and Min Δ summarize the difference in performance when we compare it against the SINGLE-BEST backbone across datasets.

| | CALTECH101 | CARS | CIFAR10 | CIFAR100 | CLEVR | COUNTRY211 | CUB | DTD | EUROSAT | FGVC | FLOWERS | FOOD | GTSRB | IMGNET-1K | MNIST | PCAM | PETS | SST2 | RESISC45 | STL10 | SUN397 | Mean Δ | Min Δ | Max Δ |
|---|---|---|---|---|---|---|---|---|---|---|---|---|---|---|---|---|---|---|---|---|---|---|---|---|
| SINGLE-BEST | 86.6 | 79.4 | 95.6 | 75.8 | 24.4 | 34.5 | 63.0 | 56.4 | 48.0 | 33.2 | 79.1 | 93.1 | 52.4 | 76.6 | 78.9 | 63.9 | 93.8 | 71.0 | 64.6 | 99.4 | 67.7 | | | |
| CONF | 83.2 | 67.6 | 82.5 | 54.4 | 21.6 | 24.4 | 57.1 | 50.2 | 35.7 | 23.4 | 70.7 | 85.4 | 40.8 | 65.2 | 66.5 | 58.8 | 88.5 | 59.2 | 54.2 | 98.3 | 63.1 | -8.9 | -21.4 | -1.1 |
| VOTE T-1 | 86.3 | 79.7 | 94.3 | 74.6 | 22.7 | 30.7 | 68.1 | 58.0 | 48.2 | 34.1 | 78.4 | 92.5 | 53.9 | 76.1 | 70.0 | 62.7 | 93.9 | 71.7 | 67.3 | 98.8 | 70.0 | -0.3 | -9.0 | 5.1 |
| VOTE T-3 | 86.4 | 81.2 | 94.9 | 77.0 | 22.5 | 32.7 | 69.3 | 59.4 | 50.7 | 35.0 | 78.9 | 93.0 | 55.9 | 76.8 | 74.0 | - | 94.1 | - | 68.1 | 98.8 | 70.4 | 0.9 | -4.9 | 6.3 |
| LOG-AVG | 86.9 | 80.9 | 94.9 | 75.0 | 23.6 | 31.3 | 69.8 | 57.7 | 49.4 | 35.3 | 77.9 | 92.9 | 54.8 | 76.6 | 74.8 | 62.3 | 94.2 | 72.6 | 68.3 | 98.8 | 70.9 | 0.5 | -4.1 | 6.7 |
| C-LOG-AVG | 86.6 | 81.2 | 94.9 | 77.3 | 23.8 | 33.1 | 69.9 | 58.8 | 51.6 | 36.3 | 78.9 | 92.9 | 48.1 | 76.8 | 60.9 | 63.3 | 94.3 | 73.3 | 68.1 | 98.9 | 70.4 | 0.1 | -18.0 | 6.9 |
| C-CONF | 88.9 | 79.7 | 88.9 | 74.1 | 14.5 | 33.2 | 65.7 | 58.8 | 53.4 | 32.7 | 78.9 | 92.5 | 44.3 | 75.9 | 57.2 | 51.2 | 93.3 | 56.7 | 64.5 | 98.0 | 67.7 | -3.2 | -21.7 | 5.4 |
| GAC | 87.4 | 81.9 | 95.4 | 78.2 | 24.3 | 34.8 | 70.0 | 58.5 | 51.9 | 36.7 | 79.0 | 93.8 | 54.2 | 78.0 | 84.7 | 61.3 | 94.8 | 72.6 | 68.8 | 98.8 | 71.6 | 1.9 | -2.6 | 6.9 |
| SL | 87.6 | 82.3 | 96.1 | 78.2 | 29.0 | 36.0 | 70.8 | 59.6 | 53.5 | 36.8 | 80.4 | 94.2 | 54.7 | 78.3 | 84.1 | 71.0 | 95.2 | 73.3 | 73.4 | **99.5** | 71.9 | 3.3 | 0.1 | 8.8 |
| Ours | **91.1** | **84.1** | **96.7** | **79.6** | **54.4** | **36.1** | **73.6** | **63.2** | **87.1** | **38.2** | **82.1** | **94.3** | **70.5** | **78.8** | **93.9** | **84.9** | **95.4** | **77.3** | **78.3** | **99.5** | **74.2** | 9.3 | 0.1 | 39.1 |

Table D.1: Linear classifier accuracy across multiple datasets and employing different backbones, showcasing the performance improvement achieved by the linear classifier compared to the zero-shot version. Also, the combination of ORACLE of linear classifiers. The final three columns present the delta statistics with respect to the zero-shot version of each backbone. In the case of ORACLE, the delta with respect to the best linear probing.

| | | Caltech101 | Cars | CIFAR10 | CIFAR100 | CLEVR | Country211 | CUB | DTD | EuroSAT | FGVC | Flowers | Food | GTSRB | ImgNet-1K | MNIST | PCAM | Pets | SST2 | Resisc45 | STL10 | SUN397 | Mean Δ | Min Δ | Max Δ |
|---|---|---|---|---|---|---|---|---|---|---|---|---|---|---|---|---|---|---|---|---|---|---|---|---|---|
| ResNet | 50 | 93.8 | 77.5 | 88.5 | 68.8 | 61.3 | 17.9 | 65.6 | 69.0 | 95.0 | 39.9 | 90.9 | 81.7 | 84.5 | 70.3 | 97.4 | 78.3 | 85.5 | 71.3 | 89.0 | 97.1 | 74.9 | 23.5 | -0.3 | 66.8 |
| | 50x4 | 96.3 | 85.2 | 90.3 | 71.7 | 53.8 | 22.0 | 76.0 | 73.9 | 94.9 | 49.6 | 95.9 | 89.7 | 86.1 | 76.2 | 97.7 | 82.2 | 91.6 | 73.6 | 91.0 | 98.1 | 78.4 | 22.6 | 1.5 | 67.5 |
| | 50x16 | 97.1 | 88.0 | 91.5 | 73.9 | 61.0 | 26.4 | 81.5 | 76.7 | 95.5 | 55.7 | 96.2 | 91.8 | 88.6 | 80.4 | 98.3 | 74.8 | 93.2 | 78.1 | 92.7 | 98.9 | 80.6 | 20.5 | 1.2 | 54.3 |
| | 50x64 | 97.2 | 90.4 | 93.8 | 77.0 | 31.5 | 33.2 | 84.9 | 79.1 | 95.6 | 61.2 | 97.9 | 93.1 | 91.1 | 83.1 | 98.5 | 76.4 | 94.1 | 80.3 | 93.7 | 99.3 | 82.1 | 17.5 | 0.5 | 47.7 |
| | 101 | 95.5 | 80.9 | 90.9 | 72.6 | 61.8 | 20.1 | 70.7 | 69.7 | 94.5 | 41.9 | 92.7 | 84.5 | 83.5 | 72.4 | 97.4 | 75.5 | 89.1 | 66.1 | 90.7 | 98.1 | 77.1 | 22.1 | 1.3 | 68.1 |
| ViT | B-32 | 95.8 | 76.8 | 94.8 | 77.6 | 59.1 | 18.8 | 70.9 | 71.8 | 95.1 | 41.3 | 92.6 | 83.2 | 85.6 | 73.0 | 98.0 | 77.6 | 87.7 | 69.9 | 90.1 | 98.5 | 74.6 | 20.8 | 0.2 | 59.9 |
| | B-16 | 96.9 | 83.2 | 95.7 | 79.4 | 61.7 | 23.4 | 76.5 | 74.9 | 95.5 | 49.2 | 94.6 | 88.1 | 87.6 | 76.6 | 98.2 | 82.4 | 91.7 | 71.4 | 91.9 | 99.2 | 77.8 | 20.1 | 0.2 | 51.4 |
| | L-14 | 97.6 | 89.1 | 98.0 | 85.5 | 68.9 | 32.2 | 83.3 | 78.3 | 97.1 | 60.9 | 98.4 | 92.4 | 92.6 | 82.2 | 98.8 | 75.5 | 94.6 | 79.4 | 94.6 | 99.8 | 81.4 | 18.2 | 0.1 | 49.9 |
| | L-14-336 | 97.7 | 90.6 | 97.7 | 85.4 | 67.1 | 35.9 | 86.3 | 78.8 | 96.9 | 63.9 | 98.5 | 94.7 | 92.4 | 84.0 | 98.8 | 78.7 | 94.6 | 80.5 | 95.1 | 99.7 | 82.5 | 18.2 | 0.3 | 51.9 |
| ORACLE | RN | 99.1 | 96.1 | 98.2 | 91.4 | 91.9 | 48.2 | 93.0 | 89.5 | 99.0 | 76.1 | 99.1 | 97.6 | 96.3 | 88.9 | 99.8 | 92.6 | 97.6 | 94.9 | 98.5 | 99.8 | 92.4 | 8.1 | 0.5 | 28.7 |
| | ViT | 99.0 | 95.7 | 99.5 | 94.1 | 91.4 | 49.4 | 92.8 | 88.6 | 99.2 | 78.1 | 99.3 | 97.6 | 96.9 | 89.1 | 99.7 | 92.7 | 97.7 | 92.6 | 98.5 | 99.9 | 91.5 | 6.5 | 0.1 | 22.5 |
| | All | 99.4 | 97.6 | 99.7 | 96.7 | 97.2 | 59.5 | 95.6 | 92.7 | 99.6 | 84.6 | 99.6 | 98.8 | 98.2 | 91.8 | 99.9 | 96.0 | 98.7 | 96.9 | 99.5 | 100.0 | 95.0 | 9.1 | 0.2 | 28.3 |

Table D.2: Our results on combining the LinearProbe CLIP predictions with different backbones, which we group into non-parametric and parametric techniques. Mean, Max and Min Δ summarize the difference in performance across datasets.

| | Caltech101 | Cars | CIFAR10 | CIFAR100 | CLEVR | Country211 | CUB | DTD | EuroSAT | FGVC | Flowers | Food | GTSRB | ImgNet-1K | MNIST | PCAM | Pets | SST2 | Resisc45 | STL10 | SUN397 | Mean Δ | Min Δ | Max Δ |
|---|---|---|---|---|---|---|---|---|---|---|---|---|---|---|---|---|---|---|---|---|---|---|---|---|
| Single-Best | 97.7 | 90.6 | 98.0 | 85.5 | 68.9 | 35.9 | 86.3 | 79.1 | 97.1 | 63.9 | 98.5 | 94.7 | 92.6 | 84.0 | 98.8 | 82.4 | 94.6 | 80.5 | 95.1 | 99.8 | 82.5 | - | - | - |
| Vote T-1 | 98.2 | 91.3 | 97.2 | 87.1 | 71.0 | 37.4 | 86.7 | 81.3 | 97.6 | 63.8 | 98.1 | 95.0 | 93.3 | 82.6 | 99.2 | 80.4 | 94.8 | 80.5 | 95.8 | 99.6 | 85.0 | 0.5 | -2.0 | 2.5 |
| Vote T-3 | 98.2 | 91.4 | 97.2 | 87.0 | 72.0 | 37.4 | 86.5 | 80.9 | 97.7 | 64.0 | 98.1 | 95.0 | 93.4 | 83.4 | 99.2 | 80.4 | 94.7 | 80.5 | 95.9 | 99.6 | 85.0 | 0.5 | -2.0 | 3.2 |
| Conf | 96.5 | 86.0 | 91.5 | 73.7 | 57.3 | 24.1 | 80.7 | 73.5 | 96.5 | 52.9 | 94.8 | 87.9 | 87.9 | 80.0 | 97.8 | 79.6 | 91.4 | 73.6 | 91.2 | 98.7 | 79.4 | -5.3 | -11.8 | -0.5 |
| Log-Avg | 98.0 | 91.5 | 98.4 | 89.0 | 73.8 | 40.5 | 87.3 | 82.0 | 98.0 | 64.7 | 98.6 | 95.3 | 94.3 | 84.6 | 99.3 | 80.8 | 95.1 | 81.3 | 96.3 | 99.8 | 85.5 | 1.3 | -1.5 | 5.0 |
| C-Conf | 97.5 | 91.3 | 96.4 | 85.0 | 62.7 | 35.2 | 87.3 | 80.1 | 96.4 | 62.9 | 98.2 | 95.0 | 91.9 | 84.6 | 98.9 | 71.9 | 94.3 | 61.9 | 94.7 | 99.3 | 83.0 | -1.8 | -18.6 | 1.0 |
| C-Log-Avg | 98.3 | 91.9 | 97.2 | 87.2 | 73.2 | 38.3 | 87.5 | 81.3 | 97.7 | 65.2 | 98.2 | 95.2 | 93.4 | 83.3 | 99.2 | 80.6 | 94.9 | 80.5 | 96.0 | 99.6 | 85.0 | 0.8 | -1.7 | 4.4 |
| GAC | 98.1 | 92.3 | 98.2 | 88.6 | 73.3 | 40.7 | 88.3 | 82.0 | 51.9 | 65.3 | 98.6 | 95.2 | 94.8 | 84.8 | 99.2 | 82.6 | 95.1 | 81.1 | 96.3 | 99.8 | 71.6 | -1.4 | -45.2 | 4.8 |
| MoE | 96.5 | 90.5 | 97.6 | 87.1 | 89.6 | 41.2 | 84.8 | 72.7 | 96.9 | 43.0 | 96.1 | 94.9 | 93.4 | 93.4 | 98.8 | 82.3 | 93.4 | 79.1 | 94.5 | 98.8 | 85.0 | 0.2 | -20.9 | 20.8 |
| SL | 98.3 | 91.9 | 98.4 | 88.8 | 74.3 | 40.8 | 87.7 | 81.0 | 98.1 | 65.5 | 98.7 | 95.3 | 94.3 | 84.8 | 99.3 | 83.6 | 95.0 | 81.9 | 96.4 | 99.8 | 85.3 | 1.6 | 0.0 | 5.4 |
| NLC | 98.3 | 91.8 | 98.3 | 88.8 | 77.0 | 40.8 | 87.2 | 79.4 | 97.8 | 66.4 | 98.6 | 95.4 | 94.4 | 84.7 | 99.3 | 83.4 | 95.0 | 80.9 | 96.3 | 99.8 | 85.9 | 1.6 | 0.0 | 8.1 |

In employing parametric methods, we utilize the entire training set of the target datasets. Our proposed approach NLC demonstrates a noteworthy capability to enhance the performance of the best backbone, achieving a substantial improvement of up to 39.1% in the case of the EUROSAT dataset. On average, our method exhibits a commendable improvement of 9.3% over the SINGLE-BEST backbone's performance across the evaluated datasets.

# D    NLC USING LINEAR CLASSIFIERS OF CLIP BACKBONES

The results of adapting CLIP using a linear classifier are shown in Table D.1. RN50 demonstrates the most significant improvement, averaging an impressive 23.5% increase across all datasets, surpassing even ViT-B-32. However, RN50 still falls short of outperforming other backbones overall. ViT-L-14 and ViT-L-14-336 continue to be the top performers across all datasets, even after linear probing.

A notable observation is the superior performance of the ORACLE of linear probes compared to any individual backbone, highlighting the potential benefits of combining multiple backbones. Although the performance gap between the ORACLE and the best linear probe is consistently observed across datasets, it is less pronounced than in the zero-shot scenario, indicating a smaller margin for improvement.

Table E.1: Detailed ImageNet robustness performance. IN is used to abbreviate for ImageNet

| | IN | IN-V2 | IN-A | IN-R | IN-Sketch |
|---|---|---|---|---|---|
| Best Backbone | 76.6 | 70.3 | **77.6** | 89.0 | 60.9 |
| NNC (Train on IN) | **78.8** | **72.4** | 75.6 | **89.3** | **61.8** |

Table F.1: Classification accuracy of models under few-shot settings.

| Few-shot | 1 | 2 | 4 | 8 | 16 |
|---|---|---|---|---|---|
| Linear-probe CLIP Radford et al. (2021) | 22.2 | 31.9 | 41.2 | 49.5 | 56.1 |
| CoOPZhou et al. (2022) | 47.6 | 50.9 | 56.2 | 59.9 | 63.0 |
| CLIP-AdapterGao et al. (2021) | 61.2 | 61.5 | 61.8 | 62.7 | 63.6 |
| Tip-AdapterZhang et al. (2021) | 60.7 | 61.0 | 61.0 | 61.5 | 62.0 |
| Tip-Adapter-FZhang et al. (2021) | 61.3 | 61.7 | 62.5 | 64.0 | 65.5 |
| NLC | **78.2** | **78.1** | **78.2** | **78.3** | **78.4** |

Table F.2: Classification accuracy of models on various vision backbones using 16-shots.

| Models | ResNet | | | ViT | |
|---|---|---|---|---|---|
| | 50 | 101 | 50x16 | B-32 | B-16 |
| Zero-shot CLIP Radford et al. (2021) | 60.3 | 62.5 | 70.9 | 63.8 | 68.7 |
| CoOPZhou et al. (2022) | 47.6 | 50.9 | 56.2 | 59.9 | 63.0 |
| CoOP | 63.0 | 66.6 | - | 66.9 | 71.9 |
| CLIP-AdapterGao et al. (2021) | 63.6 | 65.4 | - | 66.2 | 71.1 |
| Tip-AdapterZhang et al. (2021) | 62.0 | 64.8 | 73.0 | 65.6 | 70.8 |
| Tip-Adapter-FZhang et al. (2021) | 65.5 | 68.6 | 75.8 | 68.7 | 73.7 |
| NLC | | | **78.4** | | |

# E ROBUSTNESS UNDER DISTRIBUTION SHIFT OF IMGNET-1K

To assess how well the combination of backbones performs under varying conditions, we test its robustness using natural distribution shifts in the ImageNet dataset. We evaluate its performance on four datasets representing different distribution shifts: ImageNet-V2 Recht et al. (2019), ImageNet Adversarial Hendrycks et al. (2021b), ImageNet Rendition Hendrycks et al. (2021a) and ImageNet Sketch Wang et al. (2009). Specifically, we employ a NLC trained on the ZeroShot CLIP backbones using the original IMGNET-1K data Deng et al. (2009). This allows us to determine whether the learned combination of backbones can maintain its performance across these distribution shift datasets. In Table E.1, we observe that the learned combination of backbones, denoted as NLC, enhances performance in 3 out of 4 selected benchmarks, with improvements ranging from 0.3% to 2.1%. However, in the case of ImageNet Adversarial, the performance of the combination of backbones appears to suffer, possibly due to a more complex decision boundary.

# F THE COMBINATION OF NLC WITH TIP-ADAPTER

We integrate the Tip-Adapter Zhang et al. (2021) with our NLC ensembling mechanism and conduct experiments on EUROSAT, FGVC, DTD, and PETS datasets. Initially, we apply Tip-Adapter and Tip-Adapter-F independently on each backbone, following their protocol with 1, 2, 4, 8, and 16 shots. Subsequently, we employ our ensembling mechanism to fuse the adapted backbones. For training NLC, we utilize the validation set used by Tip-Adapter to combine their logits with CLIP linearly.

In Table F.1 and F.2, we compare the performance of different few-shot adapters of CLIP on the ImageNet dataset. It shows that current few-shot adapter methods also have performance differences with various backbones; still, our NLC surpassed the best backbone reported by their method.

Table F.3, F.4, F.5, and F.6 show the zero-shot performance of the CLIP used by Tip-Adapter. It also shows the Tip-Adapter and Tip-Adapter-F performance on each backbone and when we combine all the Tip-Adapter versions and backbones with NLC. Across the four datasets used for this experiment, NLC improve the performance of each version of Tip-Adapter. Notably, NLC for EUROSAT obtained an improvement of up to 15% with respect to the best Tip-Adapter backbone (L-14-336) using 1 shot.

Table F.3: Tip-Adapter and Tip-Adapter-F fused with our NLC and applied to EUROSAT dataset

Table F.4: Tip-Adapter and Tip-Adapter-F fused with our NLC and applied to FGVC dataset

| ZeroShot | | | | | | | | |
|---|---|---|---|---|---|---|---|---|
| ResNet | | | | | ViT | | | |
| 50 | 50x4 | 50x16 | 50x64 | 101 | B-32 | B-16 | L-14 | L-14-336 |
| 37.5 | 32.0 | 40.3 | 49.4 | 32.5 | 45.2 | 47.6 | 58.1 | 63.5 |

| ZeroShot | | | | | | | | |
|---|---|---|---|---|---|---|---|---|
| ResNet | | | | | ViT | | | |
| 50 | 50x4 | 50x16 | 50x64 | 101 | B-32 | B-16 | L-14 | L-14-336 |
| 17.2 | 21.4 | 27.0 | 30.2 | 18.1 | 19.3 | 24.8 | 32.6 | 33.4 |

**TiP-Adapter**

| | Model | 1 | 2 | Shots 4 | 8 | 16 |
|---|---|---|---|---|---|---|
| ResNet | 50 | 55.5 | 60.8 | 68.1 | 66.3 | 70.3 |
| | 50x4 | 53.0 | 56.8 | 67.0 | 68.0 | 71.7 |
| | 50x16 | 54.5 | 61.0 | 62.7 | 65.1 | 71.4 |
| | 50x64 | 65.1 | 66.5 | 70.5 | 73.5 | 73.9 |
| | 101 | 47.9 | 49.7 | 60.6 | 64.7 | 67.6 |
| ViT | B-32 | 54.1 | 61.7 | 68.2 | 70.6 | 69.8 |
| | B-16 | 66.9 | 68.8 | 75.1 | 72.9 | 78.1 |
| | L-14 | 73.0 | 71.1 | 78.4 | 79.2 | 82.4 |
| | L-14-336 | 73.5 | 72.9 | 75.7 | 79.2 | 80.4 |
| | With SL | 77.8 | 81.5 | 85.6 | 86.4 | 85.4 |
| | With NLC | **88.4** | **86.8** | **89.7** | **90.1** | **90.7** |

**TiP-Adapter**

| | Model | 1 | 2 | Shots 4 | 8 | 16 |
|---|---|---|---|---|---|---|
| ResNet | 50 | 19.0 | 21.4 | 22.5 | 25.9 | 29.6 |
| | 50x4 | 24.1 | 27.8 | 29.3 | 31.5 | 37.8 |
| | 50x16 | 29.0 | 32.4 | 35.7 | 38.5 | 42.6 |
| | 50x64 | 35.0 | 38.5 | 41.0 | 44.8 | 49.9 |
| | 101 | 20.4 | 22.9 | 24.8 | 28.3 | 32.1 |
| ViT | B-32 | 21.6 | 23.9 | 23.7 | 27.2 | 31.6 |
| | B-16 | 28.0 | 31.1 | 32.3 | 36.7 | 39.6 |
| | L-14 | 35.6 | 41.7 | 44.9 | 48.9 | 52.3 |
| | L-14-336 | 37.7 | 43.6 | 46.5 | 50.0 | 53.8 |
| | With SL | 39.1 | 44.2 | 47.7 | 51.2 | 54.8 |
| | With NLC | **40.9** | **44.5** | **47.9** | **51.5** | **55.0** |

**TiP-Adapter-F**

| | Model | 1 | 2 | Shots 4 | 8 | 16 |
|---|---|---|---|---|---|---|
| ResNet | 50 | 60.7 | 64.4 | 73.3 | 77.7 | 84.9 |
| | 50x4 | 59.5 | 61.9 | 76.2 | 81.9 | 84.9 |
| | 50x16 | 61.4 | 68.7 | 75.9 | 80.8 | 83.7 |
| | 50x64 | 71.2 | 69.7 | 78.7 | 81.4 | 86.6 |
| | 101 | 62.7 | 57.7 | 75.4 | 78.8 | 83.6 |
| ViT | B-32 | 59.4 | 70.1 | 76.5 | 79.9 | 84.9 |
| | B-16 | 66.6 | 71.0 | 79.1 | 83.8 | 88.9 |
| | L-14 | 74.9 | 75.0 | 86.1 | 86.4 | 90.6 |
| | L-14-336 | 72.5 | 76.4 | 86.1 | 85.3 | 91.0 |
| | With SL | 84.5 | 86.7 | 89.9 | 89.0 | 93.1 |
| | With NLC | **89.7** | **88.3** | **91.4** | **91.1** | **93.8** |

**TiP-Adapter-F**

| | Model | 1 | 2 | Shots 4 | 8 | 16 |
|---|---|---|---|---|---|---|
| ResNet | 50 | 20.5 | 22.9 | 26.5 | 30.4 | 35.5 |
| | 50x4 | 26.3 | 29.1 | 32.8 | 36.8 | 42.2 |
| | 50x16 | 31.2 | 36.9 | 38.3 | 43.6 | 49.4 |
| | 50x64 | 37.0 | 40.8 | 45.0 | 49.8 | 54.8 |
| | 101 | 21.7 | 24.0 | 26.9 | 32.0 | 38.0 |
| ViT | B-32 | 22.7 | 25.3 | 27.5 | 32.9 | 36.9 |
| | B-16 | 30.2 | 34.1 | 36.1 | 40.9 | 44.6 |
| | L-14 | 38.6 | 44.1 | 48.5 | 51.9 | 57.4 |
| | L-14-336 | 40.9 | 45.2 | 49.6 | 52.7 | 59.0 |
| | With SL | 46.7 | 49.1 | 53.4 | 57.4 | 62.4 |
| | With NLC | **46.8** | **49.2** | **53.6** | **57.5** | **62.6** |

Table F.5: Tip-Adapter and Tip-Adapter-F fused with our NLC and applied to DTD dataset

Table F.6: Tip-Adapter and Tip-Adapter-F fused with our NLC and applied to PETS dataset

| | ZeroShot | | | | | | | | |
|---|---|---|---|---|---|---|---|---|---|
| | ResNet | | | | | ViT | | | |
| 50 | 50x4 | 50x16 | 50x64 | 101 | B-32 | B-16 | L-14 | L-14-336 | |
| 42.1 | 50.8 | 50.3 | 48.4 | 38.4 | 43.7 | 44.6 | 53.1 | 54.0 | |

| | ZeroShot | | | | | | | | |
|---|---|---|---|---|---|---|---|---|---|
| | ResNet | | | | | ViT | | | |
| 50 | 50x4 | 50x16 | 50x64 | 101 | B-32 | B-16 | L-14 | L-14-336 | |
| 85.8 | 88.9 | 90.1 | 93.7 | 86.9 | 87.4 | 89.1 | 93.6 | 93.8 | |

**TiP-Adapter (DTD)**

| | Model | \multicolumn Shots | | | | |
|---|---|---|---|---|---|---|
| | | 1 | 2 | 4 | 8 | 16 |
| ResNet | 50 | 46.3 | 49.9 | 54.3 | 58.5 | 61.3 |
| | 50x4 | 52.5 | 56.3 | 59.8 | 62.4 | 65.8 |
| | 50x16 | 56.9 | 59.5 | 61.5 | 65.3 | 67.6 |
| | 50x64 | 55.6 | 61.5 | 63.2 | 67.3 | 70.0 |
| | 101 | 48.2 | 51.5 | 56.6 | 60.2 | 63.6 |
| ViT | B-32 | 49.3 | 51.9 | 55.3 | 59.4 | 63.2 |
| | B-16 | 51.5 | 52.6 | 58.6 | 62.8 | 67.0 |
| | L-14 | 59.6 | 61.6 | 64.8 | 69.7 | 71.3 |
| | L-14-336 | 59.3 | 61.8 | 64.5 | 69.5 | 72.6 |
| | With SL | 58.7 | 64.1 | 65.5 | 70.3 | 71.5 |
| | With NLC | **63.7** | **66.1** | **68.4** | **71.6** | **73.0** |

**TiP-Adapter (PETS)**

| | Model | Shots | | | | |
|---|---|---|---|---|---|---|
| | | 1 | 2 | 4 | 8 | 16 |
| ResNet | 50 | 86.2 | 87.2 | 86.0 | 86.7 | 88.8 |
| | 50x4 | 88.7 | 89.2 | 89.1 | 88.9 | 90.8 |
| | 50x16 | 90.5 | 90.8 | 91.4 | 91.7 | 90.9 |
| | 50x64 | 93.7 | 93.3 | 93.8 | 93.8 | 94.1 |
| | 101 | 86.8 | 86.6 | 86.9 | 87.3 | 88.1 |
| ViT | B-32 | 87.1 | 88.0 | 87.6 | 87.9 | 88.1 |
| | B-16 | 89.3 | 90.6 | 90.7 | 90.7 | 91.7 |
| | L-14 | 93.8 | 93.5 | 94.1 | 94.2 | 94.5 |
| | L-14-336 | 93.8 | 93.8 | 94.0 | 93.7 | 94.0 |
| | With SL | 93.8 | 93.6 | 94.3 | 94.4 | 94.5 |
| | With NLC | **94.5** | **94.2** | **94.9** | **95.0** | **95.3** |

**TiP-Adapter-F (DTD)**

| | Model | Shots | | | | |
|---|---|---|---|---|---|---|
| | | 1 | 2 | 4 | 8 | 16 |
| ResNet | 50 | 48.8 | 53.5 | 56.6 | 61.9 | 66.5 |
| | 50x4 | 53.7 | 57.2 | 61.6 | 65.7 | 70.6 |
| | 50x16 | 58.8 | 59.2 | 63.1 | 67.4 | 73.5 |
| | 50x64 | 59.0 | 61.8 | 66.6 | 69.9 | 76.0 |
| | 101 | 47.5 | 50.7 | 58.3 | 64.8 | 68.8 |
| ViT | B-32 | 50.9 | 55.3 | 60.4 | 63.8 | 68.9 |
| | B-16 | 52.7 | 56.0 | 61.0 | 67.6 | 71.0 |
| | L-14 | 60.4 | 60.7 | 67.1 | 71.6 | 75.8 |
| | L-14-336 | 58.9 | 60.9 | 67.1 | 72.4 | 75.7 |
| | With SL | 62.2 | 67.3 | 71.4 | 74.8 | 78.8 |
| | With NLC | **67.2** | **68.7** | **73.3** | **75.4** | **79.1** |

**TiP-Adapter-F (PETS)**

| | Model | Shots | | | | |
|---|---|---|---|---|---|---|
| | | 1 | 2 | 4 | 8 | 16 |
| ResNet | 50 | 86.6 | 86.8 | 87.2 | 87.5 | 89.5 |
| | 50x4 | 89.7 | 89.5 | 90.7 | 90.7 | 91.8 |
| | 50x16 | 92.8 | 92.0 | 92.9 | 92.9 | 92.9 |
| | 50x64 | 94.2 | 94.1 | 94.2 | 94.1 | 94.3 |
| | 101 | 87.6 | 88.4 | 88.7 | 88.7 | 90.5 |
| ViT | B-32 | 88.2 | 89.0 | 88.9 | 89.6 | 90.7 |
| | B-16 | 90.9 | 91.3 | 92.0 | 92.3 | 92.8 |
| | L-14 | 94.2 | 94.1 | 94.6 | 94.3 | 95.1 |
| | L-14-336 | 94.3 | 94.3 | 94.8 | 94.1 | 95.0 |
| | With SL | 94.5 | 94.6 | 95.2 | 95.2 | 95.1 |
| | With NLC | **95.2** | **95.0** | **95.3** | **95.2** | **95.5** |

Table G.1: Ablation of NLC performance when changing the number of samples used to train. We use NLC($n$) to denote NLC with $n$ samples per class. and showcase the improvement in performance compared with SINGLE-BEST backbone in a zero-shot setting.

| | Caltech101 | Cars | CIFAR10 | CIFAR100 | CLEVR | Country211 | CUB | DTD | EuroSAT | FGVC | Flowers | Food | GTSRB | ImgNet-1k | MNIST | PCAM | Pets | SST2 | Resisc45 | STL10 | SUN397 | Mean Δ | Min Δ | Max Δ |
|---|---|---|---|---|---|---|---|---|---|---|---|---|---|---|---|---|---|---|---|---|---|---|---|---|
| SINGLE-BEST | 86.6 | 79.4 | 95.6 | 75.8 | 24.4 | 34.5 | 63.0 | 56.4 | 48.0 | 33.2 | 79.1 | 93.1 | 52.4 | 76.6 | 78.9 | 63.9 | 93.8 | 71.0 | 64.6 | 99.4 | 67.7 | | | |
| NNC(1) | 87.3 | 82.3 | 93.6 | 78.0 | 25.1 | 36.0 | 71.0 | 59.3 | 55.0 | 37.1 | 79.4 | 93.6 | 54.8 | 77.6 | 83.3 | 62.5 | 94.3 | 73.8 | 70.0 | 98.9 | 71.1 | 2.2 | -2.0 | 8.0 |
| NNC(2) | 87.3 | 81.4 | 93.0 | 78.2 | 25.1 | 33.2 | 70.0 | 59.9 | 52.3 | 37.2 | 79.5 | 93.5 | 55.3 | 77.4 | 84.4 | 62.5 | 94.5 | 73.1 | 70.2 | 98.9 | 71.1 | 1.9 | -2.6 | 7.0 |
| NNC(4) | 87.1 | 81.3 | 95.3 | 75.9 | 26.9 | 35.4 | 70.2 | 59.9 | 54.0 | 35.9 | 78.2 | 93.1 | 55.2 | 78.1 | 81.6 | 62.7 | 94.5 | 72.7 | 68.8 | 99.1 | 71.2 | 1.9 | -1.3 | 7.2 |
| NNC(8) | 87.1 | 81.6 | 95.6 | 76.2 | 26.9 | 36.1 | 70.3 | 58.1 | 54.0 | 37.0 | 80.3 | 93.1 | 54.9 | 78.3 | 82.5 | 67.0 | 94.2 | 72.4 | 68.9 | 99.5 | 71.2 | 2.3 | 0.0 | 7.3 |
| NNC(16) | 87.1 | 82.8 | 95.8 | 77.2 | 27.0 | 36.2 | 71.3 | 60.4 | 53.5 | 37.0 | 80.4 | 93.5 | 55.2 | 78.4 | 83.3 | 69.6 | 94.2 | 73.6 | 69.6 | 99.4 | 72.2 | 2.9 | 0.0 | 8.3 |
| NNC(32) | 87.3 | 82.9 | 96.0 | 78.2 | 27.2 | 36.1 | 71.2 | 60.4 | 51.6 | 37.2 | 80.4 | 94.2 | 55.1 | 78.1 | 85.3 | 75.5 | 94.4 | 74.1 | 70.1 | 99.4 | 72.4 | 3.3 | 0.0 | 11.6 |

## G   TRAINING NLC UNDER LIMITED SAMPLES

In our exploration of the effectiveness of the NLC approach, we extend our analysis to a scenario where we limit the samples to combine the zero-shot CLIPs. This experiment allows us to assess the adaptability and performance of our proposed method under limited training data conditions. Table G.1 presents the performance of NLC by means of a limited number of samples. Although the performance of NLC overall improves when it has more data available to combine the backbones, in most cases, just using one sample NLC(1) per class is enough to improve its performance. Notably, there is a stop in performance degradation when we use more than 8 samples NLC(8) in all the benchmarks.

Moreover, we run five different seeds to train NLC on the different few-shots settings. Results show that the standard deviation of our obtained performance is stable and low across all of our studied datasets, suggesting that our approach is not sensitive to the effects of different samples. It varies with the number of samples used with a mean standard deviation of 0.62 when we used 1 shot and 0.48 when we used 32 shots.

## H   LEARNED TEMPERATURE VALUES

Figure H.1 and H.2 present the distribution of the temperature values using a box plot for the NLC method, normalized by their maximum value. Notably, there is a dominance in the temperature values for the best backbones in each family, *i.e.*, RN50x64 and ViT-L-14 or ViT-L-14-336. Particularly prominent in STL10, PETS, DTD, FGVCand FLOWERS, making these backbones especially relevant for NLC. Interestingly, it is observed that the most weighted backbone within the ResNet family is not consistently ResNet-101, despite its deeper architecture. This observation is evident in datasets such as PETS, CUB, FGVC and SUN397, where the mean value of the temperature corresponding to ResNet-50 surpasses that of ResNet-101.

Furthermore, the distribution of the temperature values across backbones for datasets such as CLEVR, EUROSAT, and GTSRB is more uniform compared to other datasets. This suggests that the NLC method is effectively leveraging the strengths of each backbone to arrive at accurate labels for each sample, resulting in a more balanced distribution of weights across different backbones, complementing our analysis on Fig. 2.

## I   DIVERSITY AND ORACLE PERFORMANCE OF DIFFERENT COMBINATIONS

Table I.1 shows the Oracle performance by combinaning different options of diversity. Table I.2 present the diversity of different source of complementarity.

## J   CASCADING AND ENSEMBLE

Figure J.3 shows the performance of multiple ensemble and cascading methods of 2 to 9 backbones, notice that NLC obtains the best performance, and when we add cascade it maintains the computational requirements

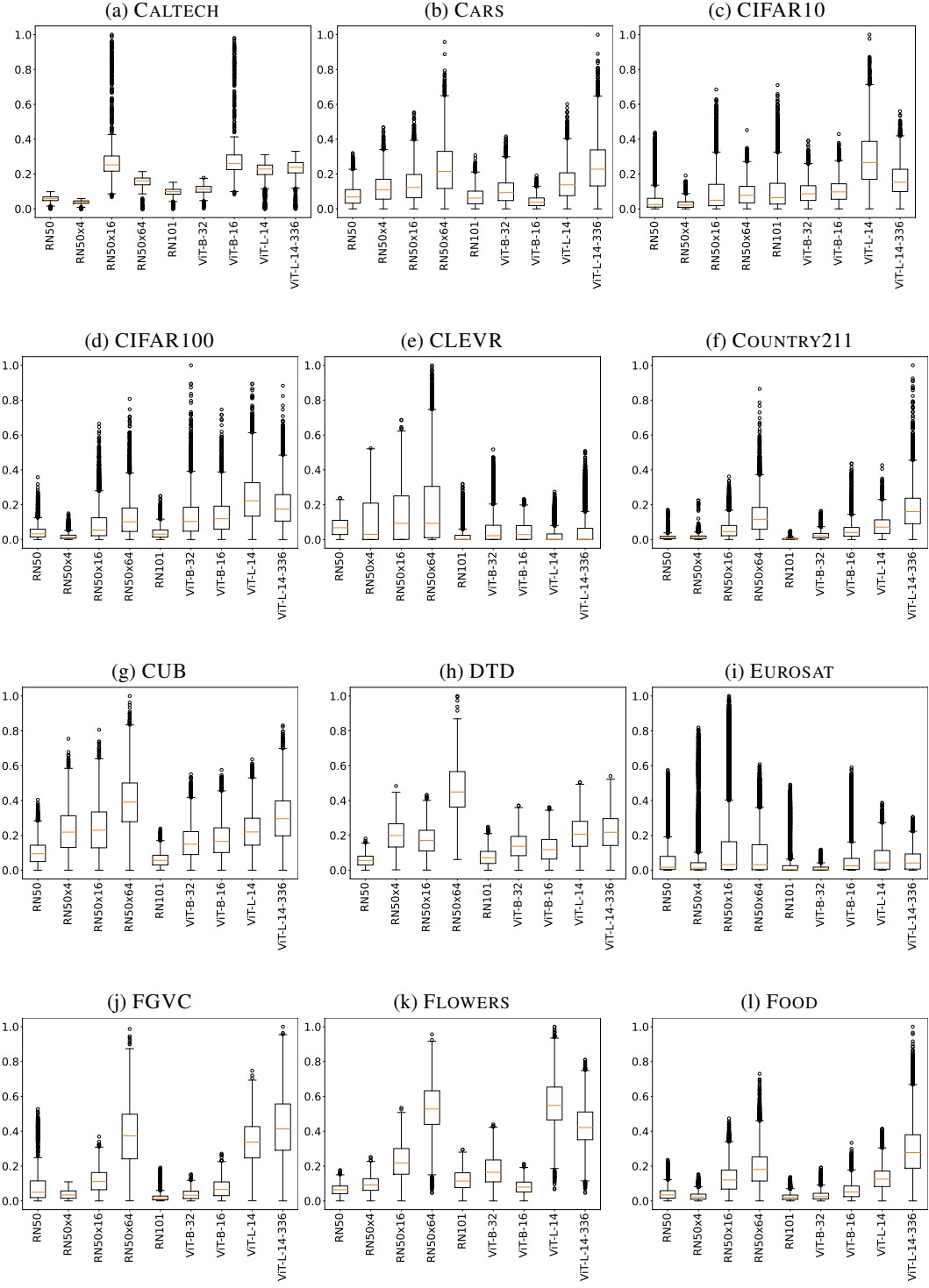

Figure H.1: Alpha values for each dataset using NLC

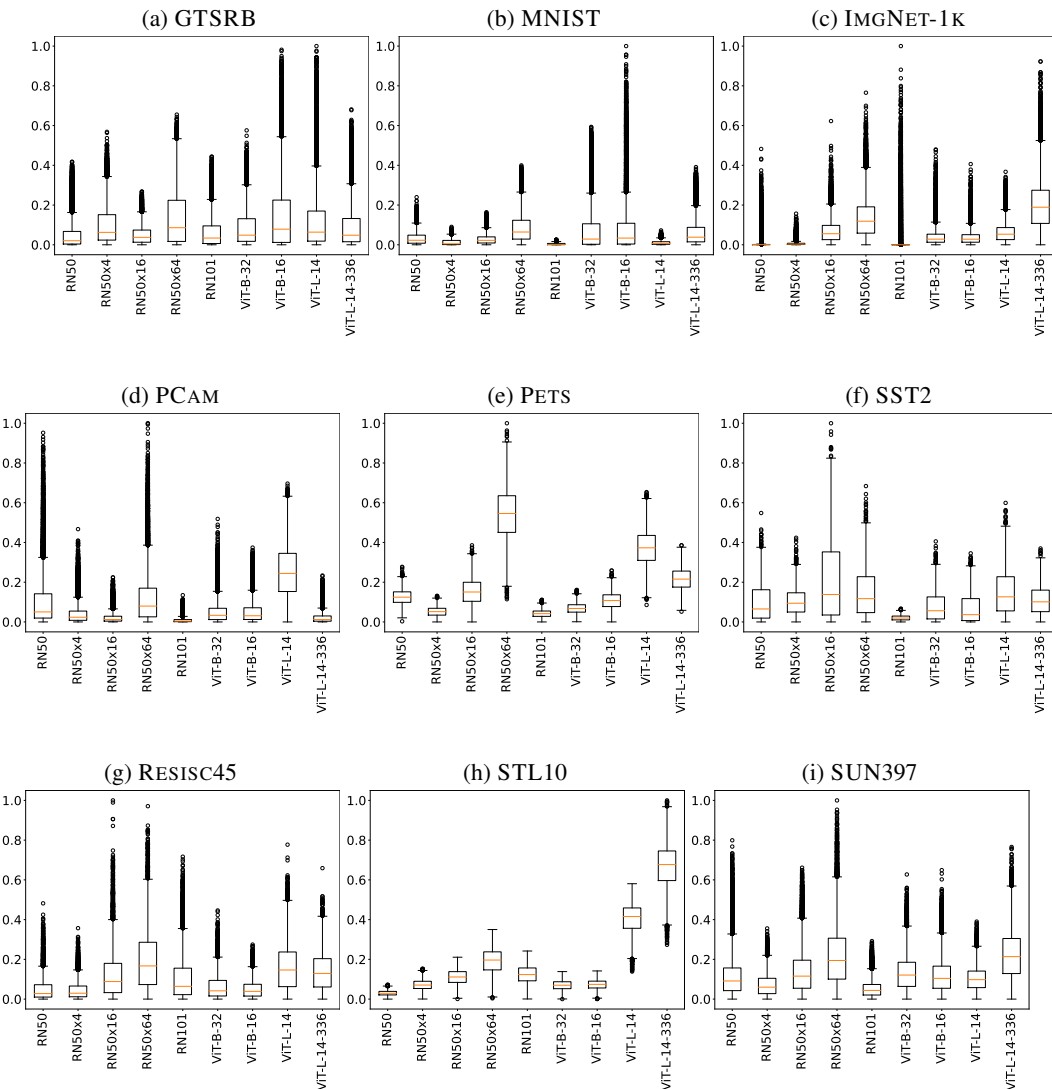

Figure H.2: Alpha values for each dataset using NLC

Table I.1: ORACLE performance by combining: 1) ViT **backbones** pretrained with same datasets. 2) Same ViT backbone trained on different **datasets** 3) Same ViT backbone trained on same dataset but two different **epochs**.

| | | Average | CALTECH101 | CARS | CIFAR10 | CIFAR100 | CLEVR | COUNTRY211 | CUB | DTD | EUROSAT | FGVC | FLOWERS | FOOD | GTSRB | IMGNET-1K | MNIST | PCAM | PETS | SST2 | RESISC45 | STL10 | SUN397 |
|---|---|---|---|---|---|---|---|---|---|---|---|---|---|---|---|---|---|---|---|---|---|---|---|
| Backbone | OpenAI | 78.2 | 91.6 | 87.4 | 98.2 | 85.9 | 33.6 | 39.1 | 76.6 | 65.3 | 70.0 | 46.6 | 85.0 | 95.8 | 64.9 | 82.9 | 83.0 | 93.8 | 97.0 | 90.6 | 76.5 | 99.6 | 79.1 |
| | LAION-400M_e31 | 76.9 | 91.7 | 94.9 | 97.4 | 85.9 | 39.9 | 30.0 | 83.3 | 70.3 | 68.0 | 36.2 | 81.6 | 93.3 | 59.7 | 79.9 | 77.1 | 79.8 | 96.1 | 90.6 | 77.5 | 98.8 | 82.1 |
| | LAION-400M_e32 | 76.8 | 91.6 | 94.9 | 97.4 | 85.9 | 39.6 | 30.1 | 83.3 | 70.5 | 67.5 | 36.8 | 81.5 | 93.4 | 59.8 | 80.0 | 76.6 | 80.2 | 96.1 | 90.3 | 77.3 | 98.9 | 82.1 |
| Dataset | ViT-B-16 | 73.7 | 90.8 | 88.9 | 95.9 | 80.0 | 45.7 | 28.8 | 75.1 | 60.4 | 57.2 | 33.3 | 79.2 | 92.8 | 55.8 | 77.6 | 76.7 | 80.8 | 94.7 | 84.3 | 72.9 | 98.9 | 78.2 |
| | ViT-B-32 | 69.1 | 88.1 | 82.9 | 94.7 | 77.4 | 26.9 | 22.4 | 68.6 | 62.3 | 60.7 | 27.4 | 76.1 | 87.9 | 51.9 | 72.9 | 60.5 | 74.9 | 93.4 | 83.3 | 65.5 | 98.0 | 74.7 |
| | ViT-L-14 | 77.4 | 90.7 | 93.6 | 98.1 | 86.6 | 31.4 | 38.1 | 81.8 | 69.7 | 62.1 | 44.0 | 85.1 | 95.5 | 62.8 | 82.8 | 83.8 | 79.4 | 96.8 | 84.3 | 77.4 | 99.6 | 80.8 |
| Epoch | ViT-B-16 | 62.9 | 85.7 | 84.2 | 92.0 | 72.1 | 29.2 | 18.7 | 65.8 | 51.8 | 45.4 | 18.3 | 70.0 | 85.1 | 44.4 | 67.8 | 58.0 | 59.8 | 89.7 | 55.6 | 60.4 | 97.1 | 70.2 |
| | ViT-B-32 | 57.8 | 83.5 | 75.3 | 88.8 | 68.6 | 15.6 | 13.9 | 57.5 | 53.1 | 45.1 | 15.5 | 66.8 | 76.4 | 41.4 | 60.9 | 45.3 | 56.2 | 85.9 | 52.7 | 52.3 | 95.0 | 65.0 |
| | ViT-L-14 | 67.2 | 88.1 | 90.0 | 94.9 | 78.1 | 24.6 | 23.6 | 74.0 | 60.5 | 50.9 | 25.9 | 76.0 | 89.6 | 50.5 | 73.3 | 71.4 | 50.7 | 92.2 | 57.5 | 67.6 | 98.1 | 73.1 |

Table I.2: Diversity of combining: 1) ViT **backbones** pretrained with same datasets. 2) Same ViT backbone trained on different **datasets** 3) Same ViT backbone trained on same dataset but two different **epochs**.

| | | Average | CALTECH101 | CARS | CIFAR10 | CIFAR100 | CLEVR | COUNTRY211 | CUB | DTD | EUROSAT | FGVC | FLOWERS | FOOD | GTSRB | IMGNET-1K | MNIST | PCAM | PETS | SST2 | RESISC45 | STL10 | SUN397 |
|---|---|---|---|---|---|---|---|---|---|---|---|---|---|---|---|---|---|---|---|---|---|---|---|
| Backbone | OpenAI | 47.2 | 13.3 | 47.5 | 14.3 | 40.8 | 70.4 | 71.8 | 51.3 | 50.8 | 76.7 | 81.2 | 32.4 | 19.3 | 68.1 | 34.8 | 66.7 | 80.1 | 15.9 | 65.8 | 46.5 | 3.2 | 39.7 |
| | LAION400M_e31 | 43.7 | 13.6 | 29.6 | 13.7 | 33.6 | 69.0 | 72.7 | 46.0 | 45.3 | 62.7 | 83.0 | 29.6 | 24.5 | 50.8 | 34.9 | 53.6 | 68.3 | 16.7 | 82.5 | 47.7 | 4.9 | 34.6 |
| | LAION400M_e32 | 43.6 | 13.6 | 29.3 | 13.6 | 33.5 | 68.6 | 72.9 | 45.7 | 45.7 | 62.1 | 83.1 | 29.9 | 24.4 | 50.9 | 35.0 | 53.6 | 67.9 | 17.1 | 81.8 | 47.5 | 4.9 | 34.5 |
| Dataset | ViT-B-16 | 37.7 | 10.6 | 33.2 | 9.7 | 27.3 | 90.0 | 57.5 | 39.7 | 40.6 | 45.3 | 73.4 | 22.5 | 14.5 | 43.7 | 25.8 | 47.9 | 64.0 | 11.8 | 64.1 | 38.5 | 2.7 | 29.6 |
| | ViT-B-32 | 39.8 | 10.3 | 37.8 | 12.0 | 29.4 | 56.5 | 62.4 | 40.3 | 45.1 | 66.3 | 73.7 | 26.0 | 20.3 | 58.0 | 30.8 | 66.0 | 44.0 | 15.1 | 67.2 | 38.7 | 4.1 | 31.9 |
| | ViT-L-14 | 32.0 | 8.2 | 21.3 | 6.2 | 23.3 | 60.8 | 56.1 | 34.5 | 34.8 | 43.4 | 71.7 | 18.6 | 10.2 | 39.9 | 21.2 | 28.9 | 72.4 | 8.8 | 52.5 | 30.1 | 1.8 | 27.7 |
| Epoch | ViT-B-16 | 2.7 | 0.8 | 1.4 | 0.6 | 2.4 | 3.2 | 6.2 | 2.9 | 2.5 | 5.4 | 7.7 | 1.9 | 1.4 | 4.1 | 2.1 | 2.9 | 1.3 | 0.9 | 4.1 | 3.5 | 0.3 | 1.9 |
| | ViT-B-32 | 3.3 | 0.7 | 1.9 | 1.0 | 2.4 | 2.1 | 5.7 | 3.8 | 3.3 | 6.0 | 11.2 | 2.3 | 1.7 | 3.0 | 2.3 | 6.9 | 4.7 | 1.2 | 3.2 | 2.6 | 0.2 | 2.1 |
| | ViT-L-14 | 2.6 | 0.7 | 1.0 | 0.6 | 1.7 | 3.4 | 5.8 | 2.3 | 2.1 | 3.4 | 8.8 | 1.3 | 0.9 | 3.1 | 1.7 | 2.9 | 5.7 | 0.9 | 4.5 | 2.4 | 0.2 | 1.8 |

## K  IMAGE-TEXT RETRIEVAL

We present additional image and text retrieval experiments in Table K.3 that hint at the potential benefits of combining multiple backbones of CLIP. We evaluated the complementarity of CLIP backbones on Flickr30k and MSCOCO. We measured the upper-bound improvement of an "oracle selection" of backbones as described in Section 2. Table K.3 shows patterns of complementarity across backbones comparable to those in the classification setting. On Flick30k, the upper bound shows possible improvements of 20 percentage points on both text and image retrieval. On MSCOCO, >25pp.

Table K.3: Zero-shot retrieval performance of CLIP backbones on the Flickr30k and MSCOCO benchmark and the performance of our empirical upper-bound, denoted as ORACLE, representing the ideal combination of ResNet (RN), ViT, and All backbones

| | | Text Retrieval | | | | | | Image Retrieval | | | | | |
|---|---|---|---|---|---|---|---|---|---|---|---|---|---|
| | | Flickr30k | | | MSCOCO | | | Flickr30k | | | MSCOCO | | |
| | | R@1 | R@5 | R@10 | R@1 | R@5 | R@10 | R@1 | R@5 | R@10 | R@1 | R@5 | R@10 |
| ResNet | 50 | 80.4 | 96.1 | 98.1 | 48.1 | 73.9 | 83.0 | 59.2 | 85.9 | 91.6 | 28.3 | 53.0 | 64.1 |
| | 50x4 | 84.8 | 96.6 | 98.9 | 52.1 | 76.6 | 84.5 | 64.6 | 87.3 | 92.7 | 32.5 | 56.7 | 67.1 |
| | 50x16 | 86.2 | 97.8 | 99.5 | 55.3 | 78.2 | 86.3 | 67.4 | 88.9 | 93.6 | 35.2 | 59.5 | 69.6 |
| | 50x64 | 88.3 | 98.4 | 99.6 | 57.8 | 80.6 | 87.7 | 71.4 | 91.6 | 95.5 | 34.7 | 60.0 | 70.0 |
| | 101 | 83.8 | 96.5 | 98.3 | 49.8 | 74.4 | 82.7 | 62.5 | 85.7 | 91.3 | 30.2 | 54.2 | 65.3 |
| ViT | B-32 | 82.2 | 95.1 | 97.5 | 50.0 | 75.0 | 83.2 | 61.4 | 86.0 | 91.7 | 30.4 | 54.8 | 66.1 |
| | B-16 | 85.1 | 97.3 | 98.9 | 51.8 | 76.8 | 84.3 | 65.2 | 87.8 | 92.8 | 32.7 | 57.8 | 68.3 |
| | L-14 | 87.0 | 98.4 | 99.8 | 56.1 | 79.6 | 86.9 | 67.8 | 89.8 | 94.3 | 35.3 | 59.6 | 70.1 |
| | L-14-336 | 88.1 | 98.2 | 99.6 | 57.5 | 80.4 | 87.6 | 71.5 | 91.8 | 95.5 | 36.1 | 60.7 | 70.8 |
| ORACLE ResNet | | **97.9** | **99.9** | **100.0** | **77.7** | **93.0** | **96.3** | **86.6** | **96.8** | **98.4** | **54.7** | **76.9** | **84.9** |
| ORACLE ViT | | **96.3** | **99.5** | **100.0** | **73.0** | **90.4** | **94.6** | **84.3** | **96.1** | **97.9** | **50.9** | **74.1** | **82.5** |
| ORACLE All | | **98.8** | **100.0** | **100.0** | **83.3** | **95.3** | **97.5** | **90.3** | **97.8** | **98.8** | **61.9** | **82.1** | **88.9** |

Figure J.3: Average accuracy across 21 datasets for zero-shot ResNets and ViTs backbones, LOG-AVG Ensemble, VOTE T-1 Ensemble, C-LOG-AVG Ensemble, NLC Ensemble, and NLC Cascade using 2 to 9 backbones, plotted against the Average GFLOPs. Demonstrates that our NLC ensembles can surpass the best zero-shot backbone, and the NLC cascade can also surpass the best zero-shot backbone with fewer GFLOPs. Moreover, our method surpasses the standard ensembling techniques.

## L    OVERLAP DIAGRAMS FOR OTHER DATASETS.

In this section, we present the linear Venn Diagrams for each of the other datasets used in the experiment section CALTECH (Figure L.4), CARS (Figure L.5), CUB (Figure L.10), CIFAR10 (Figure L.6), CIFAR100 (Figure L.7), CLEVR (Figure L.8), COUNTRY211 (Figure L.9), CUB (Figure L.10), DTD (Figure L.11), EUROSAT(Figure L.12), FGVC (Figure L.13), FLOWERS (Figure L.14), FOOD (Figure L.15), GTSRB (Figure L.16), MNIST (Figure L.17), PCAM (Figure L.18), PETS (Figure L.19), RenderedSST2 (Figure L.20), RESISC45 (Figure L.21), STL10 (Figure L.22), and SUN397 (Figure L.23). We can see that in each dataset, the CLIP backbones present possible complementarities that could be exploited.

## M    POSSIBLE COMBINATIONS

Tables M.4, M.5, M.6, M.7, M.8, M.9, M.10, and M.11 present the results of possible combinations of backbones using the non-parametric and parametric approaches proposed in the paper. Notably, the performance of NLC consistently emerges as the best across various backbone combinations and datasets when compared to other methods.

Notably, instances exist where the combination of specific backbones yields a more substantial performance boost than utilizing all backbones together. For instance, in the PETS dataset, combining ResNet 50, 101, and ViT-B-32 results in a delta improvement of 2.37%, surpassing the 0.99% improvement achieved by using the five backbones selected for this experiment. This phenomenon is consistent across datasets with different backbone combinations. In CARS, there is a boost of 5.71% when combining ResNet-101 and ViT-B-32, compared to the 2.55% boost when using all five different backbones. While the best delta improvement among backbones may not necessarily come from combining all backbones, the best overall accuracy is consistently obtained when using the combination of all backbones.

CALTECH101

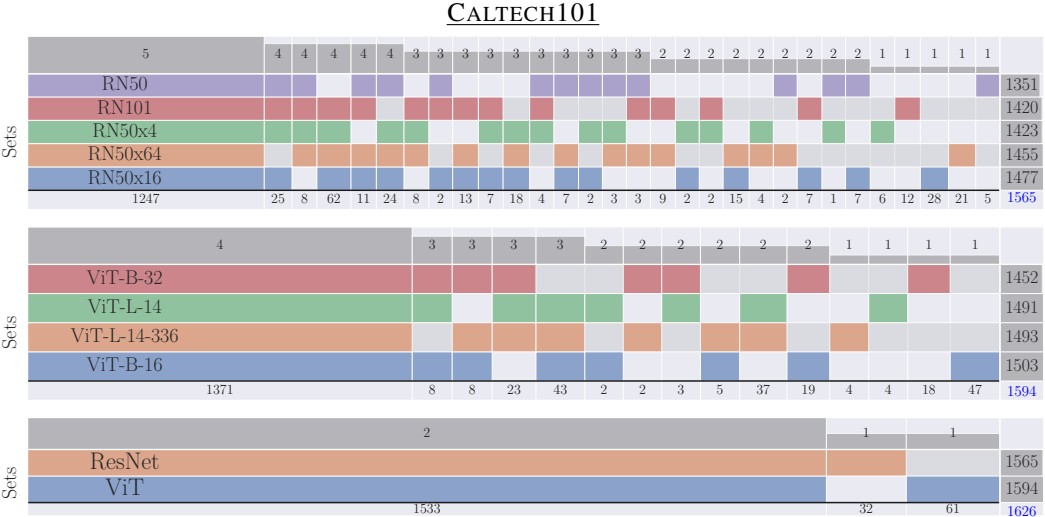

Figure L.4: CALTECH101 Overlap diagrams with the correct prediction of each backbone. The Top part of the Overlap diagram shows the number of backbones that are predicting correctly a set of images. Each column represents a set of image instances that are predicted correctly by some group of backbones. Each row in the diagram shows in colour the backbone that correctly predicts a certain set of image instances, in grey when the backbone is not correctly predicting those instances. The bottom part of the Overlap diagram shows the number of images in a certain set. The right part is the total amount of correctly predicted images per backbone.

CARS

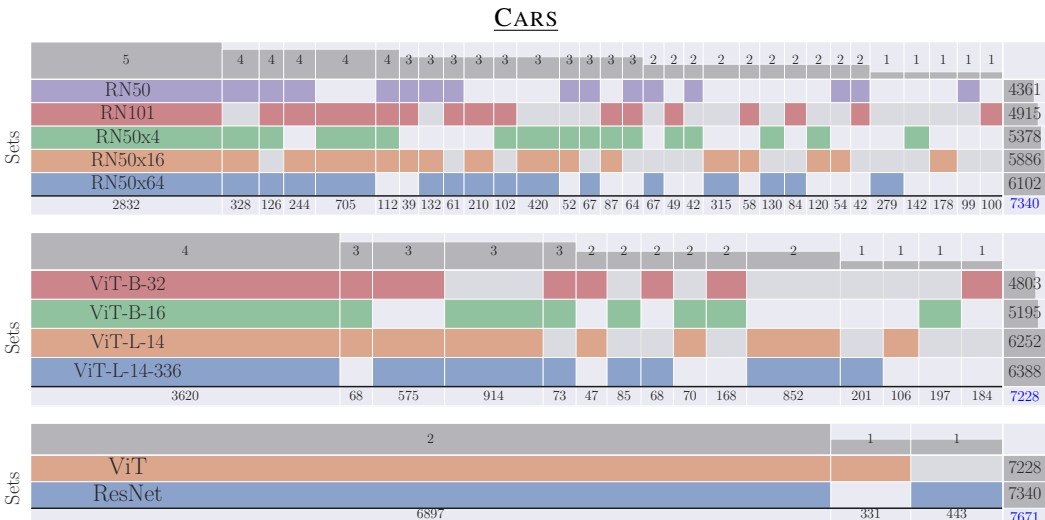

Figure L.5: Overlap diagrams for the CARS dataset with the correct prediction of each backbone. The Top part of the Overlap diagram shows the number of backbones that are predicting correctly a set of images. Each column represents a set of image instances that are predicted correctly by some group of backbones. Each row in the diagram shows in colour the backbone that correctly predicts a certain set of image instances, in grey when the backbone is not correctly predicting those instances. The bottom part of the Overlap diagram shows the number of images in a certain set. The right part is the total amount of correctly predicted images per backbone.

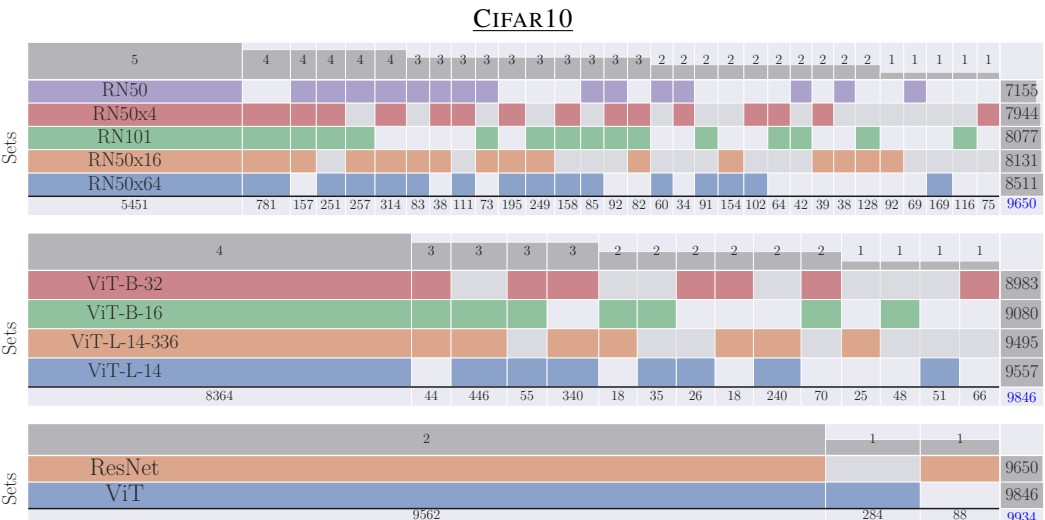

Figure L.6: CIFAR10 Overlap diagrams with the correct prediction of each backbone. The Top part of the Overlap diagram shows the number of backbones that are predicting correctly a set of images. Each column represents a set of image instances that are predicted correctly by some group of backbones. Each row in the diagram shows in colour the backbone that correctly predicts a certain set of image instances, in grey when the backbone is not correctly predicting those instances. The bottom part of the Overlap diagram shows the number of images in a certain set. The right part is the total amount of correctly predicted images per backbone.

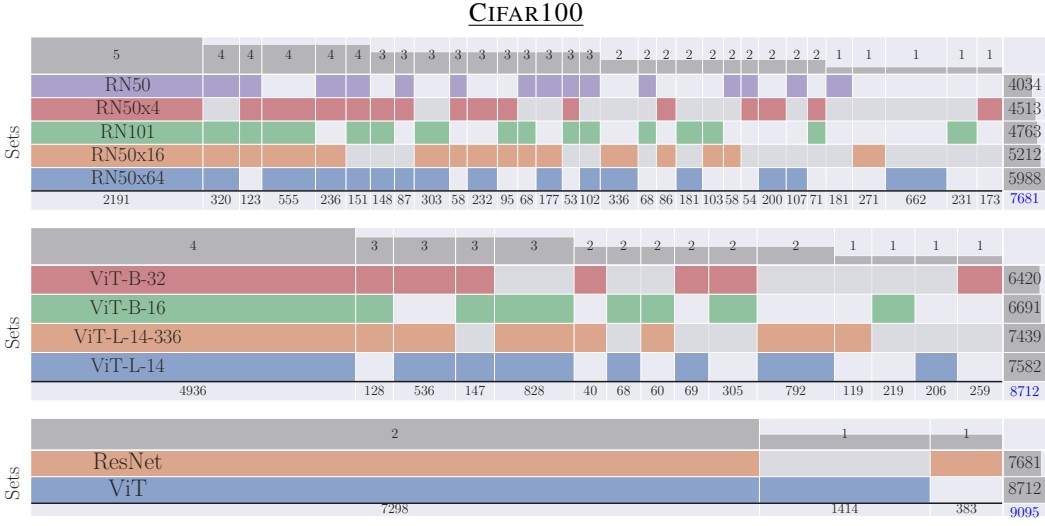

Figure L.7: CIFAR100 Venn diagrams with the correct prediction of each backbone. The Top part of the Venn diagram shows the number of backbones that are predicting correctly a set of images. Each column represents a set of image instances that are predicted correctly by some group of backbones. Each row in the diagram shows in colour the backbone that correctly predicts a certain set of image instances, in grey when the backbone is not correctly predicting those instances. The bottom part of the Venn diagram shows the number of images in a certain set. The right part is the total amount of correctly predicted images per backbone.

CLEVER

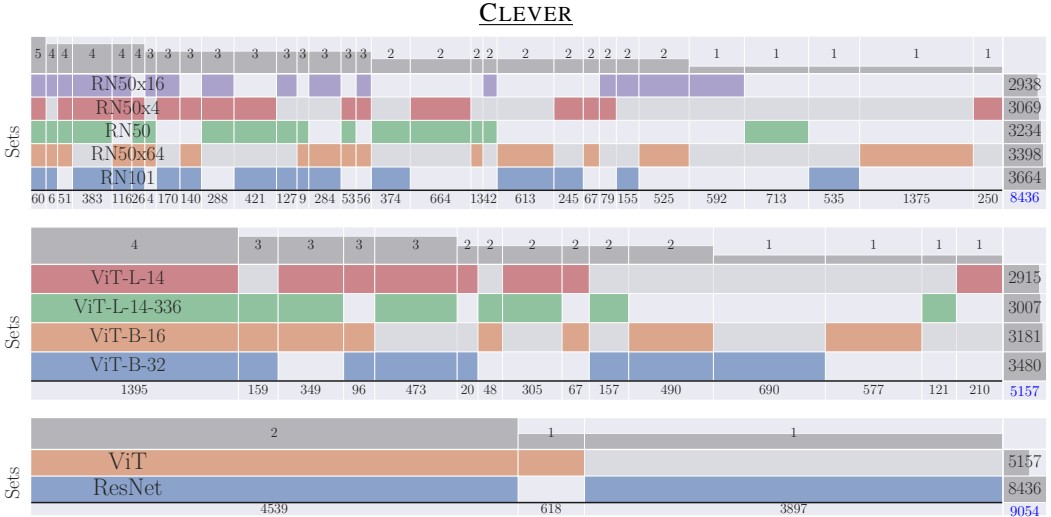

Figure L.8: CLEVER Overlap diagrams with the correct prediction of each backbone. The Top part of the Overlap diagram shows the number of backbones that are predicting correctly a set of images. Each column represents a set of image instances that are predicted correctly by some group of backbones. Each row in the diagram shows in colour the backbone that correctly predicts a certain set of image instances, in grey when the backbone is not correctly predicting those instances. The bottom part of the Overlap diagram shows the number of images in a certain set. The right part is the total amount of correctly predicted images per backbone.

COUNTRY211

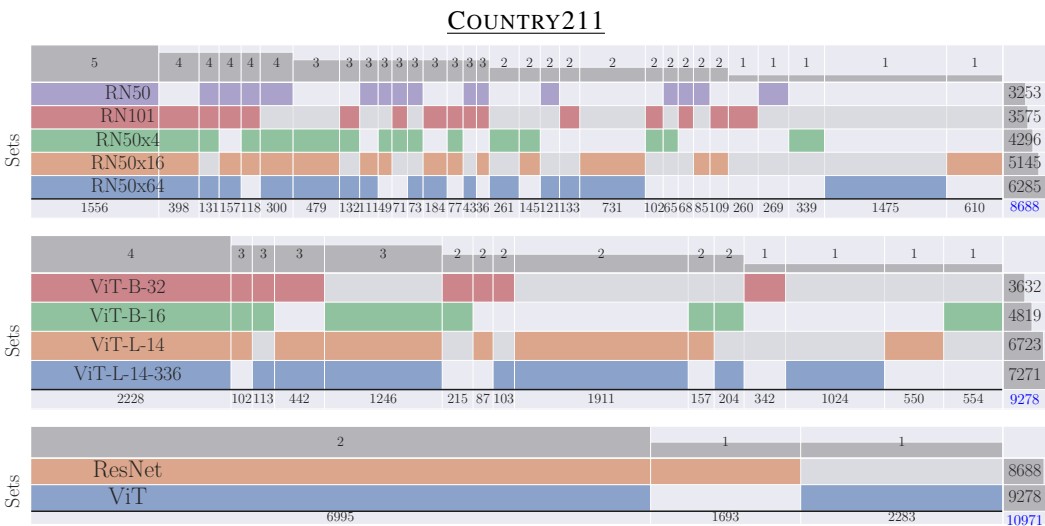

Figure L.9: COUNTRY211 Overlap diagrams with the correct prediction of each backbone. The Top part of the Overlap diagram shows the number of backbones that are predicting correctly a set of images. Each column represents a set of image instances that are predicted correctly by some group of backbones. Each row in the diagram shows in colour the backbone that correctly predicts a certain set of image instances, in grey when the backbone is not correctly predicting those instances. The bottom part of the Overlap diagram shows the number of images in a certain set. The right part is the total amount of correctly predicted images per backbone.

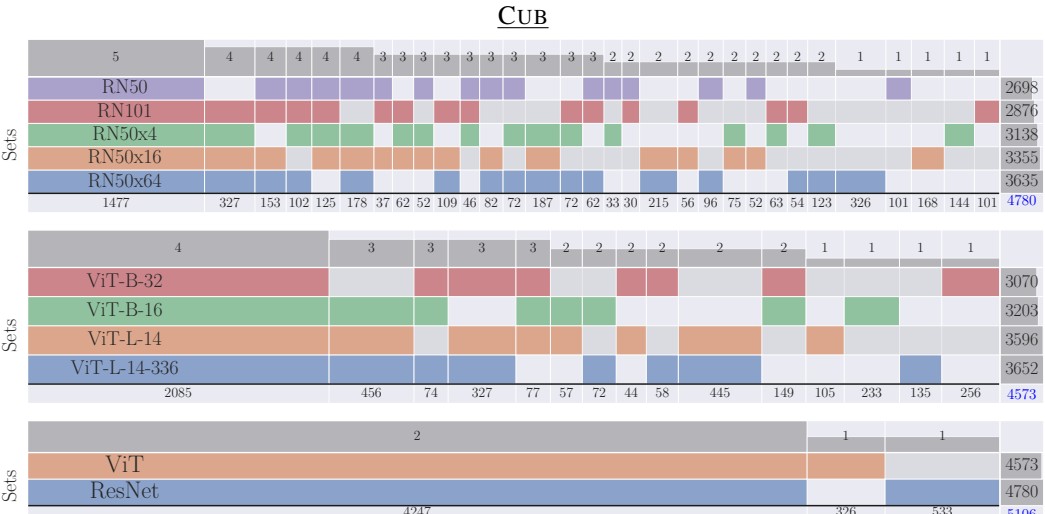

Figure L.10: CUB Overlap diagrams with the correct prediction of each backbone. The Top part of the Overlap diagram shows the number of backbones that are predicting correctly a set of images. Each column represents a set of image instances that are predicted correctly by some group of backbones. Each row in the diagram shows in colour the backbone that correctly predicts a certain set of image instances, in grey when the backbone is not correctly predicting those instances. The bottom part of the Overlap diagram shows the number of images in a certain set. The right part is the total amount of correctly predicted images per backbone.

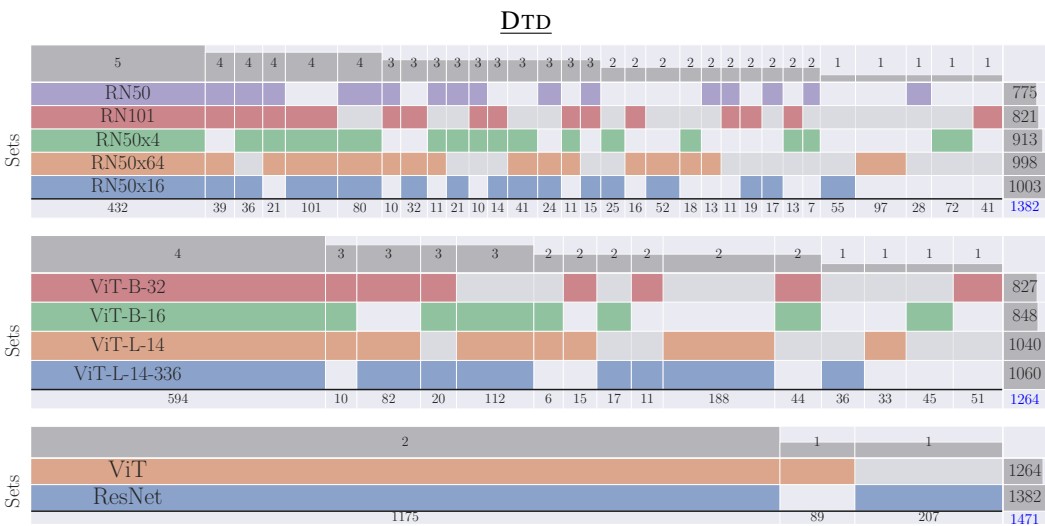

Figure L.11: DTD Overlap diagrams with the correct prediction of each backbone. The Top part of the Overlap diagram shows the number of backbones that are predicting correctly a set of images. Each column represents a set of image instances that are predicted correctly by some group of backbones. Each row in the diagram shows in colour the backbone that correctly predicts a certain set of image instances, in grey when the backbone is not correctly predicting those instances. The bottom part of the Overlap diagram shows the number of images in a certain set. The right part is the total amount of correctly predicted images per backbone.

EUROSAT

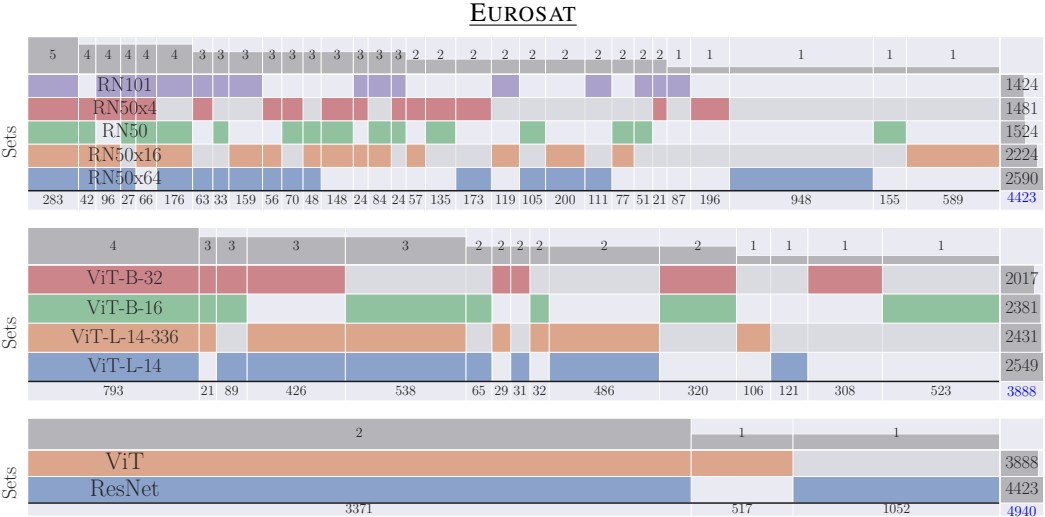

Figure L.12: EUROSAT Overlap diagrams with the correct prediction of each backbone. The Top part of the Overlap diagram shows the number of backbones that are predicting correctly a set of images. Each column represents a set of image instances that are predicted correctly by some group of backbones. Each row in the diagram shows in colour the backbone that correctly predicts a certain set of image instances, in grey when the backbone is not correctly predicting those instances. The bottom part of the Overlap diagram shows the number of images in a certain set. The right part is the total amount of correctly predicted images per backbone.

FGVC

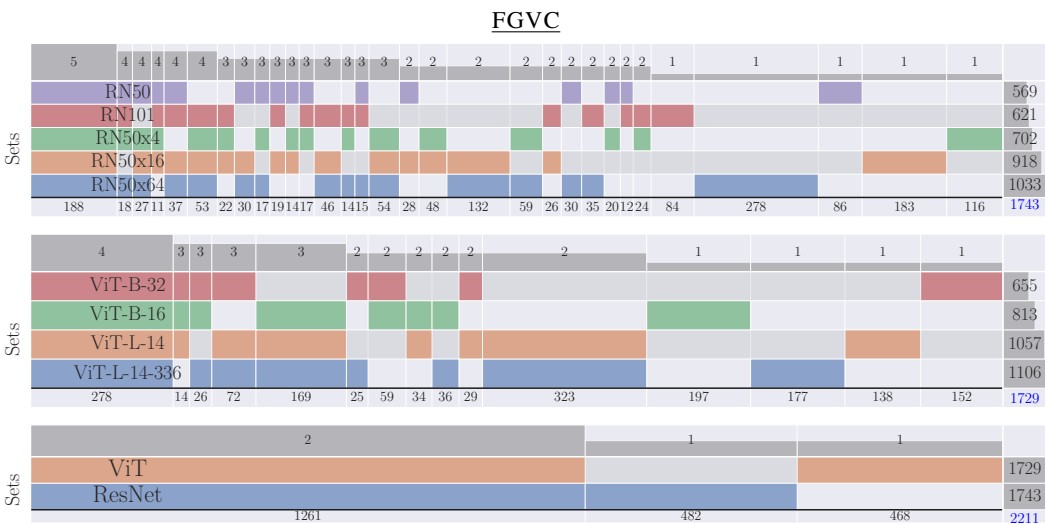

Figure L.13: FGVC Overlap diagrams with the correct prediction of each backbone. The Top part of the Overlap diagram shows the number of backbones that are predicting correctly a set of images. Each column represents a set of image instances that are predicted correctly by some group of backbones. Each row in the diagram shows in colour the backbone that correctly predicts a certain set of image instances, in grey when the backbone is not correctly predicting those instances. The bottom part of the Overlap diagram shows the number of images in a certain set. The right part is the total amount of correctly predicted images per backbone.

FLOWERS

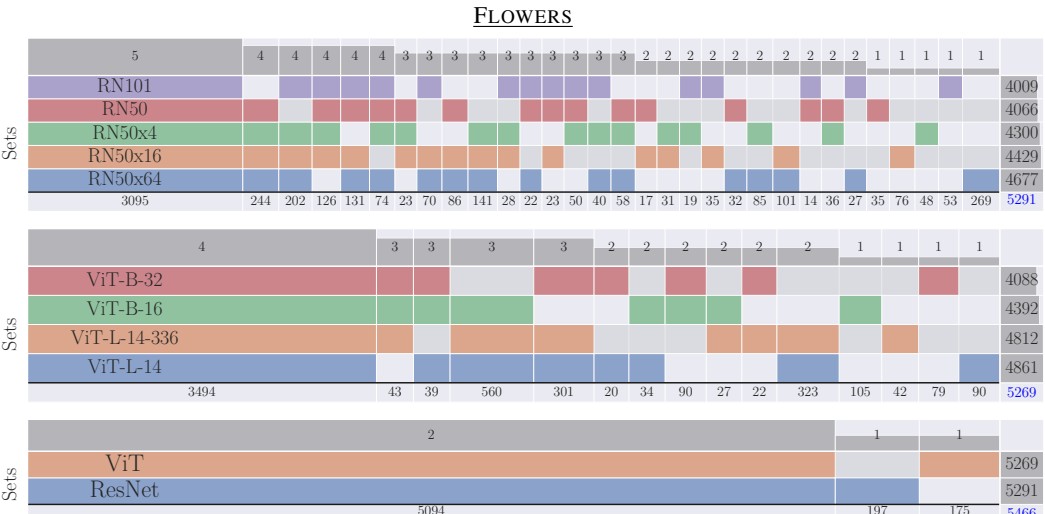

Figure L.14: FLOWERS Overlap diagrams with the correct prediction of each backbone. The Top part of the Overlap diagram shows the number of backbones that are predicting correctly a set of images. Each column represents a set of image instances that are predicted correctly by some group of backbones. Each row in the diagram shows in colour the backbone that correctly predicts a certain set of image instances, in grey when the backbone is not correctly predicting those instances. The bottom part of the Overlap diagram shows the number of images in a certain set. The right part is the total amount of correctly predicted images per backbone.

FOOD

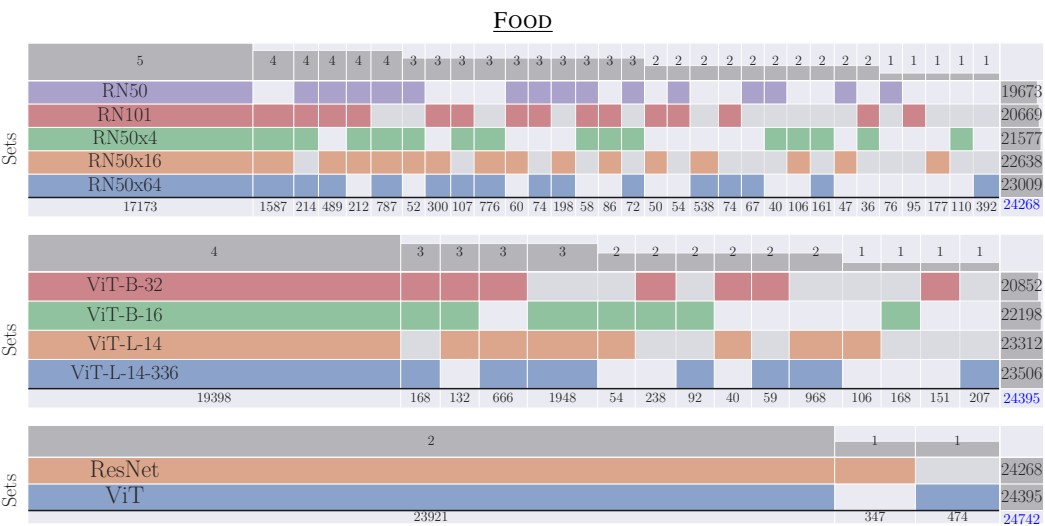

Figure L.15: FOOD Overlap diagrams with the correct prediction of each backbone. The Top part of the Overlap diagram shows the number of backbones that are predicting correctly a set of images. Each column represents a set of image instances that are predicted correctly by some group of backbones. Each row in the diagram shows in colour the backbone that correctly predicts a certain set of image instances, in grey when the backbone is not correctly predicting those instances. The bottom part of the Overlap diagram shows the number of images in a certain set. The right part is the total amount of correctly predicted images per backbone.

GTSRB

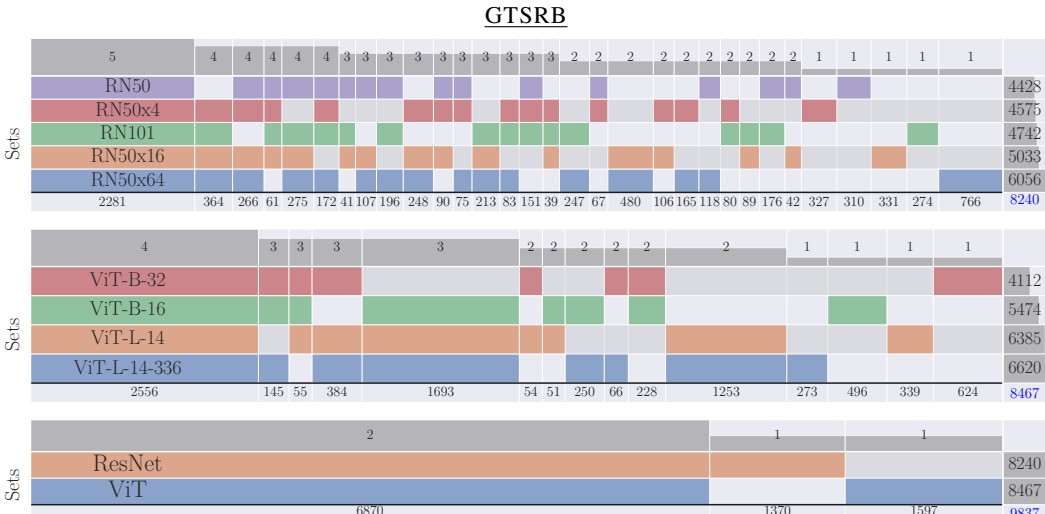

Figure L.16: GTSRB Overlap diagrams with the correct prediction of each backbone. The Top part of the Overlap diagram shows the number of backbones that are predicting correctly a set of images. Each column represents a set of image instances that are predicted correctly by some group of backbones. Each row in the diagram shows in colour the backbone that correctly predicts a certain set of image instances, in grey when the backbone is not correctly predicting those instances. The bottom part of the Overlap diagram shows the number of images in a certain set. The right part is the total amount of correctly predicted images per backbone.

MNIST

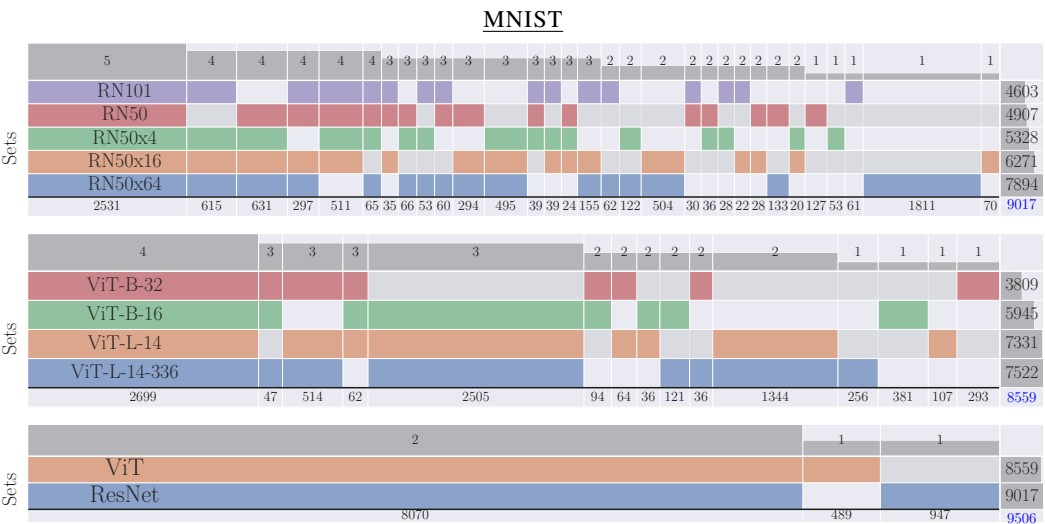

Figure L.17: MNIST Overlap diagrams with the correct prediction of each backbone. The Top part of the Overlap diagram shows the number of backbones that are predicting correctly a set of images. Each column represents a set of image instances that are predicted correctly by some group of backbones. Each row in the diagram shows in colour the backbone that correctly predicts a certain set of image instances, in grey when the backbone is not correctly predicting those instances. The bottom part of the Overlap diagram shows the number of images in a certain set. The right part is the total amount of correctly predicted images per backbone.

PCAM

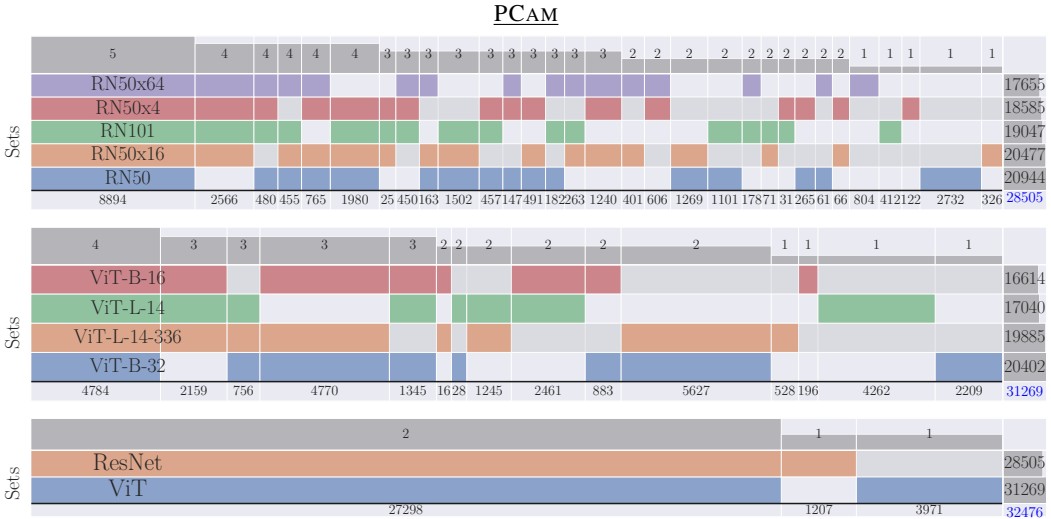

Figure L.18: PCAM Overlap diagrams with the correct prediction of each backbone. The Top part of the Overlap diagram shows the number of backbones that are predicting correctly a set of images. Each column represents a set of image instances that are predicted correctly by some group of backbones. Each row in the diagram shows in colour the backbone that correctly predicts a certain set of image instances, in grey when the backbone is not correctly predicting those instances. The bottom part of the Overlap diagram shows the number of images in a certain set. The right part is the total amount of correctly predicted images per backbone.

PETS

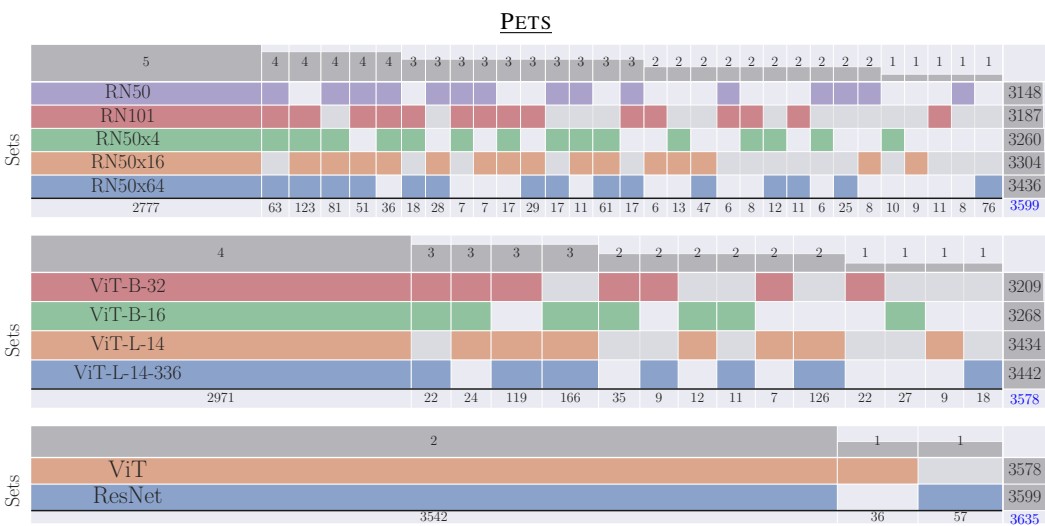

Figure L.19: PETS Overlap diagrams with the correct prediction of each backbone. The Top part of the Overlap diagram shows the number of backbones that are predicting correctly a set of images. Each column represents a set of image instances that are predicted correctly by some group of backbones. Each row in the diagram shows in colour the backbone that correctly predicts a certain set of image instances, in grey when the backbone is not correctly predicting those instances. The bottom part of the Overlap diagram shows the number of images in a certain set. The right part is the total amount of correctly predicted images per backbone.

RENDERSST2

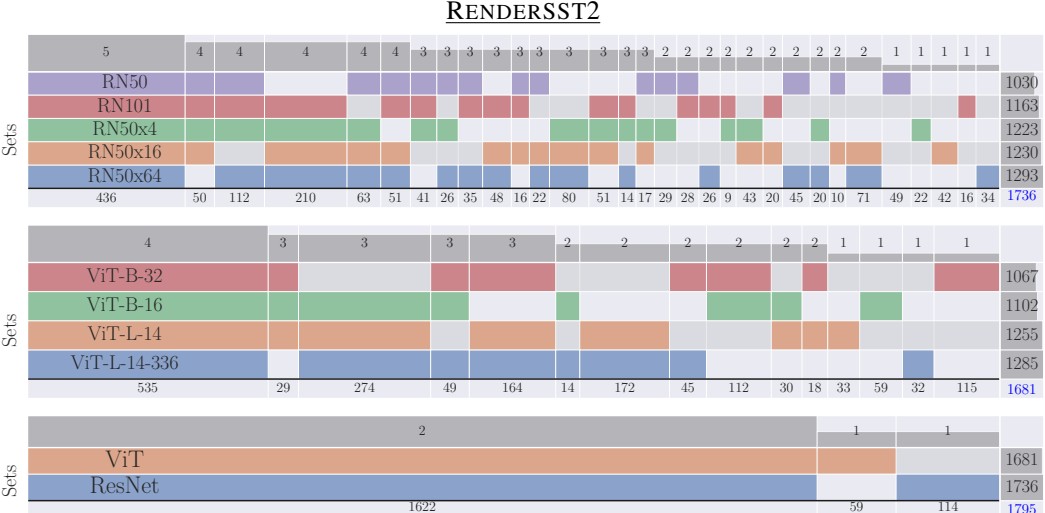

Figure L.20: RENDERSST2 Overlap diagrams with the correct prediction of each backbone. The Top part of the Overlap diagram shows the number of backbones that are predicting correctly a set of images. Each column represents a set of image instances that are predicted correctly by some group of backbones. Each row in the diagram shows in colour the backbone that correctly predicts a certain set of image instances, in grey when the backbone is not correctly predicting those instances. The bottom part of the Overlap diagram shows the number of images in a certain set. The right part is the total amount of correctly predicted images per backbone.

RESISC45

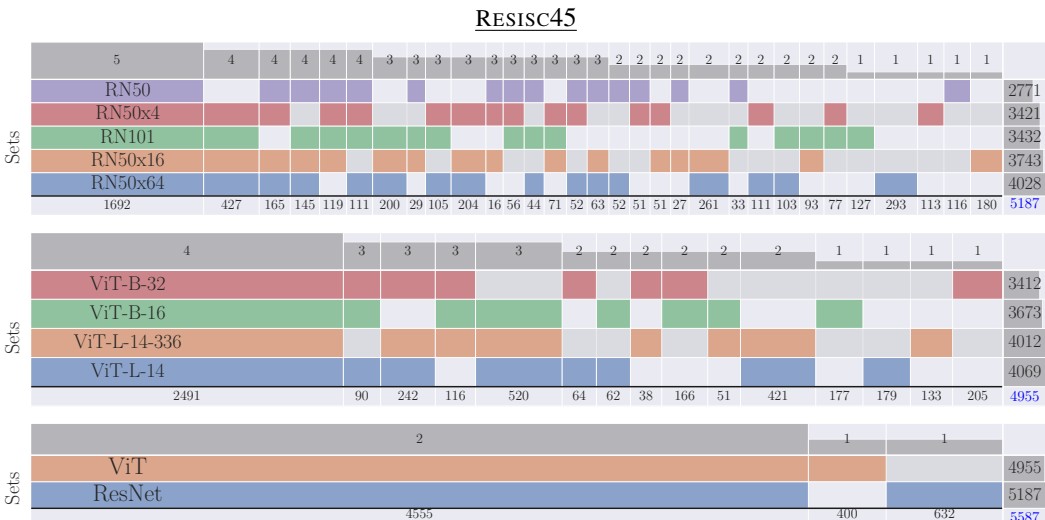

Figure L.21: RESISC45 Overlap diagrams with the correct prediction of each backbone. The Top part of the Overlap diagram shows the number of backbones that are predicting correctly a set of images. Each column represents a set of image instances that are predicted correctly by some group of backbones. Each row in the diagram shows in colour the backbone that correctly predicts a certain set of image instances, in grey when the backbone is not correctly predicting those instances. The bottom part of the Overlap diagram shows the number of images in a certain set. The right part is the total amount of correctly predicted images per backbone.

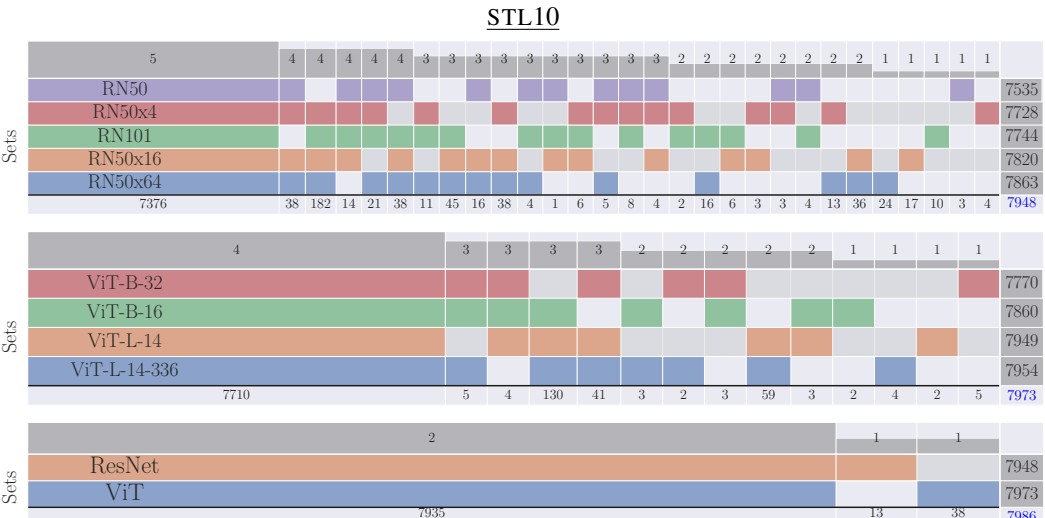

Figure L.22: STL10 Overlap diagrams with the correct prediction of each backbone. The Top part of the Overlap diagram shows the number of backbones that are predicting correctly a set of images. Each column represents a set of image instances that are predicted correctly by some group of backbones. Each row in the diagram shows in colour the backbone that correctly predicts a certain set of image instances, in grey when the backbone is not correctly predicting those instances. The bottom part of the Overlap diagram shows the number of images in a certain set. The right part is the total amount of correctly predicted images per backbone.

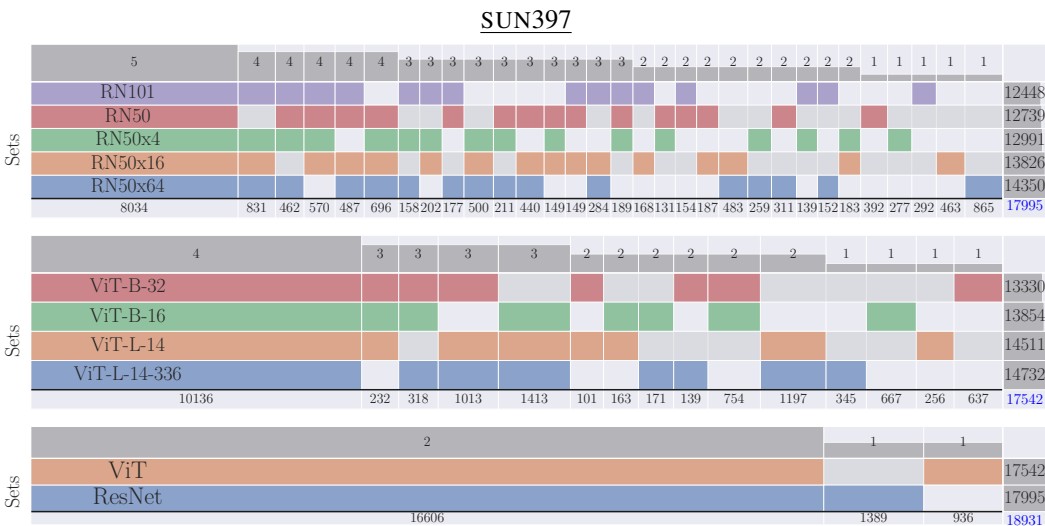

Figure L.23: SUN397 Overlap diagrams with the correct prediction of each backbone. The Top part of the Overlap diagram shows the number of backbones that are predicting correctly a set of images. Each column represents a set of image instances that are predicted correctly by some group of backbones. Each row in the diagram shows in colour the backbone that correctly predicts a certain set of image instances, in grey when the backbone is not correctly predicting those instances. The bottom part of the Overlap diagram shows the number of images in a certain set. The right part is the total amount of correctly predicted images per backbone.

Table M.4: Our results on PETS dataset for all the possible combinations of combining the zero-shot predictions of CLIP backbones, which we group intro non-parametric and parametric techniques. Also, the best-performing single backbone (SINGLE-BEST) and the ORACLE performance. We present, for each combination of backbones, the improvement , constancy — and deterioration of accuracy performance for each method when we compare it against the SINGLE-BEST backbone. Mean, Max, and Min $\Delta$ summarize the difference in performance across methods and backbone combinations.

**PETS**

| ResNet | | ViT | | | SINGLE-BEST | Non-Parametric | | | | Parametric | | | | ORACLE |
|---|---|---|---|---|---|---|---|---|---|---|---|---|---|---|
| 50 | 101 | B-32 | B-16 | L-14 | | VOTE T-1 | VOTE T-3 | CONF | LOG-AVG | C-CONF | C-LOG-AVG | GAC | NLC | |
| ✓ | | | | | | | | | | | 85.80 −0.00 | | | |
| | ✓ | | | | | | | | | | 86.86 −0.00 | | | |
| | | ✓ | | | | | | | | | 87.46 −0.00 | | | |
| | | | ✓ | | | | | | | | 89.07 −0.00 | | | |
| | | | | ✓ | | | | | | | 93.59 −0.00 | | | |
| ✓ | ✓ | | | | 86.86 | 87.84 0.98 | 87.90 1.04 | 88.20 1.34 | 88.03 1.17 | 87.93 1.06 | 87.84 0.98 | 87.14 0.27 | 88.09 1.23 | 91.88 5.01 |
| ✓ | | ✓ | | | 87.46 | 88.50 1.04 | 89.02 1.55 | 85.25 -2.21 | 89.40 1.94 | 87.98 0.52 | 89.21 1.74 | 89.48 2.02 | 89.45 1.99 | 92.64 5.18 |
| ✓ | | | ✓ | | 89.07 | 89.83 0.76 | 89.92 0.84 | 89.67 0.6 | 89.83 0.76 | 89.48 0.41 | 89.72 0.65 | 89.86 0.79 | 90.27 1.2 | 92.89 3.82 |
| ✓ | | | | ✓ | 93.59 | 94.03 0.44 | 94.00 0.41 | 94.00 0.41 | 93.81 0.22 | 93.73 0.14 | 93.98 0.38 | 93.73 0.14 | 94.19 0.6 | 96.18 2.59 |
| | ✓ | ✓ | | | 87.46 | 88.72 1.25 | 88.99 1.53 | 85.91 -1.55 | 89.07 1.61 | 88.55 1.09 | 89.07 1.61 | 89.10 1.64 | 89.04 1.58 | 93.00 5.53 |
| | ✓ | | ✓ | | 89.07 | 90.00 0.93 | 90.11 1.04 | 89.75 0.68 | 89.89 0.82 | 89.81 0.74 | 90.11 1.04 | 89.45 0.38 | 90.30 1.23 | 93.13 4.06 |
| | ✓ | | | ✓ | 93.59 | 93.32 -0.27 | 93.32 -0.27 | 93.13 -0.46 | 93.24 -0.35 | 93.27 -0.33 | 93.43 -0.16 | 93.70 0.11 | 93.70 0.11 | 96.51 2.92 |
| | | ✓ | ✓ | | 89.07 | 89.86 0.79 | 90.16 1.09 | 87.35 -1.72 | 90.41 1.34 | 89.45 0.38 | 90.38 1.31 | 90.27 1.2 | 90.30 1.23 | 93.35 4.28 |
| | | ✓ | | ✓ | 93.59 | 93.79 0.19 | 93.76 0.16 | 93.00 -0.6 | 93.38 -0.22 | 93.00 -0.6 | 93.73 0.14 | 93.89 0.3 | 93.87 0.27 | 95.99 2.4 |
| | | | ✓ | ✓ | 93.59 | 93.98 0.38 | 93.95 0.35 | 93.92 0.33 | 94.06 0.46 | 93.84 0.25 | 93.98 0.38 | 93.73 0.14 | 93.92 0.33 | 96.18 2.59 |
| ✓ | ✓ | ✓ | | | 87.46 | 88.85 1.39 | 89.29 1.83 | 84.93 -2.53 | 89.72 2.26 | 88.42 0.95 | 89.45 1.99 | 89.48 2.02 | 89.83 2.37 | 94.69 7.22 |
| ✓ | ✓ | | ✓ | | 89.07 | 89.29 0.22 | 89.81 0.74 | 90.02 0.95 | 90.11 1.04 | 89.86 0.79 | 90.08 1.01 | 90.11 1.04 | 90.38 1.31 | 94.74 5.67 |
| ✓ | ✓ | | | ✓ | 93.59 | 91.58 -2.02 | 92.97 -0.63 | 93.43 -0.16 | 93.19 -0.41 | 93.30 -0.3 | 92.86 -0.74 | 93.70 0.11 | 94.06 0.46 | 97.36 3.76 |
| ✓ | | ✓ | ✓ | | 89.07 | 90.16 1.09 | 90.46 1.39 | 86.73 -2.34 | 90.62 1.55 | 89.15 0.08 | 90.46 1.39 | 90.35 1.28 | 90.79 1.72 | 94.79 5.72 |
| ✓ | | ✓ | | ✓ | 93.59 | 92.50 -1.09 | 93.08 -0.52 | 92.86 -0.74 | 93.02 -0.57 | 92.86 -0.74 | 93.19 -0.41 | 93.32 -0.27 | 94.25 0.65 | 97.03 3.43 |
| ✓ | | | ✓ | ✓ | 93.59 | 92.97 -0.63 | 93.70 0.11 | 93.98 0.38 | 93.87 0.27 | 93.84 0.25 | 93.84 0.25 | 94.19 0.6 | 94.47 0.87 | 96.97 3.38 |
| | ✓ | ✓ | ✓ | | 89.07 | 89.83 0.76 | 90.16 1.09 | 86.54 -2.53 | 90.65 1.58 | 89.83 0.76 | 90.41 1.34 | 90.35 1.28 | 90.71 1.64 | 95.12 6.05 |
| | ✓ | ✓ | | ✓ | 93.59 | 92.20 -1.39 | 93.02 -0.57 | 92.56 -1.04 | 92.75 -0.84 | 92.70 -0.9 | 92.94 -0.65 | 94.00 0.41 | 93.98 0.38 | 97.27 3.68 |
| | ✓ | | ✓ | ✓ | 93.59 | 92.91 -0.68 | 93.46 -0.14 | 93.46 -0.14 | 93.43 -0.16 | 93.49 -0.11 | 93.84 0.25 | 93.92 0.33 | 94.17 0.57 | 97.03 3.43 |
| | | ✓ | ✓ | ✓ | 93.59 | 92.97 -0.63 | 93.27 -0.33 | 92.89 -0.71 | 93.51 -0.08 | 93.13 -0.46 | 93.40 -0.19 | 94.03 0.44 | 94.17 0.57 | 97.03 3.43 |
| ✓ | ✓ | ✓ | ✓ | | 89.07 | 90.27 1.2 | 90.41 1.34 | 86.21 -2.86 | 90.57 1.5 | 89.48 0.41 | 90.54 1.47 | 90.35 1.28 | 90.90 1.83 | 95.88 6.81 |
| ✓ | ✓ | ✓ | | ✓ | 93.59 | 92.20 -1.39 | 92.80 -0.79 | 92.10 -1.5 | 92.91 -0.68 | 92.64 -0.95 | 92.80 -0.79 | 93.16 -0.44 | 94.17 0.57 | 97.77 4.17 |
| ✓ | ✓ | | ✓ | ✓ | 93.59 | 92.56 -1.04 | 92.94 -0.65 | 93.59 −0.0 | 93.19 -0.41 | 93.54 -0.05 | 92.94 -0.65 | 93.81 0.22 | 94.52 0.93 | 97.77 4.17 |
| ✓ | | ✓ | ✓ | ✓ | 93.59 | 92.91 -0.68 | 93.27 -0.33 | 92.78 -0.82 | 93.19 -0.41 | 93.05 -0.55 | 93.27 -0.33 | 93.79 0.19 | 94.33 0.74 | 97.52 3.92 |
| | ✓ | ✓ | ✓ | ✓ | 93.59 | 92.61 -0.98 | 93.00 -0.6 | 92.12 -1.47 | 92.94 -0.65 | 92.91 -0.68 | 93.05 -0.55 | 93.68 0.08 | 94.03 0.44 | 97.79 4.2 |
| ✓ | ✓ | ✓ | ✓ | ✓ | 93.59 | 91.63 -1.96 | 93.00 -0.59 | 92.26 -1.34 | 92.86 -0.74 | 92.89 -0.71 | 92.94 -0.65 | 93.30 -0.29 | 94.58 0.99 | 98.06 4.47 |
| | | | | | Mean Δ | -0.05 | 0.35 | -0.77 | 0.42 | 0.06 | 0.40 | 0.58 | 0.98 | 4.31 |
| | | | | | Max Δ | 1.39 | 1.83 | 1.34 | 2.26 | 1.09 | 1.99 | 2.02 | 2.37 | 7.22 |
| | | | | | Min Δ | -2.02 | -0.79 | -2.86 | -0.84 | -0.95 | -0.79 | -0.44 | 0.11 | 2.40 |

Table M.5: Our results on CARS dataset for all the possible combinations of combining the zero-shot predictions of CLIP backbones, which we group intro non-parametric and parametric techniques. Also, the best-performing single backbone (SINGLE-BEST) and the ORACLE performance. We present, for each combination of backbones, the improvement , constancy — and deterioration of accuracy performance for each method when we compare it against the SINGLE-BEST backbone. Mean, Max, and Min Δ summarize the difference in performance across methods and backbone combinations.

CARS

| ResNet | | ViT | | | SINGLE-BEST | Non-Parametric | | | | Parametric | | | | ORACLE |
|---|---|---|---|---|---|---|---|---|---|---|---|---|---|---|
| 50 | 101 | B-32 | B-16 | L-14 | | VOTE T-1 | VOTE T-3 | CONF | LOG-AVG | C-CONF | C-LOG-AVG | GAC | NLC | |
| ✓ | | | | | | | | | | | | 54.23 −0.00 | | |
| | ✓ | | | | | | | | | | | 61.12 −0.00 | | |
| | | ✓ | | | | | | | | | | 59.73 −0.00 | | |
| | | | ✓ | | | | | | | | | 64.61 −0.00 | | |
| | | | | ✓ | | | | | | | | 77.75 −0.00 | | |
| ✓ | ✓ | | | | 61.12 | 62.41 1.28 | 62.67 1.54 | 53.41 -7.71 | 63.03 1.9 | 61.81 0.68 | 62.83 1.7 | 63.52 2.4 | 63.59 2.46 | 71.58 10.46 |
| ✓ | | ✓ | | | 59.73 | 61.30 1.57 | 62.26 2.52 | 60.84 1.11 | 63.39 3.66 | 61.14 1.41 | 62.52 2.79 | 63.41 3.68 | 63.40 3.67 | 71.50 11.76 |
| ✓ | | | ✓ | | 64.61 | 65.27 0.66 | 65.55 0.95 | 53.94 -10.67 | 66.01 1.41 | 64.97 0.36 | 65.89 1.28 | 66.55 1.94 | 66.80 2.19 | 73.95 9.34 |
| ✓ | | | | ✓ | 77.75 | 77.58 -0.17 | 77.76 0.01 | 77.69 -0.06 | 77.24 -0.51 | 77.47 -0.29 | 77.64 -0.11 | 78.51 0.76 | 78.57 0.82 | 83.96 6.21 |
| | ✓ | ✓ | | | 61.12 | 64.42 3.3 | 64.99 3.87 | 63.41 2.29 | 66.89 5.77 | 63.71 2.59 | 65.66 4.54 | 66.86 5.73 | 66.83 5.71 | 73.60 12.47 |
| | ✓ | | ✓ | | 64.61 | 67.16 2.55 | 67.74 3.13 | 59.28 -5.32 | 67.99 3.38 | 66.66 2.05 | 68.11 3.51 | 68.03 3.42 | 68.20 3.59 | 76.27 11.67 |
| | ✓ | | | ✓ | 77.75 | 77.47 -0.29 | 77.78 0.02 | 77.42 -0.34 | 77.73 -0.02 | 77.44 -0.31 | 78.10 0.35 | 78.92 1.17 | 79.11 1.36 | 84.32 6.57 |
| | | ✓ | ✓ | | 64.61 | 66.34 1.73 | 66.84 2.24 | 65.41 0.81 | 67.95 3.35 | 65.87 1.27 | 67.26 2.65 | 68.00 3.4 | 67.77 3.16 | 75.48 10.87 |
| | | ✓ | | ✓ | 77.75 | 77.55 -0.2 | 77.86 0.11 | 77.55 -0.2 | 77.42 -0.34 | 77.52 -0.24 | 77.80 0.05 | 78.68 0.93 | 78.86 1.11 | 83.88 6.13 |
| | | | ✓ | ✓ | 77.75 | 77.54 -0.21 | 77.45 -0.3 | 77.32 -0.44 | 77.79 0.04 | 77.18 -0.57 | 77.63 -0.12 | 78.88 1.13 | 78.91 1.16 | 84.26 6.5 |
| ✓ | ✓ | ✓ | | | 61.12 | 64.63 3.51 | 65.23 4.1 | 57.36 -3.77 | 66.56 5.43 | 63.69 2.56 | 66.01 4.89 | 66.76 5.63 | 66.76 5.63 | 78.86 17.73 |
| ✓ | ✓ | | ✓ | | 64.61 | 66.34 1.73 | 67.35 2.75 | 59.66 -4.95 | 67.60 3.0 | 66.65 2.04 | 67.77 3.16 | 67.95 3.35 | 68.20 3.59 | 80.66 16.06 |
| ✓ | ✓ | | | ✓ | 77.75 | 73.80 -3.95 | 76.23 -1.52 | 76.10 -1.65 | 76.59 -1.16 | 77.25 -0.5 | 76.86 -0.9 | 79.06 1.31 | 78.88 1.13 | 87.49 9.74 |
| ✓ | | ✓ | ✓ | | 64.61 | 66.24 1.63 | 67.06 2.45 | 56.24 -8.37 | 67.94 3.33 | 66.07 1.47 | 67.62 3.01 | 68.46 3.86 | 68.29 3.68 | 80.24 15.63 |
| ✓ | | ✓ | | ✓ | 77.75 | 75.04 -2.71 | 77.17 -0.58 | 77.50 -0.25 | 76.71 -1.04 | 77.34 -0.41 | 77.24 -0.51 | 78.73 0.98 | 78.71 0.96 | 87.15 9.4 |
| ✓ | | | ✓ | ✓ | 77.75 | 74.53 -3.22 | 76.28 -1.47 | 75.90 -1.85 | 76.76 -0.99 | 76.97 -0.78 | 76.94 -0.81 | 78.40 0.65 | 79.01 1.26 | 87.43 9.68 |
| | ✓ | ✓ | ✓ | | 64.61 | 67.86 3.26 | 68.95 4.34 | 60.40 -4.2 | 69.36 4.75 | 67.04 2.44 | 69.12 4.51 | 69.16 4.55 | 69.34 4.74 | 81.37 16.76 |
| | ✓ | ✓ | | ✓ | 77.75 | 75.04 -2.71 | 76.96 -0.8 | 77.17 -0.58 | 76.88 -0.87 | 77.18 -0.57 | 77.14 -0.61 | 79.24 1.49 | 79.26 1.5 | 87.32 9.56 |
| | ✓ | | ✓ | ✓ | 77.75 | 75.60 -2.15 | 76.55 -1.21 | 75.70 -2.05 | 77.25 -0.5 | 77.15 -0.6 | 77.37 -0.39 | 79.02 1.27 | 79.06 1.31 | 87.63 9.87 |
| | | ✓ | ✓ | ✓ | 77.75 | 75.23 -2.52 | 76.71 -1.04 | 77.28 -0.47 | 77.35 -0.4 | 77.19 -0.56 | 77.15 -0.6 | 78.42 0.67 | 79.04 1.29 | 87.39 9.64 |
| ✓ | ✓ | ✓ | ✓ | | 64.61 | 67.72 3.11 | 68.54 3.93 | 60.59 -4.02 | 68.77 4.17 | 66.98 2.38 | 68.56 3.95 | 69.06 4.45 | 69.16 4.55 | 84.18 19.57 |
| ✓ | ✓ | ✓ | | ✓ | 77.75 | 73.81 -3.94 | 75.87 -1.88 | 75.97 -1.78 | 76.04 -1.72 | 77.08 -0.67 | 76.43 -1.32 | 78.56 0.81 | 79.12 1.37 | 89.37 11.62 |
| ✓ | ✓ | | ✓ | ✓ | 77.75 | 74.69 -3.06 | 76.06 -1.69 | 75.66 -2.09 | 76.41 -1.34 | 76.93 -0.82 | 76.47 -1.28 | 79.07 1.32 | 79.22 1.47 | 89.60 11.85 |
| ✓ | | ✓ | ✓ | ✓ | 77.75 | 74.51 -3.25 | 75.95 -1.8 | 75.90 -1.85 | 76.57 -1.18 | 77.01 -0.75 | 76.05 -1.7 | 78.83 1.08 | 79.19 1.44 | 89.30 11.55 |
| | ✓ | ✓ | ✓ | ✓ | 77.75 | 75.28 -2.47 | 76.48 -1.27 | 75.54 -2.21 | 76.61 -1.14 | 76.99 -0.76 | 76.91 -0.85 | 79.07 1.32 | 79.33 1.58 | 89.49 11.74 |
| ✓ | ✓ | ✓ | ✓ | ✓ | 77.75 | 73.67 -4.08 | 75.70 -2.05 | 75.56 -2.19 | 75.75 -2.0 | 76.78 -0.97 | 75.96 -1.79 | 78.96 1.21 | 80.30 2.55 | 90.85 13.1 |
| | | | Mean Δ | | | -0.41 | 0.63 | -2.42 | 1.04 | 0.40 | 0.98 | 2.25 | 2.43 | 11.36 |
| | | | Max Δ | | | 3.51 | 4.34 | 2.29 | 5.77 | 2.59 | 4.89 | 5.73 | 5.71 | 19.57 |
| | | | Min Δ | | | -4.08 | -2.05 | -10.67 | -2.00 | -0.97 | -1.79 | 0.65 | 0.82 | 6.13 |

Table M.6: Our results on CUB dataset for all the possible combinations of combining the zero-shot predictions of CLIP backbones, which we group intro non-parametric and parametric techniques. Also, the best-performing single backbone (SINGLE-BEST) and the ORACLE performance. We present, for each combination of backbones, the improvement , constancy — and deterioration of accuracy performance for each method when we compare it against the SINGLE-BEST backbone. Mean, Max, and Min Δ summarize the difference in performance across methods and backbone combinations.

**CUB**

| ResNet | | ViT | | | SINGLE-BEST | Non-Parametric | | | | Parametric | | | | ORACLE |
|---|---|---|---|---|---|---|---|---|---|---|---|---|---|---|
| 50 | 101 | B-32 | B-16 | L-14 | | VOTE T-1 | VOTE T-3 | CONF | LOG-AVG | C-CONF | C-LOG-AVG | GAC | NLC | |
| ✓ | | | | | | | | | | | | | | 46.57 −0.00 |
| | ✓ | | | | | | | | | | | | | 49.64 −0.00 |
| | | ✓ | | | | | | | | | | | | 52.99 −0.00 |
| | | | ✓ | | | | | | | | | | | 55.28 −0.00 |
| | | | | ✓ | | | | | | | | | | 62.06 −0.00 |
| ✓ | ✓ | | | | 49.64 | 51.28 1.64 | 52.33 2.69 | 45.41 -4.23 | 55.06 5.42 | 50.72 1.09 | 53.56 3.92 | 55.06 5.42 | 55.13 5.49 | 61.13 11.49 |
| ✓ | | ✓ | | | 52.99 | 54.92 1.93 | 55.70 2.71 | 53.94 0.95 | 57.77 4.78 | 53.90 0.91 | 56.51 3.52 | 57.68 4.69 | 57.75 4.76 | 64.03 11.05 |
| ✓ | | | ✓ | | 55.28 | 56.61 1.33 | 57.51 2.23 | 55.82 0.54 | 59.39 4.11 | 55.49 0.21 | 58.44 3.16 | 59.73 4.45 | 59.68 4.4 | 65.52 10.23 |
| ✓ | | | | ✓ | 62.06 | 61.65 -0.41 | 62.32 0.26 | 61.72 -0.35 | 63.96 1.9 | 61.01 -1.05 | 62.77 0.71 | 64.12 2.05 | 64.26 2.19 | 70.49 8.42 |
| | ✓ | ✓ | | | 52.99 | 54.94 1.95 | 55.49 2.5 | 48.72 -4.26 | 57.61 4.63 | 54.04 1.05 | 56.51 3.52 | 57.42 4.44 | 57.58 4.59 | 63.82 10.84 |
| | ✓ | | ✓ | | 55.28 | 56.56 1.28 | 57.16 1.88 | 56.25 0.97 | 58.78 3.5 | 56.27 0.98 | 58.08 2.8 | 58.80 3.52 | 59.25 3.97 | 65.34 10.06 |
| | ✓ | | | ✓ | 62.06 | 62.01 -0.05 | 62.70 0.64 | 62.03 -0.03 | 63.58 1.52 | 61.67 -0.4 | 63.24 1.17 | 63.69 1.62 | 63.93 1.86 | 70.31 8.25 |
| | | ✓ | ✓ | | 55.28 | 57.87 2.59 | 58.58 3.3 | 51.05 -4.23 | 60.87 5.59 | 57.40 2.12 | 59.63 4.35 | 60.72 5.44 | 60.74 5.45 | 67.10 11.82 |
| | | ✓ | | ✓ | 62.06 | 62.58 0.52 | 63.29 1.23 | 62.62 0.55 | 64.34 2.28 | 62.32 0.26 | 64.07 2.0 | 64.58 2.52 | 64.50 2.43 | 71.33 9.27 |
| | | | ✓ | ✓ | 62.06 | 62.63 0.57 | 63.36 1.29 | 54.87 -7.2 | 64.98 2.92 | 62.29 0.22 | 63.91 1.85 | 65.07 3.0 | 65.00 2.93 | 71.18 9.11 |
| ✓ | ✓ | ✓ | | | 52.99 | 56.25 3.26 | 57.21 4.23 | 47.93 -5.06 | 59.32 6.33 | 54.50 1.52 | 58.18 5.2 | 59.60 6.61 | 59.25 6.27 | 69.71 16.72 |
| ✓ | ✓ | | ✓ | | 55.28 | 57.34 2.05 | 58.42 3.14 | 53.31 -1.97 | 60.20 4.92 | 56.08 0.79 | 59.42 4.14 | 60.58 5.3 | 60.68 5.4 | 70.90 15.62 |
| ✓ | ✓ | | | ✓ | 62.06 | 61.25 -0.81 | 62.81 0.74 | 59.89 -2.17 | 64.01 1.95 | 60.80 -1.26 | 63.41 1.35 | 65.08 3.02 | 65.03 2.97 | 74.75 12.69 |
| ✓ | | ✓ | ✓ | | 55.28 | 58.94 3.66 | 60.11 4.83 | 51.86 -3.42 | 61.48 6.2 | 57.46 2.17 | 60.99 5.71 | 61.22 5.94 | 62.01 6.73 | 72.64 17.36 |
| ✓ | | ✓ | | ✓ | 62.06 | 61.96 -0.1 | 63.88 1.81 | 62.39 0.33 | 65.36 3.3 | 61.60 -0.47 | 64.36 2.3 | 65.67 3.61 | 65.64 3.57 | 75.85 13.79 |
| ✓ | | | ✓ | ✓ | 62.06 | 62.67 0.6 | 64.19 2.12 | 55.44 -6.63 | 65.67 3.61 | 61.72 -0.35 | 64.96 2.9 | 66.02 3.95 | 66.34 4.28 | 76.04 13.98 |
| | ✓ | ✓ | ✓ | | 55.28 | 58.97 3.69 | 59.92 4.64 | 49.83 -5.45 | 61.24 5.95 | 57.73 2.45 | 60.68 5.4 | 61.18 5.9 | 61.65 6.37 | 71.99 16.71 |
| | ✓ | ✓ | | ✓ | 62.06 | 62.15 0.09 | 63.44 1.38 | 60.25 -1.81 | 65.00 2.93 | 61.82 -0.24 | 63.98 1.92 | 65.36 3.3 | 65.31 3.24 | 75.42 13.36 |
| | ✓ | | ✓ | ✓ | 62.06 | 62.43 0.36 | 64.03 1.97 | 55.47 -6.59 | 65.33 3.26 | 62.12 0.05 | 64.91 2.85 | 66.24 4.18 | 66.05 3.99 | 75.54 13.48 |
| | | ✓ | ✓ | ✓ | 62.06 | 63.12 1.05 | 64.46 2.4 | 54.45 -7.61 | 65.93 3.87 | 62.77 0.71 | 64.79 2.73 | 66.05 3.99 | 66.05 3.99 | 76.60 14.53 |
| ✓ | ✓ | ✓ | ✓ | | 55.28 | 59.44 4.16 | 60.61 5.33 | 49.57 -5.71 | 61.81 6.52 | 57.49 2.21 | 61.39 6.11 | 61.96 6.68 | 62.50 7.21 | 75.60 20.31 |
| ✓ | ✓ | ✓ | | ✓ | 62.06 | 62.00 -0.07 | 63.62 1.55 | 59.37 -2.69 | 64.96 2.9 | 61.22 -0.85 | 64.10 2.04 | 65.38 3.31 | 65.91 3.85 | 78.32 16.26 |
| ✓ | ✓ | | ✓ | ✓ | 62.06 | 62.98 0.91 | 64.41 2.35 | 54.45 -7.61 | 65.22 3.16 | 61.49 -0.57 | 64.65 2.59 | 66.66 4.59 | 66.59 4.52 | 78.58 16.52 |
| ✓ | | ✓ | ✓ | ✓ | 62.06 | 63.63 1.57 | 65.15 3.09 | 54.73 -7.34 | 65.96 3.9 | 62.20 0.14 | 65.59 3.52 | 66.21 4.14 | 66.90 4.83 | 79.67 17.6 |
| | ✓ | ✓ | ✓ | ✓ | 62.06 | 63.19 1.12 | 64.38 2.31 | 54.18 -7.89 | 65.69 3.62 | 62.29 0.22 | 65.21 3.14 | 66.28 4.21 | 66.57 4.5 | 79.01 16.95 |
| ✓ | ✓ | ✓ | ✓ | ✓ | 62.06 | 63.32 1.26 | 64.38 2.31 | 53.71 -8.35 | 65.53 3.47 | 61.79 -0.28 | 65.01 2.95 | 66.29 4.23 | 68.40 6.34 | 81.20 19.14 |
| | | Mean Δ | | | | 1.31 | 2.42 | -3.74 | 3.94 | 0.45 | 3.15 | 4.24 | 4.47 | 13.68 |
| | | Max Δ | | | | 4.16 | 5.33 | 0.97 | 6.52 | 2.45 | 6.11 | 6.68 | 7.21 | 20.31 |
| | | Min Δ | | | | -0.81 | 0.26 | -8.35 | 1.52 | -1.26 | 0.71 | 1.62 | 1.86 | 8.25 |

Table M.7: Our results on DTD dataset for all the possible combinations of combining the zero-shot predictions of CLIP backbones, which we group intro non-parametric and parametric techniques. Also, the best-performing single backbone (SINGLE-BEST) and the ORACLE performance. We present, for each combination of backbones, the improvement , constancy — and deterioration of accuracy performance for each method when we compare it against the SINGLE-BEST backbone. Mean, Max, and Min Δ summarize the difference in performance across methods and backbone combinations.

**DTD**

| ResNet | | ViT | | | SINGLE-BEST | Non-Parametric | | | | Parametric | | | | ORACLE |
|---|---|---|---|---|---|---|---|---|---|---|---|---|---|---|
| 50 | 101 | B-32 | B-16 | L-14 | | VOTE T-1 | VOTE T-3 | CONF | LOG-AVG | C-CONF | C-LOG-AVG | GAC | NLC | |
| ✓ | | | | | | | | | | | 41.22 –0.00 | | | |
| | ✓ | | | | | | | | | | 43.67 –0.00 | | | |
| | | ✓ | | | | | | | | | 43.99 –0.00 | | | |
| | | | ✓ | | | | | | | | 45.11 –0.00 | | | |
| | | | | ✓ | | | | | | | 55.32 –0.00 | | | |
| ✓ | ✓ | | | | 43.67 | 47.13 3.46 | 47.29 3.62 | 46.44 2.77 | 47.71 4.04 | 45.96 2.29 | 47.34 3.67 | 47.61 3.94 | 47.71 4.04 | 54.36 10.69 |
| ✓ | | ✓ | | | 43.99 | 44.63 0.64 | 45.00 1.01 | 41.22 -2.77 | 46.01 2.02 | 43.78 -0.21 | 45.43 1.44 | 45.74 1.76 | 45.90 1.91 | 51.70 7.71 |
| ✓ | | | ✓ | | 45.11 | 46.86 1.76 | 47.13 2.02 | 40.37 -4.73 | 47.39 2.29 | 45.85 0.74 | 47.23 2.13 | 47.87 2.77 | 47.39 2.29 | 53.24 8.14 |
| ✓ | | | | ✓ | 55.32 | 55.64 0.32 | 56.12 0.8 | 41.33 -13.99 | 55.64 0.32 | 55.05 -0.27 | 56.22 0.9 | 55.43 0.11 | 56.44 1.12 | 62.23 6.91 |
| | ✓ | ✓ | | | 43.99 | 47.02 3.03 | 47.34 3.35 | 41.60 -2.39 | 48.46 4.47 | 47.13 3.14 | 47.93 3.94 | 47.87 3.88 | 47.98 3.99 | 55.11 11.12 |
| | ✓ | | ✓ | | 45.11 | 48.46 3.35 | 48.94 3.83 | 48.30 3.19 | 49.15 4.04 | 48.67 3.56 | 48.78 3.67 | 48.78 3.67 | 48.78 3.67 | 55.74 10.64 |
| | ✓ | | | ✓ | 55.32 | 55.21 -0.11 | 55.59 0.27 | 43.35 -11.97 | 55.69 0.37 | 55.11 -0.21 | 56.33 1.01 | 56.76 1.44 | 56.70 1.38 | 61.60 6.28 |
| | | ✓ | ✓ | | 45.11 | 46.65 1.54 | 46.86 1.76 | 43.24 -1.86 | 47.71 2.61 | 45.48 0.37 | 47.23 2.13 | 47.07 1.97 | 47.61 2.5 | 53.56 8.46 |
| | | ✓ | | ✓ | 55.32 | 55.74 0.43 | 56.33 1.01 | 43.56 -11.76 | 56.22 0.9 | 55.32 –0.0 | 56.22 0.9 | 56.49 1.17 | 56.70 1.38 | 62.02 6.7 |
| | | | ✓ | ✓ | 55.32 | 57.02 1.7 | 57.39 2.07 | 43.88 -11.44 | 56.28 0.96 | 56.60 1.28 | 56.76 1.44 | 56.81 1.49 | 56.86 1.54 | 62.02 6.7 |
| ✓ | ✓ | ✓ | | | 43.99 | 47.50 3.51 | 47.71 3.72 | 40.37 -3.62 | 48.99 5.0 | 46.97 2.98 | 47.82 3.83 | 49.26 5.27 | 49.10 5.11 | 59.26 15.27 |
| ✓ | ✓ | | ✓ | | 45.11 | 48.35 3.24 | 49.41 4.31 | 44.68 -0.43 | 49.95 4.84 | 48.30 3.19 | 49.95 4.84 | 50.32 5.21 | 50.21 5.11 | 60.59 15.48 |
| ✓ | ✓ | | | ✓ | 55.32 | 54.26 -1.06 | 56.22 0.9 | 41.70 -13.62 | 55.32 –0.0 | 54.89 -0.43 | 56.76 1.44 | 57.02 1.7 | 57.18 1.86 | 65.80 10.48 |
| ✓ | | ✓ | ✓ | | 45.11 | 46.86 1.76 | 46.97 1.86 | 44.95 -0.16 | 48.19 3.09 | 45.53 0.43 | 47.34 2.23 | 47.77 2.66 | 48.24 3.14 | 57.87 12.77 |
| ✓ | | ✓ | | ✓ | 55.32 | 53.19 -2.13 | 55.43 0.11 | 41.86 -13.46 | 54.73 -0.59 | 55.11 -0.21 | 54.73 -0.59 | 56.06 0.74 | 56.49 1.17 | 65.59 10.27 |
| ✓ | | | ✓ | ✓ | 55.32 | 53.99 -1.33 | 56.33 1.01 | 42.87 -12.45 | 56.01 0.69 | 55.80 0.48 | 56.60 1.28 | 57.34 2.02 | 57.18 1.86 | 65.69 10.37 |
| | ✓ | ✓ | ✓ | | 45.11 | 48.40 3.3 | 49.68 4.57 | 41.76 -3.35 | 49.26 4.15 | 48.62 3.51 | 49.68 4.57 | 49.95 4.84 | 49.20 4.1 | 60.32 15.21 |
| | ✓ | ✓ | | ✓ | 55.32 | 54.47 -0.85 | 55.69 0.37 | 44.57 -10.74 | 55.27 -0.05 | 55.32 –0.0 | 56.01 0.69 | 56.86 1.54 | 57.18 1.86 | 65.32 10.0 |
| | ✓ | | ✓ | ✓ | 55.32 | 54.89 -0.43 | 56.76 1.44 | 43.56 -11.76 | 55.74 0.43 | 55.74 0.43 | 56.81 1.49 | 56.70 1.38 | 57.77 2.45 | 65.37 10.05 |
| | | ✓ | ✓ | ✓ | 55.32 | 54.63 -0.69 | 56.44 1.12 | 44.47 -10.85 | 55.74 0.43 | 55.74 0.43 | 56.49 1.17 | 57.23 1.91 | 57.07 1.76 | 65.32 10.0 |
| ✓ | ✓ | ✓ | ✓ | | 45.11 | 48.67 3.56 | 49.52 4.41 | 43.30 -1.81 | 49.63 4.52 | 48.46 3.35 | 49.79 4.68 | 50.27 5.16 | 49.41 4.31 | 63.19 18.09 |
| ✓ | ✓ | ✓ | | ✓ | 55.32 | 54.36 -0.96 | 55.16 -0.16 | 42.50 -12.82 | 54.73 -0.59 | 55.00 -0.32 | 54.95 -0.37 | 56.22 0.9 | 57.34 2.02 | 68.03 12.71 |
| ✓ | ✓ | | ✓ | ✓ | 55.32 | 54.89 -0.43 | 56.38 1.06 | 42.93 -12.39 | 55.00 -0.32 | 55.53 0.21 | 56.81 1.49 | 57.87 2.55 | 57.71 2.39 | 68.14 12.82 |
| ✓ | | ✓ | ✓ | ✓ | 55.32 | 53.72 -1.6 | 55.96 0.64 | 42.50 -12.82 | 54.57 -0.74 | 55.48 0.16 | 55.48 0.16 | 56.38 1.06 | 56.97 1.65 | 67.71 12.39 |
| | ✓ | ✓ | ✓ | ✓ | 55.32 | 54.68 -0.64 | 56.12 0.8 | 44.47 -10.85 | 55.37 0.05 | 55.96 0.64 | 56.06 0.74 | 56.81 1.49 | 57.50 2.18 | 67.66 12.34 |
| ✓ | ✓ | ✓ | ✓ | ✓ | 55.32 | 53.72 -1.6 | 55.69 0.37 | 42.55 -12.77 | 54.36 -0.96 | 55.59 0.27 | 55.21 -0.11 | 56.12 0.8 | 58.94 3.62 | 69.63 14.31 |
| | | Mean Δ | | | | 0.76 | 1.78 | -7.65 | 1.61 | 1.00 | 1.87 | 2.36 | 2.63 | 11.00 |
| | | Max Δ | | | | 3.56 | 4.57 | 3.19 | 5.00 | 3.51 | 4.84 | 5.27 | 5.11 | 18.09 |
| | | Min Δ | | | | -2.13 | -0.16 | -13.99 | -0.96 | -0.43 | -0.59 | 0.11 | 1.12 | 6.28 |

Table M.8: Our results on FGVC dataset for all the possible combinations of combining the zero-shot predictions of CLIP backbones, which we group intro non-parametric and parametric techniques. Also, the best-performing single backbone (SINGLE-BEST) and the ORACLE performance. We present, for each combination of backbones, the improvement , constancy — and deterioration of accuracy performance for each method when we compare it against the SINGLE-BEST backbone. Mean, Max, and Min Δ summarize the difference in performance across methods and backbone combinations.

FGVC

| ResNet | | ViT | | | SINGLE-BEST | Non-Parametric | | | | Parametric | | | | ORACLE |
|---|---|---|---|---|---|---|---|---|---|---|---|---|---|---|
| 50 | 101 | B-32 | B-16 | L-14 | | VOTE T-1 | VOTE T-3 | CONF | LOG-AVG | C-CONF | C-LOG-AVG | GAC | NLC | |
| ✓ | | | | | | | | | | | | 17.07 —0.00 | | |
| | ✓ | | | | | | | | | | | 18.63 —0.00 | | |
| | | ✓ | | | | | | | | | | 19.65 —0.00 | | |
| | | | ✓ | | | | | | | | | 24.39 —0.00 | | |
| | | | | ✓ | | | | | | | | 31.71 —0.00 | | |
| ✓ | ✓ | | | | 18.63 | 18.87 0.24 | 19.02 0.39 | 18.60 -0.03 | 19.23 0.6 | 18.48 -0.15 | 19.05 0.42 | 19.59 0.96 | 19.89 1.26 | 26.19 7.56 |
| ✓ | | ✓ | | | 19.65 | 20.52 0.87 | 20.46 0.81 | 16.62 -3.03 | 21.36 1.71 | 19.80 0.15 | 21.18 1.53 | 21.00 1.35 | 22.17 2.52 | 27.69 8.04 |
| ✓ | | | ✓ | | 24.39 | 24.24 -0.15 | 24.33 -0.06 | 17.19 -7.2 | 24.78 0.39 | 23.97 -0.42 | 25.23 0.84 | 25.65 1.26 | 25.89 1.5 | 31.02 6.63 |
| ✓ | | | | ✓ | 31.71 | 31.47 -0.24 | 31.47 -0.24 | 31.47 -0.24 | 31.68 -0.03 | 31.59 -0.12 | 32.55 0.84 | 32.49 0.78 | 33.57 1.86 | 37.59 5.88 |
| | ✓ | ✓ | | | 19.65 | 21.00 1.35 | 21.51 1.86 | 20.70 1.05 | 21.69 2.04 | 20.73 1.08 | 22.02 2.37 | 21.21 1.56 | 22.41 2.76 | 28.20 8.55 |
| | ✓ | | ✓ | | 24.39 | 23.67 -0.72 | 23.97 -0.42 | 23.52 -0.87 | 23.82 -0.57 | 23.79 -0.6 | 24.36 -0.03 | 24.81 0.42 | 25.38 0.99 | 31.92 7.53 |
| | ✓ | | | ✓ | 31.71 | 30.81 -0.9 | 31.29 -0.42 | 30.93 -0.78 | 30.54 -1.17 | 31.20 -0.51 | 31.50 -0.21 | 31.35 -0.36 | 33.03 1.32 | 38.31 6.6 |
| | | ✓ | ✓ | | 24.39 | 24.18 -0.21 | 24.30 -0.09 | 20.22 -4.17 | 25.08 0.69 | 23.76 -0.63 | 24.84 0.45 | 25.02 0.63 | 25.95 1.56 | 32.73 8.34 |
| | | ✓ | | ✓ | 31.71 | 31.86 0.15 | 31.68 -0.03 | 31.59 -0.12 | 32.04 0.33 | 31.71 —0.0 | 32.79 1.08 | 32.61 0.9 | 33.33 1.62 | 39.57 7.86 |
| | | | ✓ | ✓ | 31.71 | 31.62 -0.09 | 32.19 0.48 | 31.95 0.24 | 32.76 1.05 | 31.92 0.21 | 32.88 1.17 | 33.30 1.59 | 33.69 1.98 | 41.25 9.54 |
| ✓ | ✓ | ✓ | | | 19.65 | 20.79 1.14 | 21.51 1.86 | 18.12 -1.53 | 21.75 2.1 | 20.52 0.87 | 21.69 2.04 | 21.72 2.07 | 23.49 3.84 | 33.60 13.95 |
| ✓ | ✓ | | ✓ | | 24.39 | 22.89 -1.5 | 23.28 -1.11 | 17.55 -6.84 | 23.67 -0.72 | 23.40 -0.99 | 24.03 -0.36 | 24.69 0.3 | 26.61 2.22 | 36.57 12.18 |
| ✓ | ✓ | | | ✓ | 31.71 | 28.80 -2.91 | 30.00 -1.71 | 30.87 -0.84 | 30.21 -1.5 | 31.14 -0.57 | 31.53 -0.18 | 32.58 0.87 | 34.08 2.37 | 42.36 10.65 |
| ✓ | | ✓ | ✓ | | 24.39 | 23.94 -0.45 | 24.57 0.18 | 22.53 -1.86 | 24.81 0.42 | 23.61 -0.78 | 24.99 0.6 | 25.71 1.32 | 26.76 2.37 | 37.68 13.29 |
| ✓ | | ✓ | | ✓ | 31.71 | 30.36 -1.35 | 31.29 -0.42 | 30.93 -0.78 | 32.07 0.36 | 31.59 -0.12 | 32.73 1.02 | 32.25 0.54 | 34.74 3.03 | 43.53 11.82 |
| ✓ | | | ✓ | ✓ | 31.71 | 30.63 -1.08 | 31.77 0.06 | 30.27 -1.44 | 32.40 0.69 | 31.80 0.09 | 32.97 1.26 | 33.06 1.35 | 34.80 3.09 | 45.06 13.35 |
| | ✓ | ✓ | ✓ | | 24.39 | 23.79 -0.6 | 24.15 -0.24 | 20.55 -3.84 | 25.02 0.63 | 23.40 -0.99 | 24.72 0.33 | 25.50 1.11 | 26.31 1.92 | 38.10 13.71 |
| | ✓ | ✓ | | ✓ | 31.71 | 30.24 -1.47 | 30.90 -0.81 | 30.90 -0.81 | 31.62 -0.09 | 31.20 -0.51 | 32.07 0.36 | 31.95 0.24 | 33.72 2.01 | 44.01 12.3 |
| | ✓ | | ✓ | ✓ | 31.71 | 30.75 -0.96 | 31.62 -0.09 | 31.23 -0.48 | 31.77 0.06 | 31.44 -0.27 | 32.40 0.69 | 33.21 1.5 | 34.23 2.52 | 45.69 13.98 |
| | | ✓ | ✓ | ✓ | 31.71 | 31.14 -0.57 | 31.59 -0.12 | 31.02 -0.69 | 32.91 1.2 | 31.77 0.06 | 33.39 1.68 | 33.18 1.47 | 34.50 2.79 | 46.56 14.85 |
| ✓ | ✓ | ✓ | ✓ | | 24.39 | 23.97 -0.42 | 23.91 -0.48 | 22.11 -2.28 | 24.72 0.33 | 23.31 -1.08 | 24.60 0.21 | 25.50 1.11 | 27.45 3.06 | 41.79 17.4 |
| ✓ | ✓ | ✓ | | ✓ | 31.71 | 28.65 -3.06 | 30.63 -1.08 | 30.45 -1.26 | 31.20 -0.51 | 31.14 -0.57 | 32.04 0.33 | 32.37 0.66 | 34.80 3.09 | 47.04 15.33 |
| ✓ | ✓ | | ✓ | ✓ | 31.71 | 29.46 -2.25 | 30.72 -0.99 | 29.85 -1.86 | 31.02 -0.69 | 31.38 -0.33 | 32.31 0.6 | 32.55 0.84 | 34.92 3.21 | 48.54 16.83 |
| ✓ | | ✓ | ✓ | ✓ | 31.71 | 30.06 -1.65 | 31.44 -0.27 | 31.02 -0.69 | 32.28 0.57 | 31.65 -0.06 | 33.21 1.5 | 33.63 1.92 | 35.37 3.66 | 49.38 17.67 |
| | ✓ | ✓ | ✓ | ✓ | 31.71 | 29.82 -1.89 | 31.20 -0.51 | 30.51 -1.2 | 32.58 0.87 | 31.35 -0.36 | 32.25 0.54 | 33.33 1.62 | 34.23 2.52 | 49.83 18.12 |
| ✓ | ✓ | ✓ | ✓ | ✓ | 31.71 | 28.95 -2.76 | 30.96 -0.75 | 30.63 -1.08 | 31.53 -0.18 | 31.29 -0.42 | 31.89 0.18 | 33.18 1.47 | 35.88 4.17 | 52.09 20.37 |
| | | Mean Δ | | | | -0.83 | -0.16 | -1.64 | 0.33 | -0.27 | 0.74 | 1.06 | 2.43 | 12.01 |
| | | Max Δ | | | | 1.35 | 1.86 | 1.05 | 2.10 | 1.08 | 2.37 | 2.07 | 4.17 | 20.37 |
| | | Min Δ | | | | -3.06 | -1.71 | -7.20 | -1.50 | -1.08 | -0.36 | -0.36 | 0.99 | 5.88 |

Table M.9: Our results on FOOD dataset for all the possible combinations of combining the zero-shot predictions of CLIP backbones, which we group intro non-parametric and parametric techniques. Also, the best-performing single backbone (SINGLE-BEST) and the ORACLE performance. We present, for each combination of backbones, the improvement , constancy — and deterioration of accuracy performance for each method when we compare it against the SINGLE-BEST backbone. Mean, Max, and Min Δ summarize the difference in performance across methods and backbone combinations.

**FOOD**

| ResNet | | ViT | | | SINGLE-BEST | Non-Parametric | | | | Parametric | | | | ORACLE |
|---|---|---|---|---|---|---|---|---|---|---|---|---|---|---|
| 50 | 101 | B-32 | B-16 | L-14 | | VOTE T-1 | VOTE T-3 | CONF | LOG-AVG | C-CONF | C-LOG-AVG | GAC | NLC | |
| ✓ | | | | | | | | | | 77.91 —0.00 | | | | |
| | ✓ | | | | | | | | | 81.86 —0.00 | | | | |
| | | ✓ | | | | | | | | 82.58 —0.00 | | | | |
| | | | ✓ | | | | | | | 87.91 —0.00 | | | | |
| | | | | ✓ | | | | | | 92.32 —0.00 | | | | |
| ✓ | ✓ | | | | 81.86 | 82.63 0.78 | 82.90 1.04 | 82.26 0.4 | 83.08 1.22 | 82.47 0.61 | 83.14 1.29 | 83.28 1.43 | 83.27 1.41 | 87.16 5.3 |
| ✓ | | ✓ | | | 82.58 | 83.83 1.25 | 84.26 1.68 | 76.96 -5.62 | 84.55 1.97 | 83.60 1.02 | 84.40 1.82 | 84.41 1.83 | 84.68 2.1 | 88.42 5.84 |
| ✓ | | | ✓ | | 87.91 | 87.56 -0.36 | 87.62 -0.3 | 87.49 -0.43 | 87.58 -0.33 | 87.52 -0.4 | 87.80 -0.11 | 88.13 0.22 | 88.55 0.64 | 91.31 3.39 |
| ✓ | | | | ✓ | 92.32 | 91.94 -0.38 | 92.04 -0.29 | 91.88 -0.44 | 91.75 -0.57 | 91.75 -0.57 | 91.93 -0.39 | 92.82 0.49 | 92.79 0.47 | 94.52 2.19 |
| | ✓ | ✓ | | | 82.58 | 84.97 2.39 | 85.31 2.73 | 84.48 1.9 | 85.70 3.11 | 84.62 2.04 | 85.54 2.96 | 85.55 2.97 | 85.69 3.11 | 89.28 6.69 |
| | ✓ | | ✓ | | 87.91 | 88.17 0.26 | 88.30 0.38 | 87.92 0.01 | 88.40 0.49 | 87.87 -0.05 | 88.34 0.43 | 88.72 0.81 | 88.68 0.76 | 91.81 3.9 |
| | ✓ | | | ✓ | 92.32 | 92.25 -0.07 | 92.34 0.02 | 92.04 -0.29 | 92.08 -0.24 | 91.85 -0.48 | 92.22 -0.11 | 92.78 0.45 | 92.80 0.48 | 94.81 2.49 |
| | | ✓ | ✓ | | 87.91 | 88.08 0.16 | 88.18 0.27 | 82.55 -5.36 | 88.25 0.33 | 88.10 0.19 | 88.31 0.4 | 88.79 0.88 | 88.74 0.83 | 91.54 3.63 |
| | | ✓ | | ✓ | 92.32 | 92.13 -0.2 | 92.21 -0.12 | 92.00 -0.33 | 92.04 -0.28 | 91.92 -0.4 | 92.14 -0.19 | 92.70 0.37 | 92.73 0.41 | 94.76 2.44 |
| | | | ✓ | ✓ | 92.32 | 92.58 0.25 | 92.67 0.34 | 92.45 0.12 | 92.57 0.24 | 92.31 -0.01 | 92.59 0.27 | 92.70 0.38 | 92.88 0.55 | 94.96 2.64 |
| ✓ | ✓ | ✓ | | | 82.58 | 84.54 1.96 | 85.28 2.7 | 79.64 -2.95 | 85.59 3.01 | 84.62 2.04 | 85.38 2.8 | 85.82 3.24 | 85.84 3.26 | 91.16 8.58 |
| ✓ | ✓ | | ✓ | | 87.91 | 86.54 -1.37 | 87.34 -0.57 | 87.57 -0.34 | 87.72 -0.19 | 87.51 -0.4 | 87.63 -0.28 | 88.62 0.71 | 88.73 0.82 | 92.99 5.07 |
| ✓ | ✓ | | | ✓ | 92.32 | 88.86 -3.47 | 90.41 -1.91 | 91.66 -0.67 | 90.99 -1.34 | 91.45 -0.87 | 90.59 -1.74 | 92.79 0.47 | 92.84 0.52 | 95.47 3.15 |
| ✓ | | ✓ | ✓ | | 87.91 | 87.45 -0.47 | 87.98 0.07 | 80.07 -7.85 | 87.97 0.06 | 87.77 -0.15 | 88.09 0.17 | 88.84 0.93 | 88.85 0.94 | 93.01 5.09 |
| ✓ | | ✓ | | ✓ | 92.32 | 89.70 -2.63 | 90.90 -1.43 | 90.01 -2.32 | 91.18 -1.15 | 91.54 -0.79 | 90.93 -1.39 | 92.71 0.38 | 92.83 0.5 | 95.56 3.23 |
| ✓ | | | ✓ | ✓ | 92.32 | 91.38 -0.94 | 92.00 -0.33 | 92.11 -0.21 | 91.98 -0.34 | 91.93 -0.4 | 92.00 -0.33 | 92.77 0.44 | 93.02 0.69 | 95.77 3.45 |
| | ✓ | ✓ | ✓ | | 87.91 | 87.78 -0.13 | 88.18 0.27 | 83.45 -4.46 | 88.40 0.48 | 87.96 0.05 | 88.32 0.4 | 89.02 1.1 | 88.95 1.04 | 93.36 5.45 |
| | ✓ | ✓ | | ✓ | 92.32 | 90.17 -2.15 | 91.20 -1.13 | 91.75 -0.57 | 91.50 -0.82 | 91.54 -0.78 | 91.28 -1.05 | 92.90 0.57 | 92.93 0.61 | 95.71 3.39 |
| | ✓ | | ✓ | ✓ | 92.32 | 91.54 -0.78 | 91.98 -0.34 | 92.06 -0.26 | 92.07 -0.25 | 91.87 -0.45 | 92.02 -0.3 | 92.93 0.6 | 92.97 0.65 | 95.89 3.56 |
| | | ✓ | ✓ | ✓ | 92.32 | 91.24 -1.08 | 91.85 -0.48 | 90.73 -1.59 | 92.02 -0.31 | 92.04 -0.28 | 91.92 -0.4 | 92.93 0.61 | 92.98 0.66 | 95.79 3.47 |
| ✓ | ✓ | ✓ | ✓ | | 87.91 | 87.65 -0.26 | 87.98 0.07 | 81.00 -6.91 | 88.11 0.2 | 87.66 -0.25 | 88.10 0.19 | 89.07 1.16 | 89.03 1.12 | 94.01 6.1 |
| ✓ | ✓ | ✓ | | ✓ | 92.32 | 89.57 -2.75 | 90.21 -2.12 | 90.17 -2.15 | 90.61 -1.71 | 91.24 -1.09 | 90.24 -2.09 | 92.49 0.17 | 92.90 0.58 | 96.10 3.77 |
| ✓ | ✓ | | ✓ | ✓ | 92.32 | 91.10 -1.22 | 91.29 -1.03 | 91.83 -0.5 | 91.38 -0.95 | 91.60 -0.73 | 91.26 -1.07 | 92.93 0.61 | 93.03 0.7 | 96.26 3.94 |
| ✓ | | ✓ | ✓ | ✓ | 92.32 | 90.99 -1.33 | 91.49 -0.84 | 89.53 -2.79 | 91.53 -0.8 | 91.73 -0.59 | 91.49 -0.84 | 92.93 0.6 | 93.06 0.74 | 96.25 3.93 |
| | ✓ | ✓ | ✓ | ✓ | 92.32 | 91.26 -1.06 | 91.56 -0.76 | 90.61 -1.72 | 91.62 -0.7 | 91.67 -0.65 | 91.45 -0.88 | 92.86 0.54 | 93.03 0.71 | 96.35 4.02 |
| ✓ | ✓ | ✓ | ✓ | ✓ | 92.32 | 90.08 -2.25 | 90.88 -1.45 | 89.62 -2.7 | 91.09 -1.24 | 91.43 -0.9 | 90.91 -1.41 | 92.91 0.59 | 93.07 0.75 | 96.60 4.27 |
| | | Mean Δ | | | | -0.61 | -0.14 | -1.85 | 0.00 | -0.17 | -0.07 | 0.87 | 0.96 | 4.19 |
| | | Max Δ | | | | 2.39 | 2.73 | 1.90 | 3.11 | 2.04 | 2.96 | 3.24 | 3.26 | 8.58 |
| | | Min Δ | | | | -3.47 | -2.12 | -7.85 | -1.71 | -1.09 | -2.09 | 0.17 | 0.41 | 2.19 |

Table M.10: Our results on FLOWERS dataset for all the possible combinations of combining the zero-shot predictions of CLIP backbones, which we group intro non-parametric and parametric techniques. Also, the best-performing single backbone (SINGLE-BEST) and the ORACLE performance. We present, for each combination of backbones, the improvement , constancy — and deterioration of accuracy performance for each method when we compare it against the SINGLE-BEST backbone. Mean, Max, and Min Δ summarize the difference in performance across methods and backbone combinations.

| ResNet | | ViT | | | SINGLE-BEST | Non-Parametric | | | | Parametric | | | | ORACLE |
|---|---|---|---|---|---|---|---|---|---|---|---|---|---|---|
| 50 | 101 | B-32 | B-16 | L-14 | | VOTE T-1 | VOTE T-3 | CONF | LOG-AVG | C-CONF | C-LOG-AVG | GAC | NLC | |
| ✓ | | | | | | | | | | 66.12 −0.00 | | | | |
| | ✓ | | | | | | | | | 65.20 −0.00 | | | | |
| | | ✓ | | | | | | | | 66.48 −0.00 | | | | |
| | | | ✓ | | | | | | | 71.43 −0.00 | | | | |
| | | | | ✓ | | | | | | 79.05 −0.00 | | | | |
| ✓ | ✓ | | | | 66.12 | 68.01 1.89 | 67.95 1.82 | 67.36 1.24 | 68.63 2.5 | 67.25 1.12 | 68.43 2.31 | 68.40 2.28 | 68.84 2.72 | 73.83 7.71 |
| ✓ | | ✓ | | | 66.48 | 69.21 2.73 | 69.77 3.29 | 68.68 2.2 | 69.17 2.68 | 68.76 2.28 | 69.77 3.29 | 69.28 2.8 | 70.00 3.51 | 75.49 9.01 |
| ✓ | | | ✓ | | 71.43 | 72.45 1.02 | 72.68 1.25 | 72.03 0.6 | 72.48 1.06 | 71.87 0.44 | 72.69 1.27 | 72.42 0.99 | 73.12 1.69 | 76.63 5.2 |
| ✓ | | | | ✓ | 79.05 | 78.29 -0.76 | 78.22 -0.83 | 79.02 -0.03 | 77.33 -1.72 | 78.78 -0.28 | 78.19 -0.86 | 78.91 -0.15 | 79.41 0.36 | 83.27 4.21 |
| | ✓ | ✓ | | | 66.48 | 68.56 2.08 | 68.74 2.26 | 68.35 1.87 | 68.87 2.39 | 68.32 1.84 | 68.71 2.23 | 68.74 2.26 | 69.21 2.73 | 74.39 7.9 |
| | ✓ | | ✓ | | 71.43 | 71.98 0.55 | 72.32 0.89 | 71.70 0.28 | 72.35 0.93 | 71.41 -0.02 | 72.37 0.94 | 73.10 1.68 | 73.02 1.59 | 77.35 5.92 |
| | ✓ | | | ✓ | 79.05 | 77.87 -1.19 | 77.74 -1.32 | 78.34 -0.72 | 77.23 -1.82 | 77.87 -1.19 | 77.52 -1.53 | 77.83 -1.22 | 79.23 0.18 | 82.62 3.56 |
| | | ✓ | ✓ | | 71.43 | 72.91 1.48 | 72.69 1.27 | 65.15 -6.28 | 72.16 0.73 | 72.52 1.09 | 72.69 1.27 | 71.88 0.46 | 72.99 1.56 | 78.29 6.86 |
| | | ✓ | | ✓ | 79.05 | 78.50 -0.55 | 78.31 -0.75 | 78.45 -0.6 | 77.43 -1.63 | 78.35 -0.7 | 77.88 -1.17 | 78.94 -0.11 | 79.07 0.02 | 82.86 3.81 |
| | | | ✓ | ✓ | 79.05 | 78.01 -1.04 | 78.01 -1.04 | 78.45 -0.6 | 77.90 -1.15 | 78.37 -0.68 | 78.29 -0.76 | 78.14 -0.91 | 79.66 0.6 | 83.36 4.31 |
| ✓ | ✓ | ✓ | | | 66.48 | 69.38 2.89 | 69.98 3.5 | 69.12 2.63 | 69.57 3.09 | 69.12 2.63 | 70.09 3.61 | 69.64 3.15 | 70.92 4.44 | 78.57 12.08 |
| ✓ | ✓ | | ✓ | | 71.43 | 71.74 0.31 | 72.27 0.85 | 71.78 0.36 | 72.48 1.06 | 71.49 0.07 | 72.52 1.09 | 73.10 1.68 | 73.82 2.39 | 79.46 8.03 |
| ✓ | ✓ | | | ✓ | 79.05 | 74.70 -4.36 | 76.03 -3.02 | 78.34 -0.72 | 75.59 -3.46 | 77.77 -1.28 | 75.95 -3.11 | 78.00 -1.06 | 79.90 0.85 | 84.65 5.59 |
| ✓ | | ✓ | ✓ | | 71.43 | 72.69 1.27 | 73.20 1.77 | 66.29 -5.14 | 72.74 1.32 | 72.55 1.12 | 72.92 1.5 | 72.97 1.54 | 73.90 2.47 | 80.34 8.91 |
| ✓ | | ✓ | | ✓ | 79.05 | 75.87 -3.19 | 76.89 -2.16 | 78.48 -0.57 | 76.34 -2.72 | 78.32 -0.73 | 76.74 -2.31 | 78.83 -0.23 | 80.08 1.02 | 84.91 5.85 |
| ✓ | | | ✓ | ✓ | 79.05 | 76.84 -2.21 | 77.41 -1.64 | 78.48 -0.57 | 77.33 -1.72 | 78.35 -0.7 | 77.44 -1.61 | 78.09 -0.96 | 80.27 1.22 | 84.83 5.77 |
| | ✓ | ✓ | ✓ | | 71.43 | 71.69 0.26 | 72.55 1.12 | 66.34 -5.09 | 72.09 0.67 | 72.43 1.01 | 72.82 1.4 | 71.74 0.31 | 73.83 2.41 | 80.94 9.51 |
| | ✓ | ✓ | | ✓ | 79.05 | 75.15 -3.9 | 76.39 -2.67 | 77.92 -1.14 | 76.19 -2.86 | 77.67 -1.38 | 76.45 -2.6 | 78.53 -0.52 | 79.44 0.39 | 84.16 5.11 |
| | ✓ | | ✓ | ✓ | 79.05 | 76.34 -2.72 | 77.13 -1.92 | 78.05 -1.01 | 77.20 -1.85 | 77.67 -1.38 | 77.36 -1.69 | 78.16 -0.89 | 80.16 1.11 | 84.88 5.82 |
| | | ✓ | ✓ | ✓ | 79.05 | 77.04 -2.02 | 77.65 -1.4 | 76.11 -2.94 | 76.74 -2.31 | 78.11 -0.94 | 77.80 -1.25 | 78.57 -0.49 | 79.69 0.63 | 85.01 5.95 |
| ✓ | ✓ | ✓ | ✓ | | 71.43 | 72.52 1.09 | 72.95 1.53 | 67.02 -4.41 | 72.60 1.17 | 72.22 0.8 | 73.00 1.58 | 72.61 1.19 | 74.45 3.02 | 82.14 10.72 |
| ✓ | ✓ | ✓ | | ✓ | 79.05 | 75.22 -3.84 | 75.69 -3.37 | 77.92 -1.14 | 75.18 -3.87 | 77.59 -1.46 | 75.52 -3.53 | 78.63 -0.42 | 79.80 0.75 | 85.62 6.57 |
| ✓ | ✓ | | ✓ | ✓ | 79.05 | 76.01 -3.04 | 76.52 -2.54 | 78.00 -1.06 | 76.26 -2.8 | 77.57 -1.48 | 76.55 -2.5 | 78.24 -0.81 | 80.27 1.22 | 85.67 6.62 |
| ✓ | | ✓ | ✓ | ✓ | 79.05 | 76.71 -2.34 | 77.22 -1.84 | 76.21 -2.85 | 76.31 -2.75 | 78.14 -0.91 | 77.05 -2.0 | 78.18 -0.88 | 80.18 1.12 | 85.77 6.72 |
| | ✓ | ✓ | ✓ | ✓ | 79.05 | 76.81 -2.24 | 77.09 -1.97 | 75.80 -3.25 | 76.22 -2.83 | 77.69 -1.37 | 77.10 -1.95 | 78.19 -0.86 | 79.93 0.88 | 85.75 6.7 |
| ✓ | ✓ | ✓ | ✓ | ✓ | 79.05 | 75.70 -3.35 | 76.28 -2.77 | 75.85 -3.20 | 75.54 -3.51 | 77.60 -1.45 | 76.25 -2.80 | 78.16 -0.89 | 81.10 2.05 | 86.32 7.27 |
| | | Mean Δ | | | | -0.81 | -0.37 | -1.24 | -0.75 | -0.14 | -0.35 | 0.30 | 1.57 | 6.76 |
| | | Max Δ | | | | 2.89 | 3.50 | 2.63 | 3.09 | 2.63 | 3.61 | 3.15 | 4.44 | 12.08 |
| | | Min Δ | | | | -4.36 | -3.37 | -6.28 | -3.87 | -1.48 | -3.53 | -1.22 | 0.02 | 3.56 |

Table M.11: Our results on IMGNET-1K dataset for all the possible combinations of combining the zero-shot predictions of CLIP backbones, which we group intro non-parametric and parametric techniques. Also, the best-performing single backbone (SINGLE-BEST) and the ORACLE performance. We present, for each combination of backbones, the improvement , constancy — and deterioration of accuracy performance for each method when we compare it against the SINGLE-BEST backbone. Mean, Max, and Min Δ summarize the difference in performance across methods and backbone combinations.

IMGNET-1K

| ResNet | | ViT | | | SINGLE-BEST | Non-Parametric | | | | Parametric | | | | ORACLE |
|---|---|---|---|---|---|---|---|---|---|---|---|---|---|---|
| 50 | 101 | B-32 | B-16 | L-14 | | VOTE T-1 | VOTE T-3 | CONF | LOG-AVG | C-CONF | C-LOG-AVG | GAC | NLC | |
| ✓ | | | | | | | | | | | | 59.84 −0.00 | | |
| | ✓ | | | | | | | | | | | 62.28 −0.00 | | |
| | | ✓ | | | | | | | | | | 63.35 −0.00 | | |
| | | | ✓ | | | | | | | | | 68.34 −0.00 | | |
| | | | | ✓ | | | | | | | | 75.54 −0.00 | | |
| ✓ | ✓ | | | | 62.30 | 63.76 1.46 | 64.16 1.86 | 59.05 -3.25 | 64.61 2.31 | 63.22 0.91 | 64.47 2.17 | 64.71 2.41 | 65.14 2.84 | 70.75 8.45 |
| ✓ | | ✓ | | | 63.36 | 65.00 1.65 | 65.37 2.01 | 64.40 1.04 | 65.78 2.42 | 64.39 1.03 | 65.54 2.19 | 65.86 2.51 | 66.07 2.71 | 71.76 8.41 |
| ✓ | | | ✓ | | 68.34 | 68.47 0.12 | 68.72 0.37 | 68.15 -0.19 | 68.89 0.55 | 68.09 -0.26 | 68.85 0.51 | 69.60 1.26 | 69.86 1.52 | 74.51 6.17 |
| ✓ | | | | ✓ | 75.53 | 75.13 -0.4 | 75.27 -0.27 | 60.23 -15.3 | 75.09 -0.45 | 74.51 -1.02 | 74.97 -0.56 | 75.97 0.44 | 76.17 0.64 | 80.13 4.6 |
| | ✓ | ✓ | | | 63.36 | 66.03 2.67 | 66.41 3.05 | 60.29 -3.07 | 66.99 3.63 | 65.45 2.09 | 66.70 3.35 | 67.00 3.65 | 67.35 3.99 | 72.96 9.61 |
| | ✓ | | ✓ | | 68.34 | 69.09 0.75 | 69.29 0.95 | 68.54 0.2 | 69.58 1.24 | 68.45 0.11 | 69.40 1.05 | 69.90 1.56 | 70.29 1.95 | 75.15 6.8 |
| | ✓ | | | ✓ | 75.53 | 75.18 -0.35 | 75.35 -0.18 | 62.53 -13.0 | 75.16 -0.38 | 74.56 -0.97 | 75.10 -0.44 | 76.01 0.48 | 76.39 0.86 | 80.28 4.75 |
| | | ✓ | ✓ | | 68.34 | 68.90 0.56 | 69.17 0.82 | 63.17 -5.18 | 69.41 1.07 | 68.48 0.13 | 69.23 0.89 | 69.76 1.42 | 70.04 1.7 | 74.97 6.63 |
| | | ✓ | | ✓ | 75.53 | 75.37 -0.16 | 75.40 -0.13 | 63.53 -12.01 | 75.19 -0.35 | 74.79 -0.74 | 75.20 -0.33 | 75.98 0.45 | 76.31 0.78 | 80.49 4.96 |
| | | | ✓ | ✓ | 75.53 | 75.47 -0.06 | 75.61 0.08 | 68.42 -7.12 | 75.63 0.1 | 75.04 -0.5 | 75.50 -0.03 | 75.98 0.44 | 76.26 0.73 | 80.60 5.06 |
| ✓ | ✓ | ✓ | | | 63.36 | 66.15 2.79 | 66.76 3.4 | 60.27 -3.08 | 67.33 3.97 | 65.51 2.16 | 67.08 3.72 | 67.47 4.11 | 67.87 4.51 | 76.39 13.03 |
| ✓ | ✓ | | ✓ | | 68.34 | 68.22 -0.12 | 68.94 0.6 | 66.82 -1.53 | 69.23 0.89 | 68.17 -0.18 | 69.18 0.83 | 70.11 1.77 | 70.71 2.37 | 77.95 9.61 |
| ✓ | ✓ | | | ✓ | 75.53 | 72.15 -3.38 | 74.09 -1.45 | 62.70 -12.83 | 74.22 -1.31 | 74.03 -1.5 | 74.08 -1.45 | 76.07 0.54 | 76.42 0.88 | 82.41 6.87 |
| ✓ | | ✓ | ✓ | | 68.34 | 68.58 0.24 | 69.20 0.85 | 63.72 -4.62 | 69.45 1.11 | 68.35 −0.0 | 69.45 1.11 | 70.18 1.84 | 70.60 2.26 | 78.06 9.72 |
| ✓ | | ✓ | | ✓ | 75.53 | 72.80 -2.73 | 74.35 -1.18 | 63.26 -12.28 | 74.36 -1.17 | 74.24 -1.3 | 74.32 -1.21 | 76.14 0.61 | 76.35 0.82 | 82.61 7.08 |
| ✓ | | | ✓ | ✓ | 75.53 | 73.95 -1.58 | 74.93 -0.61 | 67.33 -8.21 | 74.98 -0.55 | 74.58 -0.95 | 74.90 -0.63 | 76.16 0.63 | 76.54 1.01 | 82.76 7.23 |
| | ✓ | ✓ | ✓ | | 68.34 | 69.15 0.81 | 69.73 1.39 | 62.46 -5.88 | 70.02 1.68 | 68.66 0.32 | 69.95 1.61 | 70.43 2.09 | 70.87 2.53 | 78.58 10.24 |
| | ✓ | ✓ | | ✓ | 75.53 | 73.32 -2.21 | 74.61 -0.92 | 63.05 -12.49 | 74.63 -0.9 | 74.27 -1.27 | 74.58 -0.95 | 76.10 0.57 | 76.60 1.07 | 82.82 7.29 |
| | ✓ | | ✓ | ✓ | 75.53 | 74.13 -1.4 | 75.04 -0.49 | 67.39 -8.14 | 75.11 -0.42 | 74.59 -0.94 | 75.14 -0.39 | 76.09 0.56 | 76.73 1.2 | 82.89 7.35 |
| | | ✓ | ✓ | ✓ | 75.53 | 74.01 -1.52 | 74.90 -0.63 | 68.56 -6.97 | 75.02 -0.51 | 74.64 -0.9 | 74.98 -0.55 | 76.07 0.54 | 76.55 1.01 | 82.91 7.38 |
| ✓ | ✓ | ✓ | ✓ | | 68.34 | 69.10 0.75 | 69.51 1.17 | 62.51 -5.83 | 69.66 1.32 | 68.43 0.08 | 69.74 1.4 | 70.52 2.18 | 71.09 2.75 | 80.31 11.97 |
| ✓ | ✓ | ✓ | | ✓ | 75.53 | 72.37 -3.16 | 73.71 -1.82 | 63.11 -12.42 | 73.67 -1.86 | 73.94 -1.6 | 73.62 -1.92 | 76.12 0.58 | 76.69 1.16 | 84.08 8.55 |
| ✓ | ✓ | | ✓ | ✓ | 75.53 | 73.47 -2.06 | 74.46 -1.07 | 67.47 -8.06 | 74.46 -1.07 | 74.21 -1.32 | 74.41 -1.12 | 76.10 0.57 | 76.63 1.09 | 84.17 8.64 |
| ✓ | | ✓ | ✓ | ✓ | 75.53 | 73.54 -1.99 | 74.40 -1.13 | 67.54 -7.99 | 74.39 -1.14 | 74.32 -1.21 | 74.39 -1.14 | 76.14 0.61 | 76.65 1.11 | 84.26 8.73 |
| | ✓ | ✓ | ✓ | ✓ | 75.53 | 73.78 -1.75 | 74.64 -0.89 | 67.46 -8.07 | 74.58 -0.95 | 74.32 -1.21 | 74.59 -0.94 | 76.19 0.66 | 76.77 1.23 | 84.42 8.89 |
| ✓ | ✓ | ✓ | ✓ | ✓ | 75.53 | 72.67 -2.86 | 73.95 -1.58 | 67.46 -8.07 | 73.89 -1.64 | 74.06 -1.47 | 73.88 -1.65 | 76.22 0.69 | 76.59 1.06 | 85.29 9.76 |
| | Mean Δ | | | | | -0.54 | 0.16 | -7.09 | 0.29 | -0.40 | 0.21 | 1.28 | 1.68 | 7.99 |
| | Max Δ | | | | | 2.79 | 3.40 | 1.04 | 3.97 | 2.16 | 3.72 | 4.11 | 4.51 | 13.03 |
| | Min Δ | | | | | -3.38 | -1.82 | -15.30 | -1.86 | -1.60 | -1.92 | 0.44 | 0.64 | 4.60 |

