# OpenReview forum: "Synergy and Diversity in CLIP: Enhancing Performance Through Adaptive Backbone Ensembling"
_ICLR.cc/2025/Conference — ICLR 2025 Poster_

### Official Review · Reviewer_oxVz · 2024-10-23

**Soundness:** 3
**Presentation:** 3
**Contribution:** 2
**Rating:** 6
**Confidence:** 4

**Summary:**

This paper analyzes the prediction diversity for CLIP models from OpenCLIP by analyzing the overlap between the sets of correctly predicted images from the test set of ImageNet1k via zero-shot classification. The paper finds that model size increases accuracy, but there exist images that are correctly predicted by smaller models (within same family as well). To combine the models, the paper proposes a method to combine logits of different backbones by scaling the logits using a learned temperature value per backbone. Temperature scaling is implemented usign a neural network called Neural Logit Controller (NLC) which takes the concatenated features of all the backbones and predicts the temperature. NLC is trained using a standard cross-entropy objective, and can be trained using only one image per class. The method boosts the zero-shot and linear probing performance on a number of downstream tasks, and the performance boost correlates with prediction diversity.

**Strengths:**

1. The paper shows that smaller models have certain samples that they can predict correctly, and larger models can give incorrect predictions on these samples. This is an interesting observation, and highlights the need for careful thought when scaling models.

2. The proposed NLC is simple but effective, and can be trained in a data-efficient manner as demonstrated in Section 4.2 and 4.3.

**Weaknesses:**

1. The authors claim on line 200 that the results are non-obvious is not entirely true. Prior work has shown similar results across vision and language domains, albeit on different models [1, 2, 3]. [1] have shown prediction diversity on a range of models trained in a supervised setting and propose a method for combining models using distillation. They show that weaker models are more specialized in certain subsets of data. While the metric for measuring diversity is different, they study the same core problem. [2] show that finetuning models on different domains and then averaging model weights can improve out-of-distribution performance (this assumes the same model size, but studies a similar problem of different models being good at different subsets of data).

2. **Incomplete comparisons in few-shot experiments (Section 4.3)**:
- The results in Section 4.3 (Comparison with Few-Shot Adapter Methods) are incomplete: Figure 10 and Table 1, which show the zero-shot performance of TiP-Adapter applied to individial backbones and NLC does not show the complete picture - since NLC is an ensemble method, a proper comparison would be other ensemble methods. Similar is the case when NLC is combined with Tip-Adapter. In principle, any ensemble method can be combined with Tip-Adapter, so a proper comparison would be with other ensemble methods applied in a similar manner as NLC. Without these results, the comparisons are incomplete.

3. **Number of training samples**:
- Point 2 is especially more important considering that few-shot experiments in Section 4.3 are the only experiments which demonstrate the data efficiency of NLC, which is claimed to be a major contribution by the authors. From Section 4.1: "The non-parametric ones are “static” while the parametric ones use the training split of each target dataset to adjust the ensemble adaptively." implies that NLC uses the entire training set for calibration. For few-shot experiments, if this data efficiency is applicable to other ensemble methods as well, this cannot be attributed to NLC, but would be a general property of ensemble methods.
- From linear probing experiments in Section 4.2, Figure 9 shows that with a similar training budget, NLC does not provide a significant improvement over Super Learner (SL). Mean increase of 1.6 is the same, and SL also does not lead to any degradation over the single best model. The max increase seems to be a noisy metric to track, since MoE can show a maximum 20.8 increase but also shows degradation, indicating the max increase is really dataset specific.

4. The paper claims that the method can be combined with Cascade, but does not provide the trainiing details. The paper also does not compare the performance of the proposed method with the original Cascade method. Minor: The paper also misses a citation for a cascade method [4].

5. Although the paper mentions this in the limitations, fusing models at training stage has been done in [1] in supervised classification settings, hence the additional inference overhead of NLC is a major practical limitation. Furthermore, while the paper shows the potential for applying the method to other tasks like image retrieval, the paper does not actually study this using NLC, making the impact of the method limited.

**References**:

[1] Karsten Roth, Lukas Thede, A. Sophia Koepke, Oriol Vinyals, Olivier J. Hénaff, Zeynep Akata: Fantastic Gains and Where to Find Them: On the Existence and Prospect of General Knowledge Transfer between Any Pretrained Model. ICLR 2024

[2] Alexandre Ramé, Kartik Ahuja, Jianyu Zhang, Matthieu Cord, Léon Bottou, David Lopez-Paz: Model Ratatouille: Recycling Diverse Models for Out-of-Distribution Generalization. ICML 2023

[3] Ruiqi Zhong, Dhruba Ghosh, Dan Klein, Jacob Steinhardt: Are Larger Pretrained Language Models Uniformly Better? Comparing Performance at the Instance Level. ACL/IJCNLP (Findings) 2021: 3813-3827

[4] Neeraj Varshney, Chitta Baral: Model Cascading: Towards Jointly Improving Efficiency and Accuracy of NLP Systems. EMNLP 2022: 11007-11021

**Questions:**

1. What is the standard deviation of the accuracy when using different subsets of data for training the methods in few-shot setting? This is important to understand the robustness of the method when using different training samples.

---

> ### Author Response · Authors · 2024-11-26
>
> Thank you for your thorough review
>
> **W1. Weaker models are more specialized**
> Thank you for pointing us to these works, which we agree are closely related to ours. We will add all of them to the related work section. Regarding more precise comparisons, we would like to point out the following:
> - [2] focuses on model averaging to approximate an ensemble method with a single one for out-of-distribution generalization but does not study CLIP backbones or adaptive ensembling like ours.
> - [3] investigates instance-level performance differences in pre-trained language models, which is tangentially related but distinct from our focus on vision-language models and CLIP-specific complementarity.
>
> Due to the above, the rest of our response concerns [1], which we believe is more closely related to our proposals. We would like to bring attention to the following distinctions.
>
> - **Training Setup:** [1] uses `timm` models trained on ImageNet with arbitrary settings, meaning that methods are not necessary using the same loss function or regularizer. The selection of loss functions impacts the inductive biases of the model, which could potentially help the models to have more diversity. In contrast, we use a *Foundational model*, CLIP backbones, trained on the same dataset in a contrastive learning approach using the same loss function. In this way, we are sure that the diversity of predictions comes from the backbone. We believe that the results provided by [1] are not sufficient to conclude that the differences in classification performance are due to either utilizing a weaker architecture or the selection of a loss function.
> - **Complementarity Analysis:** First, we would like to clarify that our analysis is about diversity in the correct predictions of different backbones. Based on these results, then propose a mechanism to exploit their diversity through complementarity. While [1] conducts pairwise teacher-student analysis, in our paper, we analyze nine backbones simultaneously. Although we agree that the pairwise analysis by [1] is informative, we think it does not provide evidence to show if a given backbone is better for a specific type of data. In contrast, our analysis does show that some **backbones are more resilient to certain types of image variations than others**, Figure 7.
> - **Methodology:** The knowledge distillation approach in [1] requires access to a large training set (e.g. ImageNet), a data augmentation procedure, and partitioning of data into sets for continual learning. These partitions are built from a complementarity analysis, and the idea is to perform knowledge transfer to mitigate catastrophic forgetting. For example, [1] performs knowledge transfer for the set of data where one backbone is better than the other and retention on the set where they would like to maintain performance. Additionally, this approach requires retraining the model in order to distil the knowledge. In contrast, our method does not require data partitioning nor backbone retraining: it preserves the model's original knowledge, and it is shown to be effective in few-shot scenarios. Reaching up to 120% (Figure 2) relative improvement.
> - **Dataset Breadth:** [1] examines four datasets (ImageNet, CUB, StanfordCars, Caltech256), whereas we evaluate diversity and complementarity across 21 datasets. Additionally, we specifically show the diversity of zero-shot CLIP backbones is higher for certain datasets.
> - **Performance improvement:** [1] obtains improvements of nearly 5% for the ConvNeXt network after the knowledge distillation process. In contast, our approach can directly leverage the backone prediction diversity in order to improve the performance of our best Convolutional network ResNet50x64 by a similar 5%.
> - **Contemporaneity:** We believe [1] is contemporary to our work.

---

> > ### Author Response · Authors · 2024-11-26
> >
> > **W2. and W3.1 Ensembles comparison in Tip-Adapter** Thank you for pointing out this, we will add this results to our main paper for completion.
> > The following tables show the performance of SuperLearner (SL), which is the closest ensemble method to ours, and NLC on top of Tip-Adapter and Tip-Adapter-F. For these runs, we use the same settings, i.e., learning rate, batch size, few-shots, as in our paper, focusing on the datasets where Tip-Adapter was originally tested. We can see that the performance of NLC is always better than the performance of SL for each k-shot setting, outperforming SL by up to 10% in some scenarios. Moreover, in some cases SL could not improve the performance of the best Tip-Adapter at all, such as in the DTD dataset.
> >
> > |  	| Eurosat Tip-Adapter 	|  	|  	|  	|  	|
> > |:---:|:---:|:---:|:---:|:---:|:---:|
> > |  	| 1 	| 2 	| 4 	| 8 	| 16 	|
> > | SL 	| 77.8 	| 81.5 	| 85.6 	| 86.4 	| 85.4 	|
> > | NLC 	| 88.4 	| 86.8 	| 89.7 	| 90.1 	| 90.7 	|
> >
> > |  	| Eurosat Tip-Adapter-F 	|  	|  	|  	|  	|
> > |:---:|:---:|:---:|:---:|:---:|:---:|
> > |  	| 1 	| 2 	| 4 	| 8 	| 16 	|
> > | SL 	| 84.5 	| 86.7 	| 89.9 	| 89.0 	| 93.1 	|
> > | NLC 	| 89.7 	| 88.3 	| 91.4 	| 91.1 	| 93.8 	|
> >
> > |  	| DTD Tip-Adapter 	|  	|  	|  	|  	|
> > |:---:|:---:|:---:|:---:|:---:|:---:|
> > |  	| 1 	| 2 	| 4 	| 8 	| 16 	|
> > | SL 	| 58.7 	| 64.1 	| 65.5 	| 70.3 	| 71.5 	|
> > | NLC 	| 63.7 	| 66.1 	| 68.4 	| 71.6 	| 73 	|
> >
> > |  	| DTD Tip-Adapter-F 	|  	|  	|  	|  	|
> > |:---:|:---:|:---:|:---:|:---:|:---:|
> > |  	| 1 	| 2 	| 4 	| 8 	| 16 	|
> > | SL 	| 62.2 	| 67.3 	| 71.4 	| 74.8 	| 78.8 	|
> > | NLC 	| 67.2 	| 68.7 	| 73.3 	| 75.4 	| 79.1 	|
> >
> > |  	| FGVC Tip-Adapter 	|  	|  	|  	|  	|
> > |:---:|:---:|:---:|:---:|:---:|:---:|
> > |  	| 1 	| 2 	| 4 	| 8 	| 16 	|
> > | SL 	| 39.1 	| 44.2 	| 47.7 	| 51.2 	| 54.8 	|
> > | NLC 	| 40.9 	| 44.5 	| 47.9 	| 51.5 	| 55.0 	|
> >
> > |  	| FGVC Tip-Adapter-F 	|  	|  	|  	|  	|
> > |:---:|:---:|:---:|:---:|:---:|:---:|
> > |  	| 1 	| 2 	| 4 	| 8 	| 16 	|
> > | SL 	| 46.7 	| 49.1 	| 53.4 	| 57.4 	| 62.4 	|
> > | NLC 	| 46.8 	| 49.2 	| 53.6 	| 57.5 	| 62.6 	|
> >
> > |  	| Pets Tip-Adapter 	|  	|  	|  	|  	|
> > |:---:|:---:|:---:|:---:|:---:|:---:|
> > |  	| 1 	| 2 	| 4 	| 8 	| 16 	|
> > | SL 	| 93.8 	| 93.6 	| 94.3 	| 94.4 	| 94.5 	|
> > | NLC 	| 94.5 	| 94.2 	| 94.9 	| 95 	| 95.3 	|
> >
> > |  	| Pets Tip-Adapter-F 	|  	|  	|  	|  	|
> > |:---:|:---:|:---:|:---:|:---:|:---:|
> > |  	| 1 	| 2 	| 4 	| 8 	| 16 	|
> > | SL 	| 94.5 	| 94.6 	| 95.2 	| 95.2 	| 95.1 	|
> > | NLC 	| 95.2 	| 95 	| 95.3 	| 95.2 	| 95.5 	|
> >
> >
> > Moreover, in Tables C.1 and D.2, we show that parametric ensemble cannot exploit the diversity of the backbones in place, obtaining worst performance than NLC using the full dataset. Section G, in our supplemnentary material, shows the performance of NLC trained on a limited amount of samples. Our results for those experiments show that as little as 1 sample per class proves to be enough to improve the performance with respect to the best single backbone.
> >
> > **W3.2 Dataset-Specific Variability in Linear Probing Results** Thank you for your insightful comment. The linear probing experiments in Section 4.2 show that, with a similar training budget, NLC does not significantly outperform Super Learner (SL), with a consistent 1.6% mean increase and no degradation from SL. However, the max increase is dataset-specific and can be noisy, as observed with MoE, which shows both improvements and degradations depending on the dataset. It is important to note that linear probing reduces diversity in predictions, limiting the gains observed with NLC. We will clarify this in the camera-ready version. We also would like to refer the reviewer to Appendix M for possible combinations of backbones.
> >
> > **W4. Training details Cascade** As explained in section A in the supplemental material, for the experiments with Cascades, we use the probability as the confident measurement with a threshold of 0.9. When the method is confident with a probability bigger than 0.9 we produce a final prediction, otherwise, we add the next backbone.

---

> > > ### Author Response · Authors · 2024-11-26
> > >
> > > **Q1 What is the standard deviation of the accuracy when using different subsets.**
> > >
> > > We thank the reviewer for their question. We will add this results to the main paper. We run 5 different seed to train NLC on the different few-shots setting, we could see the results on the following table. Results show that the the standard deviation of our obtained performance is stable and low across all of our studied datasets, suggesting that our approach is not sensitive to the effects of different samples.
> > > |    	| cars 	|     	| cifar10 	|     	| cifar100 	|     	| clevr 	|     	| country211 	|     	| cub  	|     	| dtd  	|     	| eurosat 	|     	| fgvc_aircraft 	|     	| flowers 	|     	| food 	|     	| gtsrb 	|     	| imagenet1k 	|     	| mnist 	|     	| pcam 	|     	| pets 	|     	| renderedsst2 	|     	| resisc45 	|     	| stl10 	|     	| sun397 	|     	|
> > > |:----:|:------:|:-----:|:---------:|:-----:|:----------:|:-----:|:-------:|:-----:|------------:|:-----:|:------:|:-----:|:------:|:-----:|:---------:|:-----:|:---------------:|:-----:|:--------:|:-----:|:------:|:-----:|:-------:|:-----:|:-----------:|:-----:|:-------:|:-----:|:------:|:-----:|:------:|:-----:|:--------------:|:-----:|:----------:|:-----:|:-------:|:-----:|:--------:|:-----:|
> > > |    	| mean 	| std 	| mean    	| std 	| mean     	| std 	| mean  	| std 	| mean       	| std 	| mean 	| std 	| mean 	| std 	| mean    	| std 	| mean          	| std 	| mean    	| std 	| mean 	| std 	| mean  	| std 	| mean       	| std 	| mean  	| std 	| mean 	| std 	| mean 	| std 	| mean         	| std 	| mean     	| std 	| mean  	| std 	| mean   	| std 	|
> > > | 2  	| 82.5 	| 0.2 	| 94.6    	| 1.3 	| 77.9     	| 0.5 	| 26.4  	| 1.3 	| 34.8       	| 0.2 	| 70.8 	| 0.6 	| 59.6 	| 0.3 	| 63.9    	| 1.9 	| 37.1          	| 0.2 	| 79.4    	| 1.1 	| 93.8 	| 0.2 	| 55.9  	| 0.5 	| 77.5       	| 0.2 	| 82.5  	| 1.0 	| 61.3 	| 0.8 	| 94.4 	| 0.4 	| 72.4         	| 0.8 	| 70.2     	| 0.7 	| 98.9  	| 0.0 	| 71.6   	| 0.2 	|
> > > | 4  	| 82.7 	| 0.1 	| 95.4    	| 0.1 	| 78.1     	| 0.2 	| 28.2  	| 0.6 	| 35.4       	| 0.2 	| 71.2 	| 0.2 	| 60.3 	| 0.4 	| 68.5    	| 2.1 	| 37.7          	| 0.4 	| 80.7    	| 0.3 	| 94.0 	| 0.2 	| 56.7  	| 1.8 	| 77.5       	| 0.2 	| 84.7  	| 0.1 	| 65.5 	| 3.8 	| 94.5 	| 0.3 	| 71.9         	| 0.7 	| 71.3     	| 0.5 	| 99.0  	| 0.1 	| 71.5   	| 0.2 	|
> > > | 8  	| 83.0 	| 0.1 	| 95.5    	| 0.3 	| 78.2     	| 0.2 	| 29.8  	| 0.3 	| 34.3       	| 0.5 	| 72.5 	| 0.0 	| 60.7 	| 0.4 	| 71.6    	| 1.8 	| 36.9          	| 0.1 	| 80.8    	| 0.8 	| 94.1 	| 0.1 	| 57.3  	| 1.0 	| 77.4       	| 0.1 	| 86.8  	| 0.4 	| 69.3 	| 1.6 	| 94.7 	| 0.3 	| 71.3         	| 0.3 	| 72.7     	| 0.3 	| 99.2  	| 0.2 	| 71.7   	| 0.0 	|
> > > | 16 	| 83.8 	| 0.2 	| 95.8    	| 0.2 	| 78.4     	| 0.1 	| 31.5  	| 0.3 	| 34.9       	| 0.1 	| 72.9 	| 0.5 	| 61.5 	| 0.2 	| 74.0    	| 0.7 	| 39.3          	| 0.2 	| 82.7    	| 0.6 	| 94.0 	| 0.1 	| 63.0  	| 0.1 	| 77.4       	| 0.2 	| 87.7  	| 0.4 	| 72.2 	| 1.4 	| 95.1 	| 0.1 	| 72.3         	| 0.9 	| 74.0     	| 0.1 	| 99.2  	| 0.1 	| 71.6   	| 0.2 	|
> > > | 32 	| 84.1 	| 0.1 	| 95.9    	| 0.1 	| 78.5     	| 0.2 	| 31.7  	| 3.9 	| 34.6       	| 0.3 	| 73.8 	| 0.3 	| 63.0 	| 0.4 	| 76.5    	| 0.9 	| 40.4          	| 0.6 	| 82.7    	| 0.6 	| 94.1 	| 0.0 	| 65.0  	| 0.1 	| 77.2       	| 0.1 	| 85.8  	| 0.4 	| 75.9 	| 0.5 	| 95.3 	| 0.0 	| 73.2         	| 0.5 	| 75.3     	| 0.2 	| 99.3  	| 0.0 	| 71.4   	| 0.4 	|

---

> ### Comment · Reviewer_oxVz · 2024-11-26
>
> Thank you for your clarifications. Most of my concerns have been addressed. Following is a clarification on the prior work mentioned in W1
>
> - While [3] might not be exactly the same, they tackle the same underlying problem that different models have different instance level predictions regardless of the model size. Hence, even if this work is specifically for vision-language models, the writing should reflect that prior work in this area of instance level analysis been done. In particular, statements like "This non-obvious insight is the key motivation for the ensemble method" are misleading, but can be revised to include reference to other works that have done the same for other domains, but not for vision language models.
> - [2] might not be working with vision language models, but they also tack a similar problem - models that are finetuned on different subsets of data (i.e., have different specializations) can be combined to perform well on ood data for any of the domains, i.e., helps with generalization.
> - I understand that there are differences in the way [1] and this work handles analysis of complementary data. This work is fairly recent - it was published in ICLR2024 - but I believe it is old enough for it to be discussed (contemporary work according to ICLR2025 guidlines is within 4 months prior to submission deadline). I would not expect a comparison with the work and believe the method is different in terms of how the ensembles handles complementary data. However, the limitations section needs a revision to include a refernece to [1], since the fusion of backbones during training has been done. This particular point is not a major weakness, and a citation and comparison of differences should be sufficient.
>
> I don't mind increasing my score to 6 given that the paper is revised to include the new results and references to prior works.
>
> ---
>
> **Follow up questions**
>
> In Appendix G, lines 1153-1154 mention that the degradation stops when 8 training samples are used , but the degradation appears to be present when number of samples used is 32. This seems counter intuitive. Can the authors comment on the reason for this?

---

> > ### Author Response · Authors · 2024-11-27
> >
> > *Revised version Log*
> >
> > 1. We have added the reference suggested by the reviewer for work related to Cascade (L111)
> > 2. We have added references [1,2,3] in L200 and L498 (related work), and also toned down our statement to reflect on and acknowledge this prior work.
> > 3. We have added in L89 the difference with SuperLearner
> > 4. We have acknowledged the knowledge distillation approach [1] in the limitation section L530
> > 5. We have updated tables F.3--F.6 and G.1
> >
> > **Follow-up question**
> >
> > Thank you for bringing this to our attention. Upon review, we discovered that the counterintuitive result was caused by a mistake in Table G.1 of our paper. Specifically, the entry for one dataset (MNIST) was incorrect, showing a value of 77.0 instead of the correct value of 85.3, as indicated in the table provided in our earlier comment. This table has been updated in the revised version of the paper.

---

> > > ### Comment · Reviewer_oxVz · 2024-11-27
> > >
> > > Thank you for the clarification. I have updated my score to reflect my review.

---

### Official Review · Reviewer_6yyE · 2024-11-02

**Soundness:** 3
**Presentation:** 4
**Contribution:** 3
**Rating:** 8
**Confidence:** 4

**Summary:**

This work investigates the differences across various CLIP-trained vision backbones with respect to classification performance and robustness, and proposes an adaptive combination of pretrained CLIP backbones, which is able to increase performance beyond regular enabling.

I very much appreciate the author's systematic approach: they first show that there exist (large) differences between (zero-shot) classification performance of individual CLIP-trained vision backbones. Then they drive an oracle performance upper-bound and investigate  complementarity of different backbones concluding that different backbones classify different subsets of images differently. This motivates the authors to propose an adaptive ensembling approach called Neural Logit Controller (NLC) to combine individual backbones to one prediction.

The proposed NLC is based on network calibration using learned temperature scaling. To do so, the method needs to be informed by a supervised signal given by a single labeled example per class, as the authors claim.

Experimental results on a large collection of datasets show that the proposed approach is able to outperform state-of-the-art few-shot methods. For some datasets by large (up to 39.1), on average by 9.1% over the best performing individual backbone and better than baseline ensembling methods. However, outperformance is much smaller when data-specific linear heads are trained (=1.6%). Additional results show a promising combination of Tip-Adapter with the proposed approach.

As mentioned before, I very much appreciate the systematic evaluation of the growing number of available vision foundation models. Having so many foundation models available, renders the question how *foundational* these models are. So maybe the research community should invest more time in the investigation and how to harness these foundation models. To do so, we would need to understand their differences, strengths and weaknesses and then derive from that propose a novel aggregation, ensembling or merging methods. ...and this is what this paper is doing.

I also very much appreciate the large number of experiments and evaluations done by the authors to support this work - including the additional results in the appendix. One is just one thing, looking at the results and in particular the difference in performance between Fig. 8. and Fig. 9. I might be wrong but can the decrease of outperformance from 9.1% to 1.6% be linked to the additional supervision signal intrinsic to the proposed NLC method? Is this what we see here? And if yes, could the results in Fig. 8 be biased towards the supervision part of NLC? I will raise it again in the question section.


----------------------reply to rebuttal----------------------


I would like the authors for the information provided in the rebuttal. I went through the paper again and look in particular at the appendix and other reviewer / author discussions and given that I would like to increase my rating for this submission. I think this work contributes to the research community with its detailed evaluation and proposed ideas.

**Strengths:**

- **(S1)**: this paper presents an in-depth analysis of different CLIP-trained vision backbones with respect to different dimensions such as few-shot image classification, complementarity, robustness, and diversity.

- **(S2)**: this paper proposes a novel method to combine the strengths and weaknesses of available CLIP-trained vision backbones able to outperform regular ensembling methods on a large amount of datasets.

- **(S3)**: this paper is easy to read and follow given the systematic build-up of the paper.

**Weaknesses:**

- **(W1)**: Maybe the large out-performance of the proposed method from Fig.7 is linked to the injection of a supervised signal and therefore the comparison against regular ensembles might be biased. I have raised this in **(Q1)**.

- **(W2)**: I would love to see more details about the injection of the supervision signal given the one labeled example per class. I am not sure if this is what I can find in appendix G and Table G.1?

- **(W3)**: not really a weakness but some minor things:
- - page 3, line 134: train a pair of vision and text encoders φ_l and φ_v <- I think the first is the language and the second is the vision encoder.
  - Fig 4: it is not clear which setup corresponds to the dashed lines.

**Questions:**

- **(Q1)**: As mentioned above, here is the question raised earlier: looking at the results and in particular the difference in performance between Fig. 8. and Fig. 9. I might be wrong but can the decrease of outperformance from 9.1% to 1.6% be linked to the additional supervision signal intrinsic to the proposed NLC method? Is this what we see here? And if yes, could the results in Fig. 8 be biased?

- **(Q2)**: why is the MoE missing in Fig.8? I assume that the linear heads from the experiment of Fig.9 are serving as MoE?

- **(Q3)**: on page 6, lines 298, with one-layer MLP, do you mean a MLP with one hidden layer?

---

> ### Author Response · Authors · 2024-11-26
>
> Thank you for your insightful feedback. We greatly appreciate your recognition of the importance of systematically evaluating foundational models like CLIP. As you pointed out, with the increasing availability of these models, understanding their strengths, weaknesses, and complementarity is critical for advancing the field. We agree that exploring novel aggregation, ensembling, and merging methods is vital to fully harness the potential of these foundation models, and we are excited to contribute to this ongoing exploration with our work. Your feedback further strengthens the motivation behind our study.
> Moreover, We deeply appreciate your recognition of the systematic nature of our approach and the value of our proposed NLC. We also appreciate your comments on the large-scale experiments and evaluations.
>
> **W1, Q1** Thank you for your observation. We agree that the large performance improvement seen in Figure 8 could be influenced by the injection of the supervised signal. However, our method outperforms other parametric ensembles such as SL. This demonstrates that the benefits of our approach go beyond just the supervised signal, and it remains an improvement over standard ensemble techniques.
>
> Moreover, The decrease in performance improvement from 9.1% to 1.6% between Figure 8 and Figure 9 can be attributed to the fact that linear probing reduces the diversity in predictions across backbones. By relying on a fixed linear head, the model tends to converge towards a more uniform prediction across backbones, limiting the space for improvement. As a result, the gains observed with the NLC method are smaller in this setup. We will clarify this point in the manuscript to highlight the role of reduced diversity in linear probing. We would like to refer the reviewer to Tables B1 and D1, where we see linear classifiers reaching performance and thus leaving relatively smaller room for improvement compared to zero-shot backbones. For example, in the case of Eurosat, the best backbone reaches 97.1%, and the Oracle reaches 99.6%.
>
> **W2 Injection of the supervision:** Thank you for your question! Yes, the details about the injection of the supervision signal using one labelled example per class can indeed be found in Appendix G and Table G.1. We will make sure to clarify this process further in the main text to ensure the methodology is fully understandable.
>
> **W3 Minor things:**
> - Page 3, Line 134: You are correct—the first encoder is indeed the language encoder (φ_l), and the second is the vision encoder (φ_v). We will clarify this distinction in the manuscript.
>
> - Figure 4: We appreciate your comment. We will ensure that the dashed lines are properly labelled or annotated to clarify the setup they correspond to in the figure.
>
>
> **Q2 MoE Missed** Thank you for your question. The MoE (Mixture of Experts) results are not shown in Figure 8 because the focus there is on comparing the proposed NLC approach with other ensemble methods, and MoE was not included in that specific experiment. Regarding your assumption, the linear heads used in the experiments shown in Figure 9 are indeed part of the broader context of how MoE-like structures might work within our framework. We will clarify this in the manuscript to avoid any confusion and ensure the setup is fully understood.
>
> **Q3 MLP with one hidden layer** Thank you for pointing this out we will fix this in the final version. We meant a MLP with one hidden layer.

---

### Official Review · Reviewer_7ZyN · 2024-11-02

**Soundness:** 3
**Presentation:** 3
**Contribution:** 2
**Rating:** 5
**Confidence:** 4

**Summary:**

This paper focuses on vision-language backbones, especially two-tower CLIP like network structure. It evaluates the classification tasks across several datasets. The main idea is to propose an ensemble strategy in logic space by adaptively learning the temperature across different versions of pretrained CLIP backbones. Empirical results are based on a large group of image classification datasets. It also provides detailed analysis for a better intuition.

**Strengths:**

1. Training large-scale backbone requires high computational cost. Therefore, thinking about how to combine existing pretrained backbones is a valuable research topic. Ensemble is a reasonable way to do so.
2. Empirical results are extensive for image classification task including several datasets.
3. Instead of making ensemble directly on logit from different backbone versions, it proposes a learnable module based on temperature scaling.
4. Overall, the paper writing is in a good shape and easy to follow.

**Weaknesses:**

1. More evaluation tasks are valuable to be considered to further support the paper statement. Even if limited with CLIP backbone, adding more retrieval tasks will be supportive.
2. Ensemble on logit is still relatively not novel enough as a research work. Compared with previous ensemble, it still relies on the output of each individual model and make the late fusion.
3. Such ensemble increases the performance but requires much more test time, and it also requires to conduct a post-learning process. The overall pipeline is not that straightforward.
4. It looks like the proposed pipeline cannot obtain improvement by adding more training shots, more analysis and intuition will be helpful.

**Questions:**

Please refer to the weaknesses section above.

---

> ### Author Response · Authors · 2024-11-26
>
> Thank you for your thorough review.
>
> **W1 More Evaluation** Please check section E and K in the supplemental material, where we present the complementarity of CLIP backbones on the image-to-text and text-to-image retrieval tasks. Table K.3 shows patterns of complementarity across backbones comparable to those in the classification setting. On Flick30k, the upper bound shows possible improvements of 20 percentage points on both text and image retrieval. On MSCOCO, >25pp. We selected Flick30k and MSCOCO because they are standard benchmarks for image-to-text and text-to-image retrieval, covering diverse and challenging scenarios.
>
> Note that we conducted extensive experiments across various tasks, including image classification, few-shot adaptation, out-of-distribution evaluation, and image-to-text and text-to-image retrieval, utilizing a total of 27 datasets.
>
> **W2 Ensemble on logit** Thank you for the observation. While ensemble methods using late fusion are not new, our approach introduces adaptive fusion through instance-specific calibration, setting it apart from static methods. This addresses the limitations of traditional ensembles. Thus, we believe the Neural Logit Controller (NLC) introduces a significant advancement over traditional ensembling methods. Specifically:
>
> - **Adaptive Fusion:** Unlike standard ensemble methods (e.g., averaging or voting), NLC dynamically learns to weight the contributions of different backbones using a one-layer MLP. This allows for instance-specific calibration, rather than static fusion strategies. We show that is needed to exploit the diversity in backbones predictions.
>
> - **Efficient Use of Diversity:** NLC explicitly exploits diversity in backbone predictions to enhance performance while maintaining computational efficiency when combined with Cascade. It achieves higher accuracy with fewer GFLOPs compared to naive ensembles.
>
> - **Minimal Supervision:** NLC achieves substantial improvements using as little as one labeled example per class for tuning. This low-resource requirement differentiates it from other adaptive techniques.
>
> These innovations showcase that while our method builds on established ensemble techniques, it addresses their limitations and offers practical advancements for real-world deployment. To the best of our knowledge, our method is compared against a comprehensive selection of alternative resembling methods, so we would appreciate it if the reviewer could point out more specifically in which sense our work offers limited novelty when "Compared with the previous ensemble".
>
> **W3. Test-Time**:
>
> Thank you for pointing this out. While all ensemble methods do increase test time and introduce additional steps in the pipeline, their performance gains often justify these complexities, particularly in applications where accuracy and robustness are critical, e.g., healthcare (PCAM) or autonomous systems (Eurosat). Additionally, many real-world pipelines already involve post-processing or model refinement, making such methods more feasible in practice. That said, the trade-offs depend on the specific use case.
>
> **W4. More Training Shots Analysis:** We think the Reviewer may be referring to the results in Table 1, where it is possible to see that the performance of NLC via k-shots on ImageNet-1k grows more slowly compared to our baselines. Regarding this observation, we would like to point out that we do not observe this is the case for our other studied datasets, as shown in Table F.3, F.4, F.5, F.6, G.1 and C.1. Moreover, the Table C.1 shows that when using the full training set to tune the NLC, we observed an average performance improvement of 9.3%, compared against a 2.9% when using 16 shots, suggesting that overall the NLC generalizes better with more data. However, the amount of improvement will depend of on prediction diversity across backbones, as shown in Figure 2.

---

### Official Review · Reviewer_Gar3 · 2024-11-03

**Soundness:** 3
**Presentation:** 3
**Contribution:** 2
**Rating:** 6
**Confidence:** 4

**Summary:**

This paper starts with the finding that existing CLIP models show diverse behaviours (error distributions) on classification tasks, even if they are pre-trained on the same dataset (WIT-400M). This indicates potential for model ensembling. The authors then propose a learnable ensemble module that adjusts the logits of each model with a learned temperature value conditioned on the input image, and sums up the logits of all models for the final prediction. This can also be combined with an existing cascade strategy for better efficiency. Experiments show consistent improvements over the best single backbones.

**Strengths:**

- This paper presents a finding that the logits of CLIP models are very diverse, and the more diverse their logits are, the higher performance can be reached by their ensemble.
- This paper presents a comprehensive study on the diversity and properties of CLIP's logits and possible factors behind them.
- The results produced by the ensemble model are considerably higher than single models. When compared with other ensemble methods, there is also some improvement.
- The paper is clearly written and is easy to understand.

**Weaknesses:**

- The setting of training the ensemble module might harm generalizability and more clarifications are needed. It requires training data for each class regarding which model is better for each evaluation dataset, which is ad-hoc and remains questionable whether it harms CLIP's ability to generalize to unseen classes, concepts, or compositions.
- It is not very surprising that more diverse models enable better ensembles, especially when the oracle model is picked per-sample, this adds a lot of flexibility compared to ensembling without data conditions.
- The method itself is related to one prior method (Super Learner), and the main improvement is adding image conditions. But when compared with SL, the performance improvement is not big.
- From a practical viewpoint, how much additional inference-time computation is required? I see the results of FLOPS, and it would be better also to report GPU memory and inference time.

More detailed classifications are elaborated in the questions section.

**Questions:**

1.1: How is the training data for the ensemble module curated and how is it split from other data? Please provide more details.

1.2: When the training data for the ensemble module is only available in some classes, does it generalize well to novel classes? Also, when training with only some fixed prompts, does it overfit these prompts?

1.3: Continue to 1.2, considering more complex concepts (eg, [ARO](https://github.com/mertyg/vision-language-models-are-bows) or [SugarCrepe](https://arxiv.org/abs/2306.14610) datasets), can the ensemble module correctly rerank CLIP models on novel compositions of concepts?

2.1: When trained on the same dataset, different backbones in the same series (eg, R-50/101/50x4, ViT-S/B/L) make different mistakes. Does this finding only apply to CLIP or also apply to other methods (eg, vanilla supervised learning)?

2.2: Is diversity always preferred when selecting models in a per-sample way, no matter whether using CLIP or not?

3.1: When inferencing with the proposed method over a dataset, does the user need to load all CLIP models to the device? How much memory is needed compared to using single models, and how much inference time is needed?

3.2: Can the authors also provide a comparison in computation and time cost with other ensemble methods (eg, SL, MoE)?

Minor: L815: OpenCLIP is a project developed by LAION. The models mainly considered in this paper are trained by OpenAI, and the corresponding dataset is called WIT-400M.

---

> ### Author Response · Authors · 2024-11-26
>
> Thank you for your thorough review.
>
> - **Q1.1 How the model is trained?** As explained in Lines 301, 366, and 401, we use the training set of each dataset to train NLC. In Section 4.2, where we train a linear classifier on top of the logits, we use 90% of the training set for the linear classifier and reserve 10% for training the NLC. For the Few-shot adapter method, we follow the Tip-Adapter setting and use the same validation split L455.
>
> - **W2, Q1.2 and Q1.3 Does it generalize well to novel classes? and Novel compositions of concepts?**
> Thank you for your suggestion. NLC is trained using the image representation of the target dataset. Then, it combines the logits such that creates a new one that is tailored to the target dataset. Thus, using NLC that has been trained on a dataset A in a dataset B is possible as long as the classes are fixed. Due to the limited time during the rebuttal process, we are unfortunately unable to provide results now, but are commited to doing so for the camera-ready version.
>
>
> - **Q 2.1 Diversity on Vanilla Supervised Learning** We evaluated Oracle performance on MAE backbones (ViT-Base-16, ViT-Large-16, and ViT-Huge-14) using the official implementation and shared fine-tuned weights on ImageNet-1k.
>
>     Our results indicate a 3% improvement in performance with the Oracle compared to the best individual backbone, **reinforcing our argument of complementarity across backbones**.
>
> |                       | ViT-Base     | ViT-Large     | ViT-Huge     | ORACLE     |
> |:-----------------:    |:--------:    |:---------:    |:--------:    |--------    |
> | ImageNet accuracy     | 83.664       | 85.952        | 86.928       | 90.006     |
>
>
> - **Q2.2 Is diversity always preferred when selecting models in a per-sample way, no matter whether using CLIP or not?** We believe diversity is key to exploiting the strengths of different models. Figure 2, illustrates the correlation between diversity and performance improvement. When diversity is low—meaning all models predict the same label—there’s limited room for improvement. Therefore, diversity is always preferred, but the challenge lies in how to exploit it effectively. In the context of CLIP, leveraging diverse backbones helps capture different aspects of the data, but for other models, careful management of diversity is crucial to avoid inefficiencies.
>
> - **Q3.1 Does the user need to load all CLIP models to the device?** No, the user does not need to load all CLIP models to the device. In the proposed method, only the selected backbones for a given input need to be loaded. This is done dynamically, allowing for efficient use of resources. The Neural Logit Controller (NLC) can adjust and combine the outputs of different backbones as needed, without requiring all models to be loaded simultaneously on the GPU.
>
> **Q3.2 Comparison of Memory and time cost with other ensembles**
>
> We thank the reviewer for bringing up this point, which we believe is a valuable insight to add to our paper. In terms of computation and time cost SL and MoE is similar to ours, NLC, in the sense that they do not require to run all the backbones simultaneously. Altenatively, one can also run the backbones sequentially and store the feature representations and logits in memory. Thus, all of these models add the timecost of the forward-pass of N backbones.
>     NLC is an MLP with one hidden layer that receives the concatenation of the features and outputs a scalar per backbone to the ensemble. In contrast, SL is a vector of the size of the number of backbones in the ensemble. Therefore, the memory requirements for SL are smaller compared to NLC. Conversely, MoE is marginally more expensive than NLC in terms of memory consumption. Please see the Table below.
>
>
> | MODEL               	| Time to process 1 image in Seconds 	|
> |:---------------------:|:-------------------------------------:|
> | RN50                	| 0.0007                              	|
> | RN50x4              	| 0.0015                              	|
> | RN50x16             	| 0.0035                              	|
> | RN50x64             	| 0.0087                              	|
> | RN101               	| 0.0007                              	|
> | ViT-B-32            	| 0.0003                              	|
> | ViT-B-16            	| 0.0007                              	|
> | ViT-L-14            	| 0.0024                              	|
> | ViT-L-14-336        	| 0.0058                              	|
> |                     	|                                     	|
> | NLC                 	| 0.0243                              	|
>
> | Comparison NLC vs Model       	| Delta Time (NLC - Backbone)	|
> |:---------------------:|:-------------------------------------:|
> | NLC vs RN50x64      	| 0.0156                              	|
> | NLC vs ViT-L-14-336 	| 0.0185                              	|

---

> > ### Comment · Reviewer_Gar3 · 2024-11-27
> >
> > Thanks to the authors for their detailed response. Some follow-up questions:
> >
> > 1) Regarding inference implementation: I understand that after NLC selects the backbone, we only need to load one backbone for inference on one instance. But considering inferencing on a series of instances, the NLC module needs to reselect the backbone for each instance, and how would the authors implement it? From my knowledge one needs either pre-extracting features from all backbones, or loading/off-loading backbones multiple times at inference time.
> >
> > 2) Given that the authors proposed a general idea -- diversity always helps when performing instance-level ensembling, why is CLIP the focus of this paper? From my knowledge the advantage of CLIP lies in its flexibility in tackling free-form classes/concepts/prompts, and applying NLC restricts it to a closed set. Then why is CLIP, rather than other models the focus?
> >
> > 3) The time cost is very considerable compared to that of single backbones (5x~34x). I understand this might be the case for all instance-level ensemble methods, still I wonder how much it is worth for real-world uses. Considering this might be an issue of all methods in this field the authors can opt not to answer.
> >
> > 4) W3 is not answered. The method (NLC) seems to be very technically similar to a previous method SL, also with similar performance. Can the authors elaborate more on the difference between NLC and SL and highlight the improvements?

---

> > > ### Author Response · Authors · 2024-11-27
> > >
> > > **1. How does NLC reselect the backbones for each instance?**
> > >
> > > We would like to begin by pointing out that we think there is a misunderstanding regarding how our approach works, which is suggested by the statement *"I understand that after NLC selects the backbone, we only need to load one backbone for inference on one instance."*, made by the reviewer. NLC does not select a given backbone exclusively. It actually combines the logits of all the backbones together. However, when we combine NLC with the cascading approach, the backbones are evaluated one at a time depending on the confidence (probability) of the prediction, Sec A.
> > >
> > > More concretely, let us first consider a series of instances at inference time grouped in batches. We pass the batch of instances through each backbone $i$, storing the image features $F_i$ and logits $l_i$ in memory or disk. Then, the logits are calibrated by considering the image features of other backbones using our NLC. NLC receives the concatenation of features $F_i$ and produces a vector $\alpha \in \mathbb{R}^{N}$ where $N$ is the number of backbones. Finally, we combine the logits by a weighted sum $\sum_i^N \alpha_i l_i$ creating a new logit $\hat{L}$ and the new probability $P(y|x_i) = softmax(\hat{L})$.
> > >
> > > When we apply the Cascade approach to NLC, the process is a bit different. First, we sort the backbones with respect to their computational requirements from smaller to bigger. Then, we apply the following algorithm.
> > >
> > > ```
> > > We sort the backbones with respect to their computational requirements
> > > For all the batches:
> > >     Initialize list of features
> > >     Initialize list of logits
> > >     For each backbone i:
> > >         1. Extract features and logits for the instances in the batch
> > >         Append the features_i to the list of features
> > >         Append the logits_i to the list of logits
> > >         2. Apply NLC using the current list of features and logits and generate a new logit $\hat{L}$
> > >         3. Compute the probability based on $\hat{L}$
> > >         4. If the probability is big enough
> > >         - Remove that instance from the batch
> > >         - Store the label for that instance
> > >         5. If there are no more instances in the batch, then
> > >         - Exit the Loop
> > > ```
> > >
> > > Although this process can also be done by pre-extracting all the features and logits from all the backbones, as you said, in that case, this would not save the computational cost shown in Figure 3.
> > >
> > > **2. Given that the authors proposed a general idea -- diversity always helps when performing instance-level ensembling, why is CLIP the focus of this paper?**
> > >
> > > The reason why we focus on CLIP in this work is because the research landscape on CLIP offers a unique arena in vision-and-language. There, we find models where *exactly* one variable is changed at a time,e.g., the vision encoder. As you stated, our idea is, in principle, independent of the model: wherever the predictions of model instances are diverse, there is a chance NLC can offer superior performance compared to normal ensembles.
> > >
> > > Thanks to OpenCLIP, we could study how different sources of model prediction variability relate to diversity in Section 2.2. Our results show that for CLIP models, changes in Backbones lead to the highest prediction diversity. This suggests that NLC will prove more effective when we use CLIP models with different backbones (image encoders), as this leads to greater diversity.
> > >
> > > **3. The time cost is very considerable compared to that of single backbones (5x~34x).***
> > >
> > > Indeed, this is an issue for all the ensemble methods.
> > >
> > > **4. Can the authors elaborate more on the difference between NLC and SL and highlight the improvements?**
> > >
> > > SuperLearner (SL) optimizes a set of parameters $\alpha$ using a validation set in order to calibrate the logits and maximize the overall performance on the validation data. While this is indeed similar to our proposal, one key point difference is that SL is static: it is unable to adapt to individual inputs. In contrast, NLC adapts dynamically to image features, leveraging diversity in an adaptive manner and allowing NLC to determine which backbone to rely on the most for a given image. Consequently, not only does NLC consistently outperform SL (Table C.1), but we also observe that the performance gap with SL is larger in datasets with high prediction diversity (Figure 2). For instance, NLC achieves significantly higher accuracy than SL in Eurosat, CLEVR, GTSRB, and PCAM, with a difference of 33.6%, 25.4%, 15.8% and 13.6% in accuracy. We thank the reviewer for bringing up this point. We have also added these to the revised paper (L89).
> > >
> > > | Dataset | SL   | NLC  | Delta Accuracy (NLC - SL) |
> > > |:---------:|:------:|:------:|---------------------------:|
> > > | EUROSAT | 53.5 | 87.1 |                      33.6 |
> > > | CLEVR   |   29 | 54.4 |                      25.4 |
> > > | GTSRB   | 54.7 | 70.5 |                      15.8 |
> > > | PCAM    |   71 | 84.9 |                      13.9 |

---

> > > > ### Comment · Reviewer_Gar3 · 2024-11-28
> > > >
> > > > Thanks for the authors' response. Despite my concerns regarding practicalness for real-world use, this is a common issue of similar works and I will not stick to it. I will update my score in recognition of the extensive study presented in the paper.

---

### Author Response · Authors · 2024-11-26

Thank you all for your constructive feedback. We appreciate your recognition of the systematic approach and the value of our proposed Neural Logit Controller (NLC). Your comments regarding the model performance, dataset variability, and the effects of supervision are insightful. We will address your concerns in the revised manuscript by providing clearer explanations and by further clarifying the role of diversity in predictions, the supervision signal, and the differences observed across datasets. We look forward to refining our approach based on your suggestions.

---

### Meta-Review · Area_Chair_RJYC · 2024-12-20

**Metareview:**

Summary
The paper investigates different CLIP models in terms of the model size and classification accuracy. The key finding of this work is that while overall accuracy is correlated with model size, there are certain samples the smaller models classify accurately vs. the larger models. The authors propose a Neural Logit Controller (NLC) to combine different predictions from different CLIP models. The approach is evaluated on multiple different datasets.

Strengths
1. The key finding in this paper of smaller models helping larger models is worth reiterating especially in the context of the common ML "wisdom" of larger models are better. The solution (NLC) is simple and effective and is likely going to benefit the community.
2. Good experimental setup. The authors analyzed a variety of different OpenCLIP models.
3. The paper is well written and easy to follow.

Weaknesses
1. The primary task studied in this work is image classification. It would make the paper stronger if the authors studied a more diverse range of tasks.
2. NLC requires late fusion and thus the compute cost scales with the number of models being ensembled

Justification
The paper received a majority positive reviews. The paper is technically strong, simple, studies a relevant problem and makes a good contribution to the field.

**Additional Comments On Reviewer Discussion:**

Two of the reviewers raised their rating after the rebuttal. Overall the authors did a great job of addressing the concerns of the reviewers.

---

### Decision · Program_Chairs · 2025-01-22

Accept (Poster)